# CAUSAL ATTENTION WITH LOOKAHEAD KEYS

## ABSTRACT

In standard causal attention, each token's query, key, and value (QKV) are static and encode only preceding context. We introduce CAuSal aTtention with Lookahead kEys (CASTLE), an attention mechanism that continually updates each token's keys as the context unfolds. We term these updated keys lookahead keys because they belong to earlier positions yet integrate information from tokens that appear later relative to those positions, while strictly preserving the autoregressive property. Although the mechanism appears sequential, we derive a mathematical equivalence that avoids explicitly materializing lookahead keys at each position and enables efficient parallel training. On language modeling benchmarks, CASTLE consistently outperforms standard causal attention across model scales, reducing validation perplexity and improving performance on a range of downstream tasks.

## 1 INTRODUCTION

Causal attention (Vaswani et al., 2017) is a cornerstone of autoregressive sequence modeling, allowing each token to condition on its past while preserving the autoregressive structure that underpins language generation. Building on this mechanism, large language models (LLMs) have transformed natural language processing by scaling up the model size and the number of tokens trained (Radford et al., 2019; Brown et al., 2020; Kaplan et al., 2020). While this trend has delivered impressive capabilities, it increasingly runs up against a practical bottleneck, that is high-quality tokens. This reality makes it imperative to improve the attention layer itself and to develop model architectures that are more token-efficient, delivering stronger performance under fixed training-token budgets.

In standard causal attention, each token's query, key, and value (QKV) are computed from that token's representation and remain fixed; they cannot incorporate information from subsequent tokens. As a result, a token's QKV can encode only its preceding context. Recent work shows that such causal mask, which blocks each token's access to its future information, limits models' ability to capture global context, and impairs natural language understanding (Li & Li, 2023; Du et al., 2022; Springer et al.; Xu et al., 2024; Zhang et al., 2025). Here are some vivid illustrations that give intuitions for the limitations of causal masking. Garden-path sentences (Pritchett, 1987) are structurally ambiguous, often inducing an incorrect initial parse. For example, "the old man the boat". Because garden-path sentences' correct interpretation typically depends on information that appears later in the sentence, the causal mask restricting tokens to past context can cause models to struggle in resolving such ambiguities effectively (Amouyal et al., 2025). In many tasks, the question/focus appears at the end of the input. Without access to future context, earlier tokens cannot encode the relevant information needed to anticipate the question/focus. As a result, early token representations may fail to capture important cues and global context dependencies (Xu et al., 2024).

To address the shortcomings of standard causal attention in pretraining, we propose a novel attention mechanism, CAuSal aTtention with Lookahead kEys (CASTLE). In this approach, when generating the $(t + 1)$-th token, we update keys of all preceding tokens so that keys of token $s$ ($1 \leq s \leq t$) are able to additionally encode information from token $s + 1$ to $t$. These keys are called lookahead keys. This design preserves the autoregressive structure while allowing the keys to evolve as the context unfolds. In Figure 1, we give an illustration of receptive fields of the keys in CASTLE. Although the mechanism appears recurrent, we establish a mathematical equivalence that avoids explicitly materializing the lookahead keys and enables efficient parallel training. We evaluate our approach on language modeling across multiple model scales. Experimental results show that CASTLE consistently outperforms standard causal attention in terms of validation perplexity and downstream

task performance. We further introduce a variant, CASTLE-SWL, which applies sliding windows to lookahead keys. CASTLE-SWL has the same complexities with CASTLE in training and inference and preserves the performance benefits of CASTLE while further improves efficiency in practice.

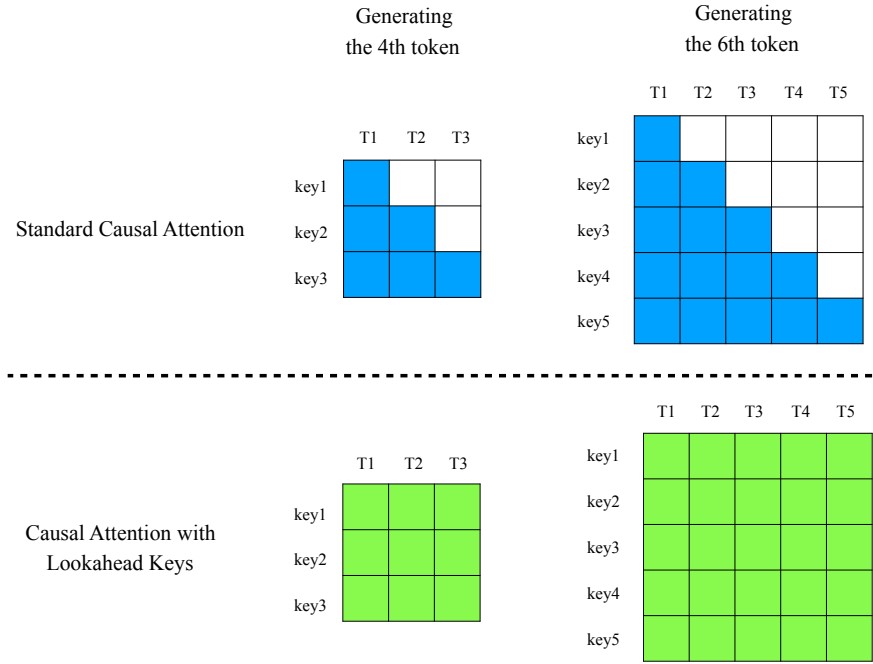

Figure 1: Receptive fields of keys in standard causal attention and CASTLE. The top row shows standard causal attention when generating the 4th token (left) and the 6th token (right). The bottom row shows CASTLE under the same settings. Here, key1, key2, $\cdots$ denote the keys corresponding to tokens 1, 2, $\cdots$, while $T_1, T_2, \cdots$ denote the tokens. In standard causal attention, keys are static: when generating the $(t + 1)$-th token, each key $i$ with $1 \leq i \leq t$ can only access information from $\{T_1, \cdots, T_i\}$, and key $i$ remains the same for all later steps. In contrast, CASTLE continuously updates keys at each prediction step, i.e., when generating the $(t + 1)$-th token, the receptive field of any key $i$ with $1 \leq i \leq t$, is expanded to contain information from $\{T_1, \cdots, T_t\}$.

The remainder of this paper is organized as follows. We discuss related work in Section 1.1. In Section 2.1, we elaborate on our motivations. In Section 2.2, we formally define CASTLE in its recurrent form. Section 2.3 proves an equivalent parallel formulation of CASTLE and develops efficient parallel training algorithms. Section 2.4 introduces efficient inference algorithms together with the counterpart of the KV cache in CASTLE. Finally, Section 3 presents empirical results demonstrating the effectiveness of CASTLE across diverse tasks and model scales.

## 1.1 RELATED WORK

Several studies have observed that the causal mask, by preventing tokens from accessing future information, can hinder a model's ability to capture global context and degrade its natural language understanding (Li & Li, 2023; Du et al., 2022; Springer et al.; Xu et al., 2024; Zhang et al., 2025).

Much effort has been made to overcome this shortcoming in sentence embedding. BeLLM (Backward Dependency Enhanced LLM) (Li & Li, 2023) modifies decoder layers to enable sentence embeddings to integrate both past and future information. This yields substantial improvements on semantic textual similarity and downstream tasks. Echo Embeddings (Springer et al.) duplicate the input sequence and extract embeddings from the second occurrence, letting early tokens attend to later ones without modifying the architecture. Similarly, Re-Reading (RE2) (Xu et al., 2024) prompts models to process inputs twice, so the second pass captures global information obtained in the first. These methods improve embedding quality and reasoning, but their benefits in large-scale pretraining remain unclear.

Selective Attention (Leviathan et al., 2024) introduces a parameter-free modification where tokens can mask out irrelevant or outdated tokens from future attention. By subtracting accumulated selection scores from the attention logits, selective attention reduces reliance on unneeded context. As a result, it achieves substantial memory and compute savings without degrading perplexity. However, selective attention primarily emphasizes filtering unneeded past tokens to enhance efficiency. As discussed in the introduction, many scenarios, such as garden-path sentences or cases where the key information appears at the end of the input, require mechanisms that actively incorporate crucial future information. Whether selective attention can address this challenge remains uncertain.

PaTH Attention (Yang et al., 2025) is a novel positional encoding scheme which introduces data-dependent encodings based on cumulative products of Householder-like transformations. Each transformation is conditioned on the input, enabling the model to dynamically adapt positional information as the sequence progresses. PaTH is related to CASTLE in the sense that in inference formulation of PaTH, keys are also updated via a rank-1 update. However, both the update mechanism of keys and the parallel training formulation in PaTH differ substantially from those in CASTLE, and PaTH remains fundamentally a positional encoding, making it orthogonal to our approach.

Encoder-only Next Token Prediction (ENTP) (Ewer et al., 2024) performs next-token prediction with encoder-only Transformers, where the keys are naturally re-computed at each position. It demonstrates stronger sample efficiency on small-scale language modeling and in-context learning benchmarks. However, the per-token compute in ENTP scales quadratically with sequence length and cubically over full sequences, which presents challenges for scaling.

## 2 CAUSAL ATTENTION WITH LOOKAHEAD KEYS

Our motivations are discussed in Section 2.1. Formal mathematical definitions of CASTLE in recurrent form are provided in Section 2.2. Direct application of the recurrent form of CASTLE in Section 2.2 is impractical for large-scale pretraining. To address this, we present efficient pretraining algorithms in Section 2.3. In Section 2.4, we describe efficient inference algorithms along with the counterparts of the KV cache in CASTLE.

### 2.1 MOTIVATIONS

To explain our motivations, we begin with the recurrent form of standard causal attention. Consider the case when generating the $(t + 1)$-th token. Given contextualized representations $\boldsymbol{X}^t = (\boldsymbol{x}_1^\top, \cdots, \boldsymbol{x}_t^\top)^\top \in \mathbb{R}^{t \times d_{\text{model}}}$, where the $s$-th row $\boldsymbol{x}_s$ is the representation of token $s$. Unless otherwise specified, all vectors in this paper are treated as row vectors rather than column vectors.

The query, key and value of token $s$ are $\boldsymbol{q}_s = \boldsymbol{x}_s \boldsymbol{W}_Q$, $\boldsymbol{k}_s = \boldsymbol{x}_s \boldsymbol{W}_K$, $\boldsymbol{v}_s = \boldsymbol{x}_s \boldsymbol{W}_V$. We also denote $\boldsymbol{K}_t = \boldsymbol{X}^t \boldsymbol{W}_K$ and $\boldsymbol{V}_t = \boldsymbol{X}^t \boldsymbol{W}_V$. Then, the standard causal attention follows the recurrent form

$$\text{causal-attention}(\boldsymbol{X}^t) = \text{softmax}\left(\frac{\boldsymbol{q}_t \boldsymbol{K}_t^\top}{\sqrt{d}}\right) \boldsymbol{V}_t = \frac{\sum_{s=1}^t \exp\left(\boldsymbol{q}_t \boldsymbol{k}_s^\top / \sqrt{d}\right) \boldsymbol{v}_s}{\sum_{s=1}^t \exp\left(\boldsymbol{q}_t \boldsymbol{k}_s^\top / \sqrt{d}\right)} \in \mathbb{R}^{1 \times d}. \quad (1)$$

Due to the autoregressive structure, each $\boldsymbol{x}_s$ only encodes information from token 1 to $s$. Thus, when generating token $t + 1$ with $t + 1 > s$, each $\boldsymbol{k}_s$ only contains information from token 1 to $s$ without containing information from token $s + 1$ to $t$. This can impair models' ability of natural language understanding, yielding high-quality text embedding and capturing global context as mentioned in the introduction.

This motivates us to propose a novel attention mechanism, causal attention with lookahead keys (CASTLE), i.e., when generating the $(t + 1)$-th token, we first update keys $\boldsymbol{k}_s$ of preceding tokens $s$ with $s < t + 1$ to additionally incorporate information from token $s + 1$ to $t$. We refer to these as *lookahead keys* because their representations renew with the growing context. In this way, lookahead keys may lead to more accurate attention scores while preserving the autoregressive property.

Before describing the details of this mechanism, we first answer the following questions.

• **Why do we use lookahead keys instead of lookahead queries?** The answer parallels the reason why key–value (KV) pairs are cached instead of queries (Q). Each $\boldsymbol{q}_s$ is only used once, namely when generating token $s+1$. Because we are designing an autoregressive model, past queries cannot

be modified after generation, making it meaningless to update $\boldsymbol{q}_s$. In contrast, $\boldsymbol{k}_s$ is multiplied by the queries $\boldsymbol{q}_t$ of all subsequent tokens $t \geq s$. Keeping $\boldsymbol{k}_s$ updated therefore can benefit all later tokens by possibly producing more accurate attention scores.

We also remark that updating the values $\boldsymbol{v}_s$ similarly to lookahead keys could be beneficial while its efficient algorithm remains future study.

• **How do we maintain the model autoregressive with lookahead keys?** When generating token $t + 1$, we update keys of each preceding token $s < t + 1$ with information from token $s + 1$ to $t$. Thus, all keys only contain information from token 1 to $t$. Queries and values are naturally only containing information from tokens up to $t$. No future information from tokens $k > t$ is used. Thus, the model maintains autoregressive property.

Further details of our design are presented in Section 2.2.

## 2.2 MATHEMATICAL DEFINITION IN RECURRENT FORM

We give mathematical definitions of CASTLE in this section. Let $L$ denote sequence length and $d_{\text{model}}$, $d$ denote the hidden dimension and head dimension, respectively.

Throughout Section 2.2, we fix $t \in \{1, 2, \cdots, L\}$ and consider the setting where $t$ tokens have been generated and the model is generating the $(t + 1)$-th token. Denote the input contextualized representations $\boldsymbol{X}^t = (\boldsymbol{x}_1^\top, \cdots, \boldsymbol{x}_t^\top)^\top \in \mathbb{R}^{t \times d_{\text{model}}}$, where $\boldsymbol{x}_s$ is the representation of token $s$.

Utilizing lookahead keys lies at the core of CASTLE. However, the way a model learns to encode information into keys of token $s$ from past tokens ($k \leq s$) may differ from how it encodes information from subsequent tokens (tokens $s < k < t+1$) when generating the $(t+1)$-th token. To address this, we adopt a hybrid design. Specifically, we keep half of the keys the same as in standard causal which we call *causal keys*, while allowing the remaining half to renew as the context progresses which we call *lookahead keys*.

For each preceding token $s$ ($1 \leq s < t + 1$), causal keys of token $s$ is a projection of $\boldsymbol{x}_s$, while lookaheads keys of token $s$ contain information from representations $\{\boldsymbol{x}_{s+1}, \cdots, \boldsymbol{x}_t\}$. The receptive fields of causal keys and lookahead keys are illustrated in Figure 10.

We first project $\boldsymbol{X}^t$ into key and value matrices $\boldsymbol{K}_t^U$, $\boldsymbol{V}_t^U$, $\boldsymbol{K}_t^C$, $\boldsymbol{V}_t^C \in \mathbb{R}^{t \times d}$ by $\boldsymbol{K}_t^U = \boldsymbol{X}^t \boldsymbol{W}_K^U$, $\boldsymbol{V}_t^U = \boldsymbol{X}^t \boldsymbol{W}_V^U$, and $\boldsymbol{K}_t^C = \boldsymbol{X}^t \boldsymbol{W}_K^C$, $\boldsymbol{V}_t^C = \boldsymbol{X}^t \boldsymbol{W}_V^C$ as well as query matrix $\boldsymbol{Q}_t^U = \boldsymbol{X}^t \boldsymbol{W}_Q^U \in \mathbb{R}^{t \times d}$ and query vector $\boldsymbol{q}_t^C = \boldsymbol{x}_t \boldsymbol{W}_Q^C \in \mathbb{R}^{1 \times d}$. Here, $\boldsymbol{W}_Q^U$, $\boldsymbol{W}_K^U$, $\boldsymbol{W}_V^U$, $\boldsymbol{W}_Q^C$, $\boldsymbol{W}_K^C$, $\boldsymbol{W}_V^C \in \mathbb{R}^{d_{\text{model}} \times d}$ are learnable matrices.

The matrices $\boldsymbol{K}_t^U$, $\boldsymbol{V}_t^U$, $\boldsymbol{Q}_t^U$ are used to generate the lookahead key $\boldsymbol{U}^t$. Then, the causal key $\boldsymbol{K}_t^C$ and the lookahead key $\boldsymbol{U}^t$ are multiplied by the query vector $\boldsymbol{q}_t^C$ to get the attention scores. Then, $\boldsymbol{V}_t^C$ are multiplied by the attention weights to get the output. Before we elaborate on details in the definition of CASTLE, we first give formal definitions of causal keys and lookahead keys.

**Causal Keys.** The causal keys in CASTLE is defined similarly to the keys in standard causal attention. More specifically, causal keys are defined as $\boldsymbol{K}_t^C = (\boldsymbol{k}_1^{C\top}, \cdots, \boldsymbol{k}_t^{C\top})^\top = \boldsymbol{X}^t \boldsymbol{W}_K^C \in \mathbb{R}^{t \times d}$. The $s$-th row $\boldsymbol{k}_s$ of $\boldsymbol{K}_t^C$ satisfying $\boldsymbol{k}_s = \boldsymbol{x}_s \boldsymbol{W}_K^C$ is the causal key of token $s$. Causal keys are static, i.e., the $s$-th rows of $\boldsymbol{K}_t^C$ and $\boldsymbol{K}_{t'}^C$ are equal to each other whenever $t, t' \geq s$.

**Lookahead Keys.** We utilize a structure similar to the attention mechanism to define lookahead keys. An illustration for lookahead keys can be found in Figure 2.

More specifically, the lookahead keys are defined as

$$\boldsymbol{U}^t = (\boldsymbol{u}_1^{t\top}, \cdots, \boldsymbol{u}_t^{t\top})^\top = \text{sigmoid}\Big(\frac{\boldsymbol{Q}_t^U \boldsymbol{K}_t^{U\top}}{\sqrt{d}} + \boldsymbol{M}_t^U\Big) \boldsymbol{V}_t^U \in \mathbb{R}^{t \times d}, \tag{2}$$

where $\boldsymbol{M}_t^U \in \mathbb{R}^{t \times t}$ is a mask matrix with

$$[\boldsymbol{M}_t^U]_{ij} = 0 \text{ if } i < j \text{ and } [\boldsymbol{M}_t^U]_{ij} = -\infty \text{ otherwise.} \tag{3}$$

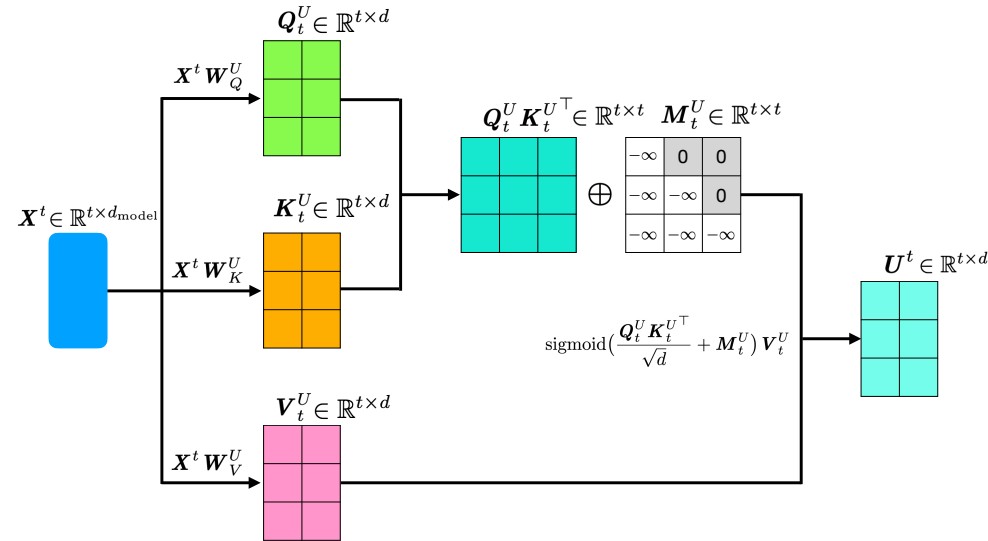

Figure 2: Illustration for the definition of lookahead keys in (2) when generating the 4-th token. Here $d_{\text{model}}$ and $d$ denote the hidden dimension and head dimension, respectively. In this figure, we set $t = 3$ and $d = 2$.

The $s$-th row $\boldsymbol{u}_s^t$ of $\boldsymbol{U}^t$ is the lookahead key of token $s$. We remark that in $\boldsymbol{u}_s^t$, the superscript $t$ indicates that $\boldsymbol{u}_s^t$ is defined when $t$ tokens have already been generated and we are about to generate the $(t+1)$-th token, while the subscript $s$ indicates $\boldsymbol{u}_s^t$ is the $s$-th row of $\boldsymbol{U}^t$.

The definition of $\boldsymbol{M}_t^U$ guarantees that the lookahead key of token $s$, $\boldsymbol{u}_s^t$, is exposed to information from $\{\boldsymbol{x}_{s+1}, \cdots, \boldsymbol{x}_t\}$. Since $\boldsymbol{u}_s^t$ keeps renewing as the context goes, it is natural that $\boldsymbol{u}_s^t \neq \boldsymbol{u}_s^{t+1}$.

We have the following remarks regarding the definition of lookahead keys in (2).

• **Why are we using sigmoid?** The sigmoid function is used in (2) instead of softmax due to the consideration that when generating token $t+1$, for token $s$ with $s < t+1$, synthesizing information contained in tokens $s+1$ to $t$ should not be compulsory. However, since the probabilities in softmax sum up to 1, which forces $\boldsymbol{u}_s^t$ to incorporate information from tokens $s+1$ to $t$ and is not desired.

• **Lookahead keys $\boldsymbol{U}^t$ maintains autoregressive property.** First, CASTLE is defined in a recurrent form which is naturally autoregressive. Second, when generating the $(t+1)$-th token, each $\boldsymbol{u}_s^t$ is only exposed to information from representations of tokens $s+1$ to $t$ as in (2). No information from tokens which are not yet generated is exposed.

• **Lookahead keys $\boldsymbol{U}^t$ only occur in CASTLE's recurrent form definition and inference, but cannot be materialized in parallel training.** Since $\boldsymbol{u}_s^t$ and $\boldsymbol{u}_s^{t+1}$ may vary, this prevents us from materializing $\boldsymbol{U}^t$ for each $t$. The computation cost in (2) is $O(t^2 d)$. If we materialize all $\boldsymbol{U}^t$ in parallel, the computational cost is at least $\sum_{t=1}^{L} t^2 d = O(L^3 d)$ which makes training on large-scale datasets impractical. In Section 2.3, we will give an equivalent form which removes the need of materializing each $\boldsymbol{U}^t$ and enables efficient parallel training.

**CASTLE in Recurrent Form.** After defining causal keys and lookahead keys, we are ready to give the formula of CASTLE in recurrent form. To generate the $(t+1)$-th token, we utilize both the causal keys $\boldsymbol{K}_t^C \in \mathbb{R}^{t \times d}$ and the lookahead keys $\boldsymbol{U}^t$. An illustration of CASTLE in its recurrent form can be found in Figure 3. More specifically, let the *causal-key attention scores* be

$$\boldsymbol{s}_t^C = \frac{\boldsymbol{q}_t^C \boldsymbol{K}_t^{C\top}}{\sqrt{d}} \in \mathbb{R}^{1 \times t}. \tag{4}$$

Let the *lookahead-key attention scores* be

$$\boldsymbol{s}_t^U = \frac{\boldsymbol{q}_t^C \boldsymbol{U}^{t\top}}{\sqrt{d}} \in \mathbb{R}^{1 \times t}. \tag{5}$$

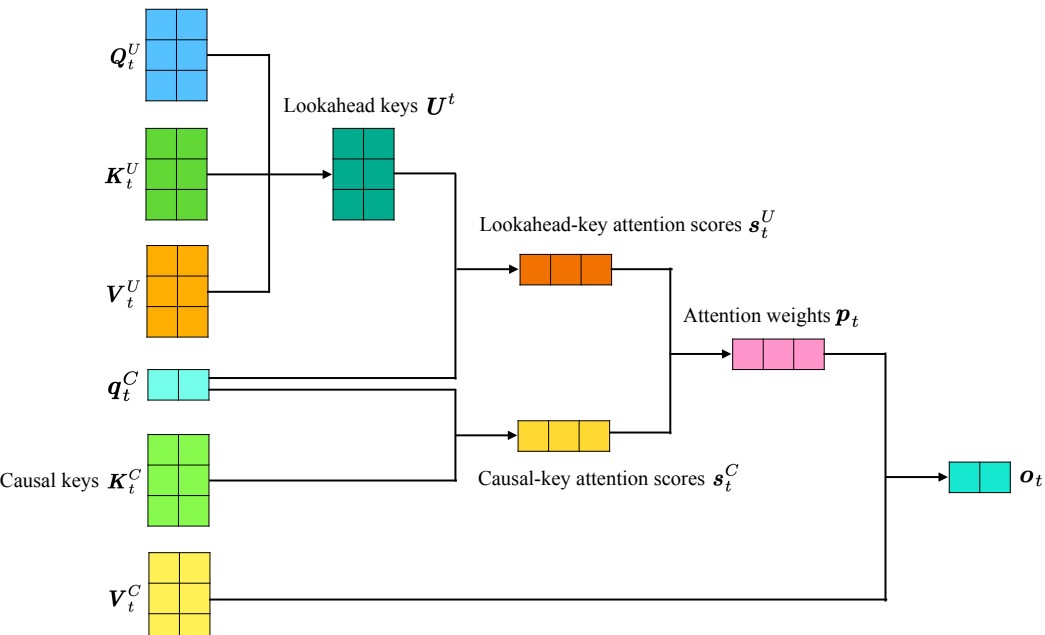

Figure 3: Illustration of CASTLE in recurrent form when generating the 4th token.

Then, we define attention weights by combining the above attention scores as follows

$$\boldsymbol{p}_t = \text{softmax}\left(\boldsymbol{s}_t^C - \text{SiLU}\left(\boldsymbol{s}_t^U\right)\right) \in \mathbb{R}^{1 \times t}, \tag{6}$$

where $\text{SiLU}(\boldsymbol{x}) = \boldsymbol{x} \odot \text{sigmoid}(\boldsymbol{x})$. Then, the output is calculated as

$$\text{attention}\left(\boldsymbol{X}^t\right) = \boldsymbol{p}_t \boldsymbol{V}_t^C \in \mathbb{R}^{1 \times d}. \tag{7}$$

We remark that SiLU is applied in (6) because our ablation study indicates that it plays a crucial role in ensuring training stability. We give a comprehensive study of different variants of (6) and why we use SiLU in Appendix A.5.3.

**CASTLE-SWL in Recurrent Form.** We propose a variant of CASTLE, termed CASTLE-SWL, where we apply Sliding Window to Lookahead keys. More specifically, denote sliding window size by $W$. When generating the $(t + 1)$-th token, lookahead keys of token $s$ $(s < t + 1)$ have access to information from $\{\boldsymbol{x}_k : s + 1 \le k \le \min\{s + W, t\}\}$. Formally, the definition of lookahead keys in CASTLE-SWL follows (2) with $\boldsymbol{M}_t^U$ defined as

$$[\boldsymbol{M}_t^U]_{ij} = 0 \text{ if } i < j \le \min\{t, i + W\} \text{ and } [\boldsymbol{M}_t^U]_{ij} = -\infty \text{ otherwise.} \tag{8}$$

An illustration for $\boldsymbol{M}_t^U$ applied to lookahead keys in CASTLE-SWL can be found in Figure 11. Sliding window modifies only lookahead keys while causal keys remains unchanged. Apart from changing the definition of $\boldsymbol{M}_t^U$ from (3) to (8), all other components of CASTLE remain the same, yielding the definition of CASTLE-SWL. The motivation for this design is that the semantic contribution of tokens typically decays as the context unfolds. Allowing lookahead keys to aggregate information from distant tokens may introduce noise rather than useful signal. In addition, although CASTLE-SWL has the same complexities with CASTLE in both training and inference as analyzed in Section 2.3 and 2.4, it can reduce FLOPs and further improves efficiency in practice.

## 2.3 EFFICIENT PARALLEL TRAINING

In this section, we introduce our efficient parallel training algorithm. As discussed in Section 2.2, a straightforward materializing each $\boldsymbol{U}^t$ in parallel incurs at least $O(L^3 d)$ computational costs. Such complexity makes training on large-scale datasets infeasible. In addition, it is hard to derive efficient parallel training algorithm based on the reccurrent formulation introduced in Section 2.2

above. To address this, we derive an equivalent parallel formulation of CASTLE and CASTLE-SWL that eliminates the need to materialize lookahead keys. This formulation then serves as the basis for designing an efficient algorithm.

**CASTLE and CASTLE-SWL in Parallel Form.** Let attention $(X^t) \in \mathbb{R}^{1 \times d}$ denote the output when generating the $(t+1)$-th token as in (7). Then, given the inputs $X^L \in \mathbb{R}^{L \times d_{\text{model}}}$, the concatenated outputs are denoted by

$$\text{Attention}(X^L) = \left(\text{attention}(X^1)^\top, \cdots, \text{attention}(X^L)^\top\right)^\top \in \mathbb{R}^{L \times d_{\text{model}}}. \quad (9)$$

The following theorem provides a unified parallel formulation for both CASTLE and CASTLE-SWL that is equivalent to their recurrent forms introduced in Section 2.2. Its proof is in Appendix B.

**Theorem 1.** *Consider inputs $X^L \in \mathbb{R}^{L \times d_{\text{model}}}$, where $L$ is the sequence length and $d_{\text{model}}$ is the hidden dimension. Let $Q^U = X^L W_Q^U$, $K^U = X^L W_K^U$, $V^U = X^L W_V^U$, $Q^C = X^L W_Q^C$, $K^C = X^L W_K^C$, $V^C = X^L W_V^C$. Define matrix $S^U \in \mathbb{R}^{L \times L}$ as*

$$S^U = \left(\frac{Q^C V^{U\top}}{\sqrt{d}} \odot \widetilde{M}^C\right) \left(\text{sigmoid}\left(\frac{Q^U K^{U\top}}{\sqrt{d}} + M^U\right)\right)^\top. \quad (10)$$

*Then, the outputs $\text{Attention}(X^L)$ as in (9) satisfies that*

$$\text{Attention}(X^L) = \text{row\_softmax}\left(\frac{Q^C K^{C\top}}{\sqrt{d}} + M^C - \text{SiLU}(S^U)\right) V^C. \quad (11)$$

*Here, $M^C$, $\widetilde{M}^C$ are the causal masks which prevent tokens from attending to their future tokens, i.e., $M_{ij}^C = 0$ if $i \geq j$ and $M_{ij}^C = -\infty$ otherwise; $\widetilde{M}_{ij}^C = 1$ if $i \geq j$ and $\widetilde{M}_{ij}^C = 0$ otherwise. For CASTLE, $M^U = M_L^U \in \mathbb{R}^{L \times L}$ with $M_L^U$ defined in (3), while for CASTLE-SWL, $M^U = M_L^U \in \mathbb{R}^{L \times L}$ with $M_L^U$ defined in (8).*

**Efficient Parallel Training.** Theorem 1 establishes the equivalence between the recurrent and parallel formulations for CASTLE and CASTLE-SWL. However, computing $\text{Attention}(X^L)$ directly from Theorem 1 still requires $\Omega(L^3)$ complexity, since (10) involves matrix multiplications between $L$-by-$L$ matrices.

To reduce this cost, notice that in (10), the term $\left(Q^C V^{U\top}\right) \odot \widetilde{M}^C$ is a masked low-rank matrix because the matrix $Q^C V^{U\top}$ is of rank $d$ which is typically much smaller than $L$. This structure enables a more efficient computation of $S^U$, which we exploit to design a parallel training algorithm as stated in Theorem 2. The proof of Theorem 2 is given in Appendix D.

**Theorem 2.** *Given $X^L$'s query, key and value matrices $Q^U = X^L W_Q^U$, $K^U = X^L W_K^U$, $V^U = X^L W_V^U$, $Q^C = X^L W_Q^C$, $K^C = X^L W_K^C$, $V^C = X^L W_V^C$, for both CASTLE and CASTLE-SWL, Algorithm 1 (forward pass) and Algorithm 3 (backward pass) enable efficient parallel training and can compute $\text{Attention}(X^L)$ and the gradients with computational complexity $O(L^2 d)$ and space complexity $O(Ld)$, i.e., both forward pass and backward pass have computational complexity $O(L^2 d)$ and space complexity $O(Ld)$.*

## 2.4 EFFICIENT INFERENCE WITH UQ-KV CACHE

In this section, we introduce the inference algorithm which has unified forms for CASTLE and CASTLE-SWL. A detailed description of our inference algorithm is provided in Appendix E. We first introduce the decoding algorithm. The decoding algorithm consists of the following 2 steps: *updating step* and *combining step*. Fix $t \in \{1, 2, \ldots, L\}$, and consider generating the $(t+1)$-th token.

**Rank-1 Updating step.** We generate lookahead keys $U^t$ in the updating step. First, we compute $q_t^U = x_t W_Q^U$, $k_t^U = x_t W_K^U$ and $v_t^U = x_t W_V^U$. Next, rather than computing $U^t$ directly from (2),

which requires $O(t^2 d)$ computation, we obtain $\boldsymbol{U}^t$ through rank-1 update from $\boldsymbol{U}^{t-1}$

$$\boldsymbol{U}^t = \left( \boldsymbol{U}^{t-1} + \text{sigmoid}\left( \frac{\boldsymbol{Q}_{t-1}^U \boldsymbol{k}_t^{U\top}}{\sqrt{d}} + [\boldsymbol{M}_t^U]_{1:t-1,t} \right) \boldsymbol{v}_t^U \atop \boldsymbol{0}^{1\times d} \right), \qquad (12)$$

where $[\boldsymbol{M}_t^U]_{1:t-1,t}$ is the $t$-th column of $\boldsymbol{M}_t^U$ removing the last entry. The proof of (12) is given in Appendix E. Next, we discuss the caching strategy and FLOPs of the updating step.

• **Caching in updating step.** First, it is obvious that we need to cache $\boldsymbol{U}^t$ so that we can implement the recursive formula (12). Second, we need to cache $\boldsymbol{Q}_t^U$ because $\boldsymbol{q}_s^U$ is used in any update from $\boldsymbol{U}^{t-1}$ to $\boldsymbol{U}^t$ with $s \leq t-1$. As $\boldsymbol{k}_t^U$ and $\boldsymbol{v}_t^U$ are only used in the update from $\boldsymbol{U}^{t-1}$ to $\boldsymbol{U}^t$ and never used again in update from $\boldsymbol{U}^{j-1}$ to $\boldsymbol{U}^j$ with $j \neq t$, we do not need to cache any other variables except $\boldsymbol{U}^t$ and $\boldsymbol{Q}_t^U$ for the updating step.

• **FLOPs in updating step.** With cached $\boldsymbol{U}^{t-1}$ and $\boldsymbol{Q}_{t-1}^U$, the updating formula (12) needs only $O(td)$ FLOPs. And computing $\boldsymbol{q}_t^U$, $\boldsymbol{k}_t^U$ and $\boldsymbol{v}_t^U$ needs $O(dd_{\text{model}})$ FLOPs, yielding total FLOPs of $O(dd_{\text{model}} + td)$.

**Combining step.** In the combining step, we compute $\boldsymbol{q}_t^C = \boldsymbol{x}_t \boldsymbol{W}_Q^C$, $\boldsymbol{k}_t^C = \boldsymbol{x}_t \boldsymbol{W}_K^C$ and $\boldsymbol{v}_t^C = \boldsymbol{x}_t \boldsymbol{W}_V^C$. Next, the attention outputs are then obtained by applying (4), (5), (6) and (7) with $O(td)$ FLOPs. At this stage, since we already cached $\boldsymbol{U}^t$ in the updating step, only $\boldsymbol{K}_t^C$ and $\boldsymbol{V}_t^C$ need to be stored.

**UQ-KV cache.** From the above analysis, the counterpart of the KV cache in CASTLE consists of $\boldsymbol{U}^t$, $\boldsymbol{Q}_t^U$, $\boldsymbol{K}_t^C$, and $\boldsymbol{V}_t^C$, which we collectively refer to as the *UQ-KV cache*. All other variables, including $\boldsymbol{q}_t^C$, $\boldsymbol{k}_t^U$, and $\boldsymbol{v}_t^U$, can be safely disposed of after use.

For the prefilling stage, let the prompt length be $L$ and inputs $\boldsymbol{X}^L \in \mathbb{R}^{L \times d_{\text{model}}}$. We first compute $\boldsymbol{Q}_L^U = \boldsymbol{X}^L \boldsymbol{W}_Q^U$, $\boldsymbol{K}_L^U = \boldsymbol{X}^L \boldsymbol{W}_K^U$, $\boldsymbol{V}_L^U = \boldsymbol{X}^L \boldsymbol{W}_V^U$, $\boldsymbol{Q}_L^C = \boldsymbol{X}^L \boldsymbol{W}_Q^C$, $\boldsymbol{K}_L^C = \boldsymbol{X}^L \boldsymbol{W}_K^C$, $\boldsymbol{V}_L^C = \boldsymbol{X}^L \boldsymbol{W}_V^C$. Then, we apply the forward pass of the efficient parallel training algorithm (Algorithm 1) to get $\text{Attention}(\boldsymbol{X}^L)$. For the UQ-KV cache, since we already have $\boldsymbol{Q}_L^U$, $\boldsymbol{K}_L^C$ and $\boldsymbol{V}_L^C$, we only need to obtain $\boldsymbol{U}^L$. This can be done similarly to FlashAttention-2 (Dao, 2024) due to the similarity between (2) and standard causal attention. The complete prefilling procedure is given in Algorithm 5. The analysis above leads to the following theorem. Its proof and more detailed introduction of CASTLE's inference is in Appendix E.

**Theorem 3.** *Given inputs $\boldsymbol{X}^L \in \mathbb{R}^{L \times d_{\text{model}}}$, prefilling has computational complexity $O(Ldd_{\text{model}} + L^2 d)$ and space complexity $O(Ld)$. During the decoding stage, when generating the $t$-th token, the computational complexity is $O(td + dd_{\text{model}})$ and the UQ-KV cache requires $O(td)$ memory.*

### 2.5 Multi-Head Causal Attention with Lookahead Keys

As in standard causal attention, we also utilize multi-head mechanism. Let $n$ denote the number of heads. In each head $i$, when generating the $t$-th token, denote the output as in (7) by $\text{attention}_i(\boldsymbol{X}^t) \in \mathbb{R}^{1\times d}$. Then, the output of multi-head causal attention with lookahead keys (multi-head CASTLE) is calculated as

$$\text{multi-head-attention}(\boldsymbol{X}^t) = \text{Concat}\left( \text{attention}_1(\boldsymbol{X}^t), \dots, \text{attention}_n(\boldsymbol{X}^t) \right) \boldsymbol{W}^O \in \mathbb{R}^{1\times d}, \quad (13)$$

where $\boldsymbol{W}^O \in \mathbb{R}^{nd \times d_{\text{model}}}$ is a learnable matrix. The formula for forward pass in parallel training and more details of multi-head CASTLE are in Appendix C. Multi-head CASTLE-SWL shares the same formulation and parameter count as CASTLE, and is omitted here to avoid redundancy.

## 3 Experiments

### 3.1 Experimental Setup

We use the nanoGPT (Karpathy, 2023) code base. Our baseline follows the improved Transformer architecture with SwiGLU (Shazeer, 2020), Rotary Positional Embeddings (Su et al., 2024), and

RMSNorm (Zhang & Sennrich, 2019) following LLaMA (Touvron et al., 2023). We train models on four scales from scratch: small (0.16B), medium (0.35B), large (0.75B) and XL (1.3B). The medium, large and XL baseline models mirror the configuration of GPT-3 (Brown et al., 2020). For the small setting, we increase the number of heads and the hidden dimension relative to the original GPT-3 configuration to better align parameter counts between standard causal attention and CASTLE. To isolate the effect of the attention mechanism, we replace standard causal attention with dynamic attention with evolving keys while keeping all other components unchanged for a fair comparison. We use the AdamW optimizer (Loshchilov & Hutter, 2019) and follow the training recipe of (Dao & Gu, 2024). All models are trained on FineWeb-Edu dataset (Lozhkov et al.) for 50B tokens. Further details of experimental setup can be found in Appendix A.1.

## 3.2 TRAINING & VALIDATION LOSS

We report training and validation loss and perplexity in Table 1. Training loss and validation loss curves are shown in Figure 4, Figure 6, Figure 7 and Figure 8. CASTLE and CASTLE-SWL consistently outperform the baselines in both training and validation loss across all model scales.

Specifically, after training 50B tokens, CASTLE outperforms baselines across all model scales and achieves validation losses that are 0.0059, 0.0245, 0.0356, and 0.0348 lower than the baseline for the small, medium, large, and XL models, respectively. CASTLE-SWL matches CASTLE's performance and its validation loss outperforms baselines by 0.0084, 0.0241, 0.0366, 0.0369, respectively. The performance gains are particularly significant in the medium, large, and XL models. Furthermore, as shown in Table 3, both CASTLE and CASTLE-SWL have the same or fewer parameters than their baseline counterparts. This further underscores CASTLE and CASTLE-SWL's effectiveness in improving model performance.

Table 1: Training and validation loss and perplexity for models with CASTLE, CASTLE-SWL and standard causal attention. We use "S", "M", "L", "XL" to denote model scales. Each model is trained on FineWeb-Edu for 50B tokens. The best and second best results (when showing clear improvements upon baselines) are highlighted in bold and underline, respectively.

|  | $n_{\text{params}}$ | Train | | Eval | |
|---|---|---|---|---|---|
|  |  | Loss | PPL | Loss | PPL |
| Baseline-S | 160M | 2.795 | 16.364 | 2.798 | 16.411 |
| CASTLE-S | 160M | 2.789 | 16.259 | 2.792 | 16.315 |
| CASTLE-SWL-S | 160M | 2.786 | 16.213 | 2.790 | 16.273 |
| Baseline-M | 353M | 2.641 | 14.030 | 2.639 | 14.004 |
| CASTLE-M | 351M | **2.616** | **13.684** | **2.615** | **13.665** |
| CASTLE-SWL-M | 351M | 2.618 | 13.708 | 2.615 | 13.670 |
| Baseline-L | 756M | 2.513 | 12.346 | 2.507 | 12.269 |
| CASTLE-L | 753M | 2.476 | 11.897 | 2.472 | 11.840 |
| CASTLE-SWL-L | 753M | **2.476** | **11.890** | **2.471** | **11.832** |
| Baseline-XL | 1.310B | 2.430 | 11.360 | 2.426 | 11.309 |
| CASTLE-XL | 1.304B | **2.401** | **11.031** | 2.391 | 10.922 |
| CASTLE-SWL-XL | 1.304B | 2.401 | 11.036 | **2.389** | **10.900** |

## 3.3 DOWNSTREAM EVALUATION

We evaluate CASTLE and CASTLE-SWL on a diverse set of downstream benchmarks, including ARC (Yadav et al., 2019), BoolQ (Clark et al., 2019), HellaSwag (Zellers et al., 2019), MMLU (Hendrycks et al., 2020), OBQA (Mihaylov et al., 2018), PIQA (Bisk et al., 2020), Winograde (Sakaguchi et al., 2020) using lm-evaluation-harness (Gao et al., 2024). We report normalized accuracy for ARC-Challenge, HellaSwag, OBQA, and PIQA, and standard accuracy for the other benchmarks. Results are reported in Table 4 (0-shot) and Table 2 (5-shot). Across all model scales and evaluation settings, both CASTLE and CASTLE-SWL consistently outperform the baselines in average downstream accuracy under both 0-shot and 5-shot settings. These findings accompany lower loss and perplexity in Section 3.2 demonstrate that CASTLE and CASTLE-SWL not only

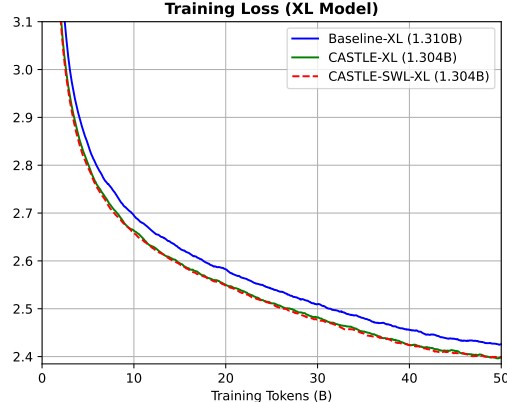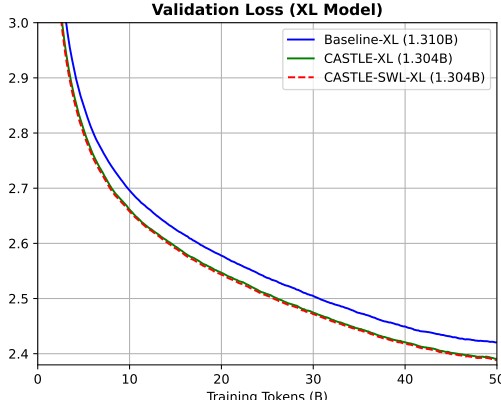

Figure 4: Training and validation loss curves of XL models. Training loss curve is smoothened with a moving window of 2000 training steps. Validation loss is evaluated every 100 training steps on 40M tokens, and its curve is smoothened by a moving window of 20 evaluation intervals. Loss curves for the small, medium and large models can be found in Figure 6, Figure 7 and Figure 8 of Appendix A.6. After 50B training tokens, CASTLE-XL achieves a 0.0294 lower training loss and a 0.0348 lower validation loss compared to Baseline-XL, while CASTLE-SWL-XL achieves a 0.0290 lower training loss and a 0.0369 lower validation loss compared to Baseline-XL

lower perplexity but also translate these gains into stronger performance on diverse downstream tasks.

Table 2: Evaluation results (5-shot) for downstream tasks of different model scales. The higher accuracy values are shown in bold. All values denote accuracy in percentage (%). Each model is pretrained on FineWeb-Edu for 50B tokens. Hella.=HellaSwag, Wino.=Winograde. 0-shot results can be found in Table 4.

| Model Name | ARC-C | ARC-E | BoolQ | Hella. | MMLU | OBQA | PIQA | Wino. | Avg. |
|---|---|---|---|---|---|---|---|---|---|
| Baseline-S | 25.68 | 54.97 | 56.09 | 33.81 | **25.54** | 28.20 | 63.98 | **52.57** | 42.60 |
| CASTLE-S | 26.02 | 54.25 | **57.13** | 35.24 | 25.22 | 29.80 | 64.53 | 50.99 | 42.90 |
| CASTLE-SWL-S | **27.39** | **56.19** | 56.09 | **35.46** | 25.34 | **30.20** | **64.85** | 51.22 | **43.34** |
| Baseline-M | 31.06 | 62.46 | 48.47 | 42.83 | 25.22 | 33.00 | 68.39 | 51.78 | 45.40 |
| CASTLE-M | 32.17 | **64.06** | **54.62** | 43.47 | 25.22 | **33.80** | **69.48** | 52.49 | 46.91 |
| CASTLE-SWL-M | **32.85** | 63.85 | 54.19 | **44.18** | **26.43** | **33.80** | 69.04 | **53.43** | **47.22** |
| Baseline-L | 33.36 | 63.64 | 59.24 | 46.16 | **26.82** | 33.40 | 69.53 | **54.06** | 48.28 |
| CASTLE-L | **37.37** | **67.89** | 50.95 | 47.71 | 26.11 | 34.20 | **70.18** | **54.06** | 48.56 |
| CASTLE-SWL-L | 36.26 | 65.53 | **60.58** | **48.55** | 24.70 | **34.80** | 69.10 | 53.67 | **49.15** |
| Baseline-XL | 35.58 | 65.78 | 61.07 | 50.84 | **26.71** | 36.20 | 71.27 | 52.72 | 50.02 |
| CASTLE-XL | **39.08** | **70.24** | **62.60** | 51.63 | 24.16 | **37.40** | 71.00 | **58.33** | 51.80 |
| CASTLE-SWL-XL | 38.99 | 70.08 | 61.74 | **52.35** | 25.85 | 37.20 | **72.52** | 56.75 | **51.93** |

## 4 CONCLUSION

We introduced CAuSal aTtention with Lookahead kEys (CASTLE), a novel attention mechanism that continually updates keys as the context evolves. This design allows each key representation to incorporate more recent information at every prediction step while strictly preserving the autoregressive property. Although CASTLE is defined recurrently, we derived a mathematical equivalence that eliminates the need to explicitly materialize lookahead keys at each position, enabling efficient parallel training. Experimental results on language modeling demonstrate that CASTLE consistently outperforms standard causal attention, achieving lower perplexity and stronger downstream performance.

## ETHICS STATEMENT

This work adheres to the ICLR Code of Ethics. Our research focuses on developing a new attention mechanism for large language models. It does not involve human subjects, private or sensitive data, or applications with known risks of harm. We are not aware of any negative social impacts in our results.

## USE OF LARGE LANGUAGE MODELS (LLMS)

We used large language models (LLMs) as an assistive tool to polish part of this paper. The roles of LLMs in this work were restricted to improving readability and presentation.

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

## A EXPERIMENTAL DETAILS AND ADDITIONAL RESULTS

### A.1 EXPERIMENTAL SETUP

We give details of our experimental setup in this section.

#### A.1.1 MODEL ARCHITECTURE

We use the improved Transformer architecture with SwiGLU (Shazeer, 2020), Rotary Positional Embeddings (Su et al., 2024), and RMSNorm (Zhang & Sennrich, 2019) following LLaMA (Touvron et al., 2023). More specifically, in each layer $l$, given the contextualized representations $\boldsymbol{X}^{(l)} \in \mathbb{R}^{L \times d_{\text{model}}}$ where $L$ is the sequence length and $d_{\text{model}}$ is the hidden dimension, then, $\boldsymbol{X}^{(l+1)}$ is obtained by

$$\boldsymbol{Y}^{(l)} = \text{MultiHead-Attention}(\text{RMSNorm}(\boldsymbol{X}^{(l)}))$$

$$\boldsymbol{X}^{(l+1)} = \text{SwiGLU}(\text{RMSNorm}(\boldsymbol{Y}^{(l)})),$$

where the $\text{SwiGLU}(\boldsymbol{X}) = \big(\text{Swish}(\boldsymbol{X}\,\boldsymbol{W}_1) \odot (\boldsymbol{X}\,\boldsymbol{W}_2)\big)\boldsymbol{W}_3$. Here, $\boldsymbol{W}_1 \in \mathbb{R}^{d_{\text{model}} \times \frac{8}{3}d_{\text{model}}}$, $\boldsymbol{W}_2 \in \mathbb{R}^{d_{\text{model}} \times \frac{8}{3}d_{\text{model}}}$, $\boldsymbol{W}_3 \in \mathbb{R}^{\frac{8}{3}d_{\text{model}} \times d_{\text{model}}}$ are learnable parameters.

The function MultiHead-Attention is instantiated with either the standard multi-head causal attention or the multi-head CASTLE, or the multi-head CASTLE-SWL.

#### A.1.2 MODEL CONFIGURATION AND TRAINING RECIPE

We train models on four scales from scratch: small (0.16B), medium (0.35B), large (0.75B) and XL (1.3B), where the medium, large and XL baseline models follow the configurations of GPT-3 (Brown et al., 2020). For the small setting, we increase the number of heads and the hidden dimension in the original GPT-3 configuration to better align parameter counts between standard causal attention and CASTLE. The configurations of the models can be found in Table 3. To ensure a fair comparison between CASTLE and standard causal attention, we align the number of model parameters by adjusting the number of attention heads and keeping hidden dimensions and head dimensions invariant. This avoids changes to the representational upper bound of the models' hidden states and the behavior of RoPE, both of which depend on the hidden dimension. As shown in Appendix C, the number of parameters in CASTLE matches that of standard causal attention when the number of heads satisfies the relation $n_{\text{CASTLE}} = \frac{4}{7}n_{\text{standard}}$, where $n_{\text{CASTLE}}$ and $n_{\text{standard}}$ denote the number of heads in CASTLE and standard causal attention, respectively. For the small model, the baseline uses 14 heads. By setting CASTLE to 8 heads, we align the parameter counts. For the other settings, the baseline models use 16 heads. As $16 * \frac{4}{7} \approx 9.14$, we choose 9 heads for CASTLE, resulting in a slightly smaller number of parameters than the baseline.

The model configurations and training recipes of CASTLE-SWL are identical to those of CASTLE.

The sliding window size for lookahead keys in CASTLE-SWL is set to 128 in CASTLE-SWL-S, and 512 in CASTLE-SWL-M, CASTLE-SWL-L, and CASTLE-SWL-XL. Ablation studies on sliding window sizes of CASTLE-SWL are provided in Appendix A.5.4.

We adopt the training hyper-parameters of Dao & Gu (2024). We use the AdamW optimizer (Loshchilov & Hutter, 2019) with $\beta_1 = 0.9$, $\beta_2 = 0.95$, weight decay rate coefficient 0.1, and gradient clipping coefficient 1.0. All models are trained with sequence length 2K and batch size 0.5M tokens. The small, medium, large and XL models use peak learning rates of $6.0 \times 10^{-4}$, $3.0 \times 10^{-4}$, $2.5 \times 10^{-4}$ and $2.0 \times 10^{-4}$, respectively. All models are trained with 2000 warmup steps, and then, the cosine scheduler decays the learning rate to 10% of the peak learning rate. All models are trained on the FineWeb-Edu dataset (Lozhkov et al.) for 50 billion tokens. The efficient parallel training algorithm of CASTLE and CASTLE-SWL (forward pass in Algorithm 1 and backward pass in Algorithm 3) is implemented in Triton (Tillet et al., 2019).

### A.2 ADDITIONAL DOWNSTREAM EVALUATION RESULTS

Evaluation results (0-shot) are provided in Table 4. 5-shot results have been shown in Table 2.

Table 3: Configurations and learning hyper-parameters (batch size in tokens and learning rate) of the models which we trained. CASTLE-SWL has exactly the same configurations and training hyper-parameters with CASTLE on each model scale, and is omitted from this table for clarity.

| Model Name | $n_{\text{params}}$ | $n_{\text{layers}}$ | $d_{\text{model}}$ | $n_{\text{heads}}$ | $d$ | Batch Size | Learning Rate |
|---|---|---|---|---|---|---|---|
| Baseline-S | 160M | 12 | 896 (=14 * 64) | 14 | 64 | 0.5M | $6.0 \times 10^{-4}$ |
| CASTLE-S | 160M | 12 | 896 | 8 | 64 | | |
| Baseline-M | 353M | 24 | 1024 (=16 * 64) | 16 | 64 | 0.5M | $3.0 \times 10^{-4}$ |
| CASTLE-M | 351M | 24 | 1024 | 9 | 64 | | |
| Baseline-L | 756M | 24 | 1536 (=16 * 96) | 16 | 96 | 0.5M | $2.5 \times 10^{-4}$ |
| CASTLE-L | 753M | 24 | 1536 | 9 | 96 | | |
| Baseline-XL | 1.310B | 24 | 2048 (=16 * 128) | 16 | 128 | 0.5M | $2.0 \times 10^{-4}$ |
| CASTLE-XL | 1.304B | 24 | 2048 | 9 | 128 | | |

Table 4: Evaluation results (0-shot) for downstream tasks of different model scales. Each model is pretrained on FineWeb-Edu for 50B tokens. All values denote accuracy in percentage (%). The higher accuracy values are shown in bold. Hella.=HellaSwag, Wino.=Winograde.

| Model Name | ARC-C | ARC-E | BoolQ | Hella. | MMLU | OBQA | PIQA | Wino. | Avg. |
|---|---|---|---|---|---|---|---|---|---|
| Baseline-S | **26.71** | 54.76 | 52.51 | 35.78 | 22.89 | 30.40 | 63.98 | **52.57** | 42.45 |
| CASTLE-S | 26.19 | **56.69** | **59.85** | **36.28** | 23.00 | 31.60 | 64.25 | 52.25 | **43.76** |
| CASTLE-SWL-S | 26.62 | 54.12 | 59.60 | 35.97 | 22.87 | **32.60** | **64.31** | 50.51 | 43.33 |
| Baseline-M | 28.58 | 60.90 | 53.61 | 43.01 | 23.21 | 33.40 | 67.95 | 50.91 | 45.20 |
| CASTLE-M | **30.20** | 61.36 | **58.01** | 43.24 | **25.34** | **34.60** | 67.95 | 52.64 | **46.67** |
| CASTLE-SWL-M | 29.52 | **61.41** | 57.03 | **43.76** | 23.29 | 33.00 | **68.12** | **53.83** | 46.24 |
| Baseline-L | 32.59 | 65.07 | 57.49 | 47.45 | 23.57 | 32.60 | 70.51 | 50.75 | 47.50 |
| CASTLE-L | 32.34 | **65.15** | 57.65 | 47.87 | **24.51** | 35.60 | **70.78** | 53.51 | 48.43 |
| CASTLE-SWL-L | **32.76** | 63.51 | **60.95** | **48.50** | 23.72 | **36.00** | 70.02 | **54.85** | **48.79** |
| Baseline-XL | 33.79 | 66.62 | 61.04 | 51.40 | **26.72** | 36.20 | **72.58** | 54.06 | 50.30 |
| CASTLE-XL | 35.32 | 67.51 | **62.81** | **52.15** | 23.74 | 37.00 | 70.67 | **56.59** | 50.72 |
| CASTLE-SWL-XL | **36.43** | **69.07** | 60.24 | 51.99 | 24.47 | **37.40** | 71.27 | 55.09 | **50.74** |

## A.3 Needle In a Haystack

We evaluate the models' performance on the needle-in-a-haystack tasks without any finetuning or length extrapolation in this section. Table 5 reports results on the RULER S-NIAH tasks (Hsieh et al., 2024). CASTLE-SWL-XL demonstrates consistently stronger retrieval capabilities than Baseline-X. In particular, on the S-NIAH-1 task, CASTLE-SWL-XL achieves nonzero scores at the 4K sequence length.

## A.4 Length Extrapolation

We evaluate the models' compatibility with YaRN (Peng et al., 2023) for length extrapolation. Table 6 reports the performance of the models on the RULER S-NIAH tasks (Hsieh et al., 2024). Across all tasks, CASTLE-SWL-XL achieves substantially higher accuracy than Baseline-XL, indicating that CASTLE-SWL has markedly better compatibility with YaRN than the baseline MHA model.

Table 5: Accuracy on the RULER S-NIAH tasks across different sequence lengths. Significant higher accuracy is in bold.

| | S-NIAH-1 | | | S-NIAH-2 | | | S-NIAH-3 | | |
| --- | --- | --- | --- | --- | --- | --- | --- | --- | --- |
| | 1K | 2K | 4K | 1K | 2K | 4K | 1K | 2K | 4K |
| Baseline-XL | 97.0 | 26.6 | 0.0 | 97.2 | 54.2 | 0.0 | 85.8 | 73.2 | 0.0 |
| CASTLE-SWL-XL | **99.4** | **85.4** | **24.6** | **99.2** | **86.0** | 2.2 | **89.4** | **84.2** | 0.2 |

Table 6: Accuracy on the RULER S-NIAH tasks under YaRN-based length extrapolation. Significant higher accuracy is in bold.

| | S-NIAH-1 | | S-NIAH-2 | | S-NIAH-3 | |
| --- | --- | --- | --- | --- | --- | --- |
| | 4K | 8K | 4K | 8K | 4K | 8K |
| Baseline-XL | 11.0 | 0.0 | 47.8 | 1.8 | 8.6 | 0.0 |
| CASTLE-SWL-XL | **63.4** | **39.0** | **68.4** | **35.0** | **30.2** | 1.4 |

## A.5 ABLATION STUDIES

We conduct ablation studies to better understand the contributions of different design components in CASTLE and CASTLE-SWL. These studies address three key questions:

- Are causal keys necessary, or could lookahead keys alone suffice?
- Do the observed improvements arise from the mechanism of lookahead keys, or simply from increasing the number of keys?
- What is the role of the SiLU function in (6)?
- How do sliding window sizes of lookahead keys in CASTLE-SWL affect the performance?
- How do we design the mask $M_t^U$ in the definition of CASTLE-SWL?
- Do CASTLE and CASTLE-SWL consistently outperform the baseline models across different token budgets?

We systematically investigate each question in the following sections.

### A.5.1 ABLATIONS ON CAUSAL KEYS

As described in Section 2.2, CASTLE adopts a hybrid design that partitions keys into two groups: causal keys and lookahead keys. If all lookahead keys are replaced with causal keys, CASTLE reduces to standard causal attention. Thus, the performance gains demonstrated in Section 3 can be attributed to the introduction of lookahead keys. A natural follow-up question is whether causal keys are necessary, or if lookahead keys alone suffice.

To investigate this, we construct a variant of CASTLE in which all causal keys are removed. To ensure a fair comparison, we adjust the configurations so that the total parameter count remains the same, as shown in Table 7.

Table 7: Configurations of CASTLE and its variant without causal keys used in ablation study on causal keys.

| CASTLE TYPE | $n_{\text{params}}$ | $n_{\text{layers}}$ | $d_{\text{model}}$ | $n_{\text{heads}}$ | $d$ |
| --- | --- | --- | --- | --- | --- |
| CASTLE | 120M | 12 | 768 | 6 | 64 |
| CASTLE w/o causal keys | 120M | 12 | 768 | 7 | 64 |

The above 2 models are both trained on FineWeb-Edu for 25B tokens for efficiency, using the same learning hyper-parameters with the small models in Section A.1.2. Their training and validation loss and perplexity are presented in Table 8.

Table 8: Training and validation loss and perplexity of CASTLE and its variant without causal keys. Each model is trained on FineWeb-Edu for 25B tokens.

|  | Train | | Eval | |
| --- | --- | --- | --- | --- |
|  | Loss | PPL | Loss | PPL |
| CASTLE | **2.913** | **18.417** | **2.920** | **18.541** |
| CASTLE w/o causal keys | 3.006 | 20.213 | 3.021 | 20.505 |

As shown in Table 8, removing causal keys results in a clear degradation in performance. This demonstrates that causal keys are indispensable in CASTLE.

While these results establish the necessity of both causal and lookahead keys, our current formulation in (6) employs a one-to-one pairing of a causal key and a lookahead key. An alternative design could involve grouping multiple causal keys with a single lookahead key, or vice versa. Exploring the optimal ratio between causal keys and lookahead keys is left for future work.

### A.5.2 Ablations on the Number of Keys

As discussed in Section C, when CASTLE uses half as many heads as standard causal attention, its parameter count becomes $\frac{7}{8}$ of the baseline. To maintain comparable parameter counts, we adjust the number of heads accordingly. However, unlike the baseline where each head corresponds to one key, each head in CASTLE introduces two keys—one causal key and one lookahead key.

This design results in CASTLE models having 16, 18, 18, and 18 keys for the small, medium, large, and XL scales, respectively, compared to the corresponding baselines with 14, 16, 16, and 16 keys (Table 3). Thus, CASTLE naturally uses slightly more keys than its baseline counterparts. A natural question arises: are the observed improvements due to the introduction of lookahead keys, or simply from having more keys overall?

To disentangle this effect, we construct CASTLE variants with only half as many heads as their baselines, ensuring that the total number of keys ($n_{\text{CausalKeys}} + n_{\text{LookaheadKeys}}$) matches the baselines. This adjustment results in CASTLE having a notably smaller parameter count than the baselines.

We also did the same ablation studies for CASTLE-SWL.

For efficiency, we train the medium and XL variants on FineWeb-Edu for 25B tokens, using the same hyper-parameters as in Section A.1.2. Results are reported in Table 10 (CASTLE) and Table 11 (CASTLE-SWL).

As shown in Table 10, despite having clearly fewer parameters, both CASTLE-M-16 and CASTLE-XL-16 outperform their baselines: CASTLE-M-16 lags behind CASTLE-M by only 0.005 in validation loss, yet surpasses the baseline by 0.026; CASTLE-XL-16 trails CASTLE-XL by 0.008 in validation loss, while exceeding the baseline by 0.032. Similar performance of CASTLE-SWL can be observed in Table 11.

These findings confirm that CASTLE and CASTLE-SWL's advantage stems from its mechanism of incorporating lookahead keys, rather than from increasing the number of keys from 16 to 18. This further consolidates the advantage of CASTLE and CASTLE-SWL.

### A.5.3 Ablations on SiLU function in (6)

In this section, we introduce the design of (6). We begin with the most direct formulation, which combines the causal-key and lookahead-key attention scores as

$$p_t = \text{softmax}\left(s_t^C - s_t^U\right). \tag{14}$$

Since $s_t^U$ may take positive or negative values and, by its definition in (5) together with the Gaussian initialization used when training from scratch, the distributions of $s_t^U$ and $-s_t^U$ are symmetric, the variants $p_t = \text{softmax}\left(s_t^C - s_t^U\right)$ and $p_t = \text{softmax}\left(s_t^C + s_t^U\right)$ should have equivalent performance. For consistency with the strategies discussed later in this section, we write it as $p_t = \text{softmax}\left(s_t^C - s_t^U\right)$ here.

Table 9: Configurations of baseline models, CASTLE, and its variants used in the ablation study on the number of keys. Baseline-M, Baseline-XL, CASTLE-M, and CASTLE-XL follow the same configurations as in Table 3. CASTLE-M-16 and CASTLE-XL-16 are constructed by reducing the number of heads in CASTLE-M and CASTLE-XL, respectively, so that the total number of keys ($n_{\text{LookaheadKeys}} + n_{\text{CausalKeys}}$) matches the number of keys of the corresponding baseline models. CASTLE-SWL-M, CASTLE-SWL-XL, CASTLE-SWL-M-16, CASTLE-SWL-XL-16 have identical configurations with CASTLE-M, CASTLE-XL, CASTLE-M-16, CASTLE-XL-16, respectively and are omitted from this table for clarity.

| Model Name | $n_{\text{params}}$ | $n_{\text{layers}}$ | $d_{\text{model}}$ | $n_{\text{heads}}$ | $n_{\text{LookaheadKeys}} + n_{\text{CausalKeys}}$ | $d$ |
|---|---|---|---|---|---|---|
| Baseline-M | 353M | 24 | 1024 (=16 * 64) | 16 | 16 | 64 |
| CASTLE-M | 351M | 24 | 1024 | 9 | 18 | 64 |
| CASTLE-M-16 | 340M | 24 | 1024 | 8 | 16 | 64 |
| Baseline-XL | 1.310B | 24 | 2048 (=16 * 128) | 16 | 16 | 128 |
| CASTLE-XL | 1.304B | 24 | 2048 | 9 | 18 | 128 |
| CASTLE-XL-16 | 1.260B | 24 | 2048 | 8 | 16 | 128 |

Table 10: Training and validation loss and perplexity of baseline models, CASTLE and CASTLE variants with the same number of keys as the baselines, after training for 25B tokens on FineWeb-Edu. The lowest loss and perplexity are shown in bold, and the second-lowest values are underlined.

| | | Train | | Eval | |
|---|---|---|---|---|---|
| | $n_{\text{params}}$ | Loss | PPL | Loss | PPL |
| Baseline-M | 353M | 2.740 | 15.483 | 2.742 | 15.523 |
| CASTLE-M | 351M | **2.709** | **15.018** | **2.711** | **15.039** |
| CASTLE-M-16 | 340M | 2.714 | 15.093 | 2.716 | 15.126 |
| Baseline-XL | 1.310B | 2.548 | 12.779 | 2.543 | 12.723 |
| CASTLE-XL | 1.304B | **2.507** | **12.267** | **2.503** | **12.219** |
| CASTLE-XL-16 | 1.260B | 2.514 | 12.349 | 2.511 | 12.316 |

Table 11: Training and validation loss and perplexity of baseline models, CASTLE-SWL and CASTLE-SWL variants with the same number of keys as the baselines, after training for 25B tokens on FineWeb-Edu. The lowest loss and perplexity are shown in bold, and the second-lowest values are underlined.

| | | Train | | Eval | |
|---|---|---|---|---|---|
| | $n_{\text{params}}$ | Loss | PPL | Loss | PPL |
| Baseline-M | 353M | 2.740 | 15.483 | 2.742 | 15.523 |
| CASTLE-SWL-M | 351M | **2.710** | **15.036** | **2.713** | **15.068** |
| CASTLE-SWL-M-16 | 340M | 2.716 | 15.117 | 2.718 | 15.150 |
| Baseline-XL | 1.310B | 2.548 | 12.779 | 2.543 | 12.723 |
| CASTLE-SWL-XL | 1.304B | **2.506** | **12.255** | **2.503** | **12.217** |
| CASTLE-SWL-XL-16 | 1.260B | 2.513 | 12.339 | 2.508 | 12.276 |

When training CASTLE-XL and CASTLE-SWL-XL without SiLU, i.e., replace (6) by (14), we consistently observed the loss blows up and shows `NaN` then. In particular, CASTLE-SWL-XL blows up around step 24300, whereas CASTLE-XL blows up earlier, around step 1500. Lowering the learning rate mitigates this instability. For example, CASTLE-SWL-XL without SiLU can remain stable up to at least 25B tokens when trained with peak learning rates of $1 \times 10^{-4}$ or $5 \times 10^{-5}$. However, such reductions in learning rate lead to noticeably worse performance, as shown in Table 12.

Table 12: Ablation study of the SiLU function in (6). Removing SiLU causes training instability. More specifically, training CASTLE-SWL-XL without SiLU in (6) under the same learning rate ($2 \times 10^{-4}$) as in Table 3 will have the loss curve blow up at around 24300 training step. Reducing the learning rate alleviates this instability issue but results in significant performance degradation.

| | | Train | | Eval | |
|---|---|---|---|---|---|
| | Learning Rate | Loss | PPL | Loss | PPL |
| CASTLE-SWL-XL | $2 \times 10^{-4}$ | 2.506 | 12.255 | 2.503 | 12.217 |
| CASTLE-SWL-XL w/o SiLU | $1 \times 10^{-4}$ | 2.523 | 12.468 | 2.520 | 12.424 |
| CASTLE-SWL-XL w/o SiLU | $5 \times 10^{-5}$ | 2.571 | 13.084 | 2.571 | 13.075 |

To address the instability observed with (14), we explored the following design variants:

- **Attempt I**: scale down the lookahead-key attention scores $s_t^U$, i.e., with a constant $C > 1$,

$$p_t = \mathrm{softmax}\left(s_t^C - s_t^U/C\right). \tag{15}$$

  We remark that $s_t^U$ can be either positive or negative, and by the definition of $s_t^U$ in (5) and the fact that we initialized model weights from Gaussian distribution when we train the models from scratch, defining $p_t = \mathrm{softmax}\left(s_t^C - s_t^U/C\right)$ and $p_t = \mathrm{softmax}\left(s_t^C + s_t^U/C\right)$ should have equivalent performance.

- **Attempt II**: Clipping $s_t^U$ on both sides. For differentiability, we use $\tanh$ instead of clipping directly, i.e.,

$$p_t = \mathrm{softmax}\left(s_t^C - \tanh\left(s_t^U\right)\right). \tag{16}$$

  By the same reason with Attempt I, defining $p_t = \mathrm{softmax}\left(s_t^C - \tanh\left(s_t^U\right)\right)$ and $p_t = \mathrm{softmax}\left(s_t^C + \tanh\left(s_t^U\right)\right)$ should have equivalent performance.

- **Attempt III**: Clipping $s_t^U$ on one side. For differentiability, we use SiLU instead of one-side clipping directly. We have tried the following 2 variants:

  1. $p_t = \mathrm{softmax}\left(s_t^C - \mathrm{SiLU}\left(s_t^U\right)\right)$
  2. $p_t = \mathrm{softmax}\left(s_t^C + \mathrm{SiLU}\left(s_t^U\right)\right)$.

  We remark that similarly to discussions in Attempt I and II above, $p_t = \mathrm{softmax}\left(s_t^C - \mathrm{SiLU}\left(s_t^U\right)\right)$ and $p_t = \mathrm{softmax}\left(s_t^C - \mathrm{SiLU}\left(-s_t^U\right)\right)$ should be equivalent design. However, it is clear that $p_t = \mathrm{softmax}\left(s_t^C - \mathrm{SiLU}\left(s_t^U\right)\right)$ and $p_t = \mathrm{softmax}\left(s_t^C + \mathrm{SiLU}\left(s_t^U\right)\right)$ are not equivalent because $\mathrm{SiLU}(x)$ is not symmetric and $\mathrm{SiLU}(-x) \neq -\mathrm{SiLU}(x)$

Among the design choices mentioned above, $p_t = \mathrm{softmax}\left(s_t^C + \mathrm{SiLU}\left(s_t^U\right)\right)$ shows even earlier blowing up than (14). More specifically, on the XL size model, its loss curve starts to increase at the 1100-th step. The other choices show stable training. We attach our test results of Attempt I and Attempt II below. Attempt III is what we did in CASTLE(-SWL).

**Attempt I to solve instability of** (14)

Our Attempt I primarily targets the instability that arises when training lookahead keys with a sliding window while using (14) in place of (6). To mitigate this issue, we scale $s_t^U$ by a constant factor $C$, motivated by the intuition that excessively large $s_t^U$ values may induce training instability under (14). By definition, the magnitude of $s_t^U$ is tied to the lookahead-key window size, suggesting that $C$ should naturally be chosen in relation to this window size. For a fair comparison, we adopt the same window size for both Attempt II and CASTLE-SWL-XL. In our experiments for Attempt II, we set $C = \sqrt{W}$, where $W$ is the window size of lookahead keys.

The performance of Attempt I (which incorporates the modification in (15)) is reported in Tables 13 and 14. In terms of training and validation loss, Attempt I improves slightly over the baseline but remains significantly worse than CASTLE-SWL-XL. Downstream evaluation results follow the same pattern: Attempt I outperforms the baseline on average, yet consistently underperforms relative

to CASTLE-SWL-XL. The fact that Attempt I surpasses the baseline demonstrates the effectiveness of the lookahead mechanism. However, its performance deficit compared to CASTLE-SWL-XL may stem from the scaling factor which may constrain the model's expressivity.

We note that in Attempt I both the lookahead-key window size and the scaling factor $C$ can influence model performance. A more systematic study, including making $C$ a learnable parameter, is left for future work.

Table 13: Training and validation loss and perplexity comparison between Attempt I to solve instability of (14) (using (15)) and the baseline model. Each model is trained on FineWeb-Edu for 50B tokens.

|  | | Train | | Eval | |
| --- | --- | --- | --- | --- | --- |
|  | $n_{\text{params}}$ | Loss | PPL | Loss | PPL |
| Baseline-XL | 1.310B | 2.430 | 11.360 | 2.426 | 11.309 |
| CASTLE-SWL-XL | 1.304B | **2.401** | **11.036** | **2.389** | **10.900** |
| Attempt I (15) | 1.304B | 2.424 | 11.286 | 2.417 | 11.215 |

Table 14: Downstream evaluation (5-shot) comparison between Attempt I to solve instability of (14) (using (15)) and the baseline model. Each model is trained on FineWeb-Edu for 50B tokens.

| Model Name | ARC-C | ARC-E | BoolQ | Hella. | MMLU | OBQA | PIQA | Wino. | Avg. |
| --- | --- | --- | --- | --- | --- | --- | --- | --- | --- |
| Baseline-XL | 35.58 | 65.78 | 61.07 | 50.84 | **26.71** | 36.20 | 71.27 | 52.72 | 50.02 |
| CASTLE-SWL-XL | **38.99** | **70.08** | **61.74** | **52.35** | 25.85 | **37.20** | **72.52** | **56.75** | **51.93** |
| Attempt I (15) | 37.29 | 68.64 | 61.71 | 50.26 | 25.44 | 35.20 | 69.97 | 56.20 | 50.59 |

**Attempt II to solve instability of** (14)

Table 15 reports the training and validation losses and perplexities for Attempt II (which incorporates the modification in (16)) compared with the baseline model. As shown, Attempt II achieves slightly lower training and validation losses than the baseline, but remains noticeably worse than CASTLE-M. Downstream evaluation results in Table 16 also exhibit a similar trend, i.e., Attempt II outperforms Baseline-M on average, yet still lags behind CASTLE-M across most tasks.

Following a similar trend with Attempt I, the fact that Attempt II outperforms the baseline demonstrates the effectiveness of the lookahead mechanism again. However, its lower performance compared to CASTLE-M may come from the clipping on both sides (using `tanh`) which overly constrains the model's expressivity.

Table 15: Training and validation loss and perplexity comparison between Attempt II to solve instability of (14) (using (16)) and the baseline model. Each model is trained on FineWeb-Edu for 50B tokens.

|  | | Train | | Eval | |
| --- | --- | --- | --- | --- | --- |
|  | $n_{\text{params}}$ | Loss | PPL | Loss | PPL |
| Baseline-M | 353M | 2.641 | 14.030 | 2.639 | 14.004 |
| CASTLE-M | 351M | **2.616** | **13.684** | **2.615** | **13.665** |
| Attempt II (16) | 351M | 2.636 | 13.961 | 2.634 | 13.926 |

Table 16: Downstream evaluation (5-shot) comparison between Attempt II to solve instability of (14) (using (16)) and the baseline model. Each model is trained on FineWeb-Edu for 50B tokens.

| Model Name | ARC-C | ARC-E | BoolQ | Hella. | MMLU | OBQA | PIQA | Wino. | Avg. |
|---|---|---|---|---|---|---|---|---|---|
| Baseline-M | 31.06 | 62.46 | 48.47 | 42.83 | **25.22** | 33.00 | 68.39 | 51.78 | 45.40 |
| CASTLE-M | **32.17** | **64.06** | 54.62 | **43.47** | **25.22** | **33.80** | **69.48** | **52.49** | **46.91** |
| Attempt II (16) | 31.83 | 63.47 | **59.94** | 42.31 | 24.53 | 32.00 | 68.55 | 50.91 | 46.69 |

### A.5.4 ABLATIONS ON SLIDING WINDOW SIZE IN LOOKAHEAD KEYS OF CASTLE-SWL

We investigate the effect of sliding window size on CASTLE-SWL across different model scales. For efficiency, each model is trained on the FineWeb-Edu dataset for 25B tokens. We train CASTLE-SWL with window sizes of 64, 128, 256, 512, and 1024, and report both training and validation loss and perplexity. The results for small, medium, large, and XL models are shown in Table 17, Table 18, Table 19, and Table 20, respectively. Throughout, we use the number following "SWL" to denote the sliding window size, e.g., CASTLE-SWL128-S refers to CASTLE-SWL-S with a window size of 128.

As shown in Table 17, a window size of 128 achieves the best performance for the small model. For medium, large, and XL models (Tables 18, 19, and 20), the optimal performance is obtained with a window size of 512. Based on these findings, we adopt window sizes of 128, 512, 512, and 512 for small, medium, large, and XL models, respectively, in the experiments presented in Section 3.

Overall, CASTLE-SWL performance is not sensitive to the choice of sliding window size in the range [128, 1024]. For example, for the medium model, the best sliding window size (512) reduces validation loss by 0.0297 compared to the baseline, but the gap between the best (512) and worst window sizes (256) is only 0.0038. Similarly, for the XL model, the best sliding window (512) improves upon the baseline by 0.0406 while the difference between the best (512) and worst (128) sliding window sizes is just 0.0099. These results suggest that while CASTLE-SWL consistently improves over the baselines across all model scales and sliding window sizes tested, and its performance is relatively robust to the choices of window sizes.

However, for the medium, large, and XL models, using a window size of 64 results in substantial performance degradation. In both training and validation metrics, CASTLE-SWL64 performs at least 0.01 worse than its counterpart CASTLE-SWL512. Moreover, in downstream evaluation, CASTLE-SWL64 shows a clear disadvantage not only compared to CASTLE-SWL512, but even relative to the baseline model. The downstream evaluation results are presented in Table 21.

Table 17: Ablations on sliding window sizes for CASTLE-SWL-S. Training and validation loss and perplexity of baseline models, CASTLE-SWL-S with different sliding window sizes after training for 25B tokens on FineWeb-Edu are reported. The lowest loss and perplexity are shown in bold, and the second-lowest values are underlined.

| | | Train | | Eval | |
|---|---|---|---|---|---|
| | $n_{params}$ | Loss | PPL | Loss | PPL |
| Baseline-S | 160M | 2.892 | 18.037 | 2.901 | 18.197 |
| CASTLE-SWL64-S | 160M | 2.884 | 17.888 | 2.890 | 18.024 |
| CASTLE-SWL128-S | 160M | 2.883 | 17.859 | 2.889 | 17.971 |
| CASTLE-SWL256-S | 160M | 2.888 | 17.952 | 2.895 | 18.092 |
| CASTLE-SWL512-S | 160M | 2.885 | 17.910 | 2.892 | 18.023 |
| CASTLE-SWL1024-S | 160M | 2.885 | 17.901 | 2.892 | 18.031 |

Table 18: Ablations on sliding window sizes for CASTLE-SWL-M. Training and validation loss and perplexity of baseline models, CASTLE-SWL-M with different sliding window sizes after training for 25B tokens on FineWeb-Edu are reported. The lowest loss and perplexity are shown in bold, and the second-lowest values are underlined.

| | | Train | | Eval | |
|---|---|---|---|---|---|
| | $n_{params}$ | Loss | PPL | Loss | PPL |
| Baseline-M | 353M | 2.740 | 15.483 | 2.742 | 15.523 |
| CASTLE-SWL64-M | 351M | 2.726 | 15.273 | 2.728 | 15.306 |
| CASTLE-SWL128-M | 351M | 2.715 | 15.098 | 2.716 | 15.124 |
| CASTLE-SWL256-M | 351M | 2.715 | 15.112 | 2.716 | 15.127 |
| CASTLE-SWL512-M | 351M | **2.710** | **15.036** | **2.713** | **15.068** |
| CASTLE-SWL1024-M | 351M | 2.713 | 15.071 | 2.715 | 15.103 |

Table 19: Ablations on sliding window sizes for CASTLE-SWL-L. Training and validation loss and perplexity of baseline models, CASTLE-SWL-L with different sliding window sizes after training for 25B tokens on FineWeb-Edu are reported. The lowest loss and perplexity are shown in bold, and the second-lowest values are underlined.

| | | Train | | Eval | |
|---|---|---|---|---|---|
| | $n_{params}$ | Loss | PPL | Loss | PPL |
| Baseline-L | 756M | 2.740 | 15.483 | 2.742 | 15.523 |
| CASTLE-SWL64-L | 753M | 2.608 | 13.572 | 2.607 | 13.552 |
| CASTLE-SWL128-L | 753M | 2.597 | 13.425 | 2.596 | 13.411 |
| CASTLE-SWL256-L | 753M | 2.589 | 13.314 | 2.587 | 13.290 |
| CASTLE-SWL512-L | 753M | **2.582** | **13.219** | **2.580** | **13.202** |
| CASTLE-SWL1024-L | 753M | 2.582 | 13.229 | 2.581 | 13.209 |

Table 20: Ablations on sliding window sizes for CASTLE-SWL-XL. Training and validation loss and perplexity of baseline models, CASTLE-SWL-XL with different sliding window sizes after training for 25B tokens on FineWeb-Edu are reported. The lowest loss and perplexity are shown in bold, and the second-lowest values are underlined.

| | | Train | | Eval | |
|---|---|---|---|---|---|
| | $n_{params}$ | Loss | PPL | Loss | PPL |
| Baseline-XL | 1.310B | 2.548 | 12.779 | 2.543 | 12.723 |
| CASTLE-SWL64-XL | 1.304B | 2.530 | 12.548 | 2.525 | 12.491 |
| CASTLE-SWL128-XL | 1.304B | 2.517 | 12.393 | 2.513 | 12.339 |
| CASTLE-SWL256-XL | 1.304B | 2.514 | 12.353 | 2.510 | 12.300 |
| CASTLE-SWL512-XL | 1.304B | **2.506** | **12.255** | **2.503** | **12.217** |
| CASTLE-SWL1024-XL | 1.304B | 2.514 | 12.351 | 2.509 | 12.294 |

Table 21: Downstream evaluation (5-shot) comparison among CASTLW-SWL64-XL, CASTLE-SWL512-XL and Baseline-XL. Each model is trained on FineWeb-Edu for 25B tokens.

| Model Name | ARC-C | ARC-E | BoolQ | Hella. | MMLU | OBQA | PIQA | Wino. | Avg. |
|---|---|---|---|---|---|---|---|---|---|
| Baseline-XL | 33.96 | 64.06 | **61.99** | 46.92 | **26.95** | 33.40 | 69.10 | 52.88 | 48.66 |
| CASTLE-SWL512-XL | **34.81** | **69.32** | 55.35 | **49.50** | 26.04 | **36.60** | **71.00** | **53.99** | **49.58** |
| CASTLE-SWL64-XL | 33.11 | 64.98 | 57.95 | 44.51 | 24.70 | 32.60 | 69.04 | 52.09 | 47.37 |

### A.5.5 ABLATIONS ON BASELINE MODELS WITH SLIDING WINDOW

In this section, we provide evidence that the performance gains of CASTLE-SWL over baseline models stem from the core mechanism of lookahead keys, rather than from the mere use of a sliding window.

To isolate the effect of sliding windows, we augment Baseline-S and Baseline-L with sliding-window sizes of 128 and 512, respectively, matching the window sizes used for lookahead keys in CASTLE-SWL-S and CASTLE-SWL-L. We refer to these modified models as Baseline-SW-S and Baseline-SW-L.

All models are trained on 50B tokens. Their training and validation loss and perplexity are reported in Table 22, and downstream evaluation results are presented in Table 23.

We observe that both Baseline-SW-S and Baseline-SW-L experience substantial degradation in training and validation loss. Their downstream task performance is either comparable to or worse than the original baselines.

These results indicate that applying a sliding window to lookahead keys is fundamentally different from applying it to the causal keys in standard causal attention. And the advantages of CASTLE-SWL cannot be attributed simply to the use of a sliding-window mechanism.

Table 22: Training and validation loss and perplexity for models with CASTLE, standard causal attention and standard causal attention with sliding windows. Each model is trained on FineWeb-Edu for 50B tokens. The best and second best results are highlighted in bold and underline, respectively.

| | | Train | | Eval | |
|---|---|---|---|---|---|
| | $n_{\text{params}}$ | Loss | PPL | Loss | PPL |
| Baseline-S | 160M | 2.795 | 16.364 | 2.798 | 16.411 |
| Baseline-SW-S | 160M | 2.861 | 17.487 | 2.868 | 17.608 |
| CASTLE-SWL-S | 160M | **2.786** | **16.213** | **2.790** | **16.273** |
| Baseline-L | 756M | 2.513 | 12.346 | 2.507 | 12.269 |
| Baseline-SW-L | 756M | 2.526 | 12.503 | 2.524 | 12.480 |
| CASTLE-SWL-L | 753M | **2.476** | **11.890** | **2.471** | **11.832** |

Table 23: Evaluation results (5-shot) for downstream tasks of models with CASTLE, standard causal attention and standard causal attention with sliding windows. The higher accuracy values are shown in bold. All values denote accuracy in percentage (%). Each model is pretrained on FineWeb-Edu for 50B tokens. Hella.=HellaSwag, Wino.=Winograde.

| Model Name | ARC-C | ARC-E | BoolQ | Hella. | MMLU | OBQA | PIQA | Wino. | Avg. |
|---|---|---|---|---|---|---|---|---|---|
| Baseline-S | 25.68 | 54.97 | **56.09** | 33.81 | 25.54 | 28.20 | 63.98 | 52.57 | 42.60 |
| Baseline-SW-S | 26.28 | 53.91 | 54.62 | **35.51** | **25.83** | 29.00 | 62.73 | **53.20** | 42.63 |
| CASTLE-SWL-S | **27.39** | **56.19** | **56.09** | 35.46 | 25.34 | **30.20** | **64.85** | 51.22 | **43.34** |
| Baseline-L | 33.36 | 63.64 | 59.24 | 46.16 | **26.82** | 33.40 | **69.53** | 54.06 | 48.28 |
| Baseline-SW-L | 32.94 | 65.36 | 59.24 | 46.05 | 24.86 | 32.40 | 69.31 | **54.14** | 48.04 |
| CASTLE-SWL-L | **36.26** | **65.53** | **60.58** | **48.55** | 24.70 | **34.80** | 69.10 | 53.67 | **49.15** |

### A.5.6 ABLATIONS ON MASK $M_t^U$ IN THE DEFINITION OF CASTLE

In this section, we describe why we define $\boldsymbol{M}_t^U$ as it is in (3). We have tested all the following 3 cases of $\boldsymbol{M}_t^U \in \mathbb{R}^{t \times t}$:

- Option I: removing $\boldsymbol{M}_t^U$ in (2), which is equivalent to defining

$$[\boldsymbol{M}_t^U]_{ij} = 1 \quad \text{for any } i, j \in [t]. \tag{17}$$

When generating the $(t+1)$-th token, Option I allows lookahead keys of each token $s$ to have access to information from representations of all tokens in $\{1, \cdots, t\}$.

- Option II: define $\boldsymbol{M}_t^U$ as the causal mask, i.e.,

$$[\boldsymbol{M}_t^U]_{ij} = 0 \text{ if } i \geq j \quad \text{and} \quad [\boldsymbol{M}_t^U]_{ij} = -\infty \text{ otherwise.} \tag{18}$$

In Option II, lookahead keys no longer look ahead, i.e., when generating the $(t+1)$-th token, Option I restricts lookahead keys of each token $s$ to have access to information only from tokens in $\{1, \cdots, s\}$.

- Option III: define $M_t^U$ as in (3). This is what we choose for CASTLE. More specifically, when generating the $(t+1)$-th token, Option I allows lookahead keys of each token $s$ to have access to information from representations of its subsequent tokens in $\{s+1, \cdots, t\}$

It is shown that both Option I and Option II lead to worse training and validation losses compared with the baseline. Although their downstream evaluation performance is comparable to the baseline, it remains significantly below that of CASTLE.

A likely explanation for the poor performance of Option I is that the model learns features from preceding and subsequent tokens in different ways. Allowing lookahead keys to attend to both directions simultaneously may introduce conflicts and then degrade learning.

The poor performance of Option II highlights the importance of the lookahead mechanism itself, i.e., the improvement of lookahead keys does not come from the structure in (6) in which attention scores from different heads are combined together. Instead, the superiority of lookahead keys comes from the "lookahead" mechanism itself, i.e., allowing keys to have access to information after it (while remaining its autoregressive property).

Table 24: Training and validation loss and perplexity comparison among Option I (17) and Option II (18) of $M_t^U$, CASTLE and the baseline model. Each model is trained on FineWeb-Edu for 25B tokens.

|  |  | Train | | Eval | |
| --- | --- | --- | --- | --- | --- |
|  | $n_{\text{params}}$ | Loss | PPL | Loss | PPL |
| Baseline-M | 353M | 2.740 | 15.483 | 2.742 | 15.523 |
| CASTLE-M | 351M | **2.709** | **15.011** | **2.711** | **15.039** |
| Option I (17) | 351M | 2.744 | 15.545 | 2.746 | 15.573 |
| Option II (18) | 351M | 2.758 | 15.764 | 2.761 | 15.822 |

Table 25: Downstream evaluation (0-shot) comparison among Option I (17) and Option II (18) of $M_t^U$, CASTLE and the baseline model. Each model is trained on FineWeb-Edu for 25B tokens.

| Model Name | ARC-C | ARC-E | BoolQ | Hella. | MMLU | OBQA | PIQA | Wino. | Avg. |
| --- | --- | --- | --- | --- | --- | --- | --- | --- | --- |
| Baseline-M | 27.90 | 59.09 | 58.13 | 40.08 | 23.64 | 31.80 | 66.05 | 51.07 | 44.72 |
| CASTLE-M | 28.33 | 59.13 | **59.05** | 40.79 | 24.25 | 31.60 | **66.54** | 50.51 | **45.02** |
| Option I (17) | 28.16 | 57.83 | 54.34 | 39.72 | 23.53 | **34.80** | 66.16 | **53.04** | 44.70 |
| Option II (18) | **28.58** | **59.89** | 58.26 | 39.90 | **24.77** | 29.40 | 66.21 | 51.62 | 44.83 |

### A.5.7 EXPERIMENTS ON DIFFERENT TOKEN-BUDGETS

We trained Baseline-XL, CASTLE-Xl and CASTLE-SWL on 5B, 10B, 20B, 30B and 50B tokens separately. The results are reported in Table 26 and Table 27 together with the scaling curves in Figure 5. It is shown that both CASTLE and CASTLE-SWL consistently outperform the Baseline-XL model across all training-token budgets from 5B to 50B tokens. The scaling trends appear approximately linear when plotted in log–log space. Overall, the token-scaling experiments validate that CASTLE and CASTLE-SWL achieves better scaling properties than the baseline, indicating superior sample efficiency.

Table 26: Training loss for XL-size models trained with different token budgets.

| Model | 5B | 10B | 20B | 30B | 50B |
| --- | --- | --- | --- | --- | --- |
| Baseline-XL | 2.798 | 2.650 | 2.544 | 2.485 | 2.430 |
| CASTLE-SWL-XL | **2.733** | **2.599** | 2.497 | **2.449** | **2.401** |
| CASTLE | 2.736 | 2.602 | **2.491** | 2.450 | **2.401** |

Table 27: Validation loss for XL-size models trained with different token budgets.

| Model | 5B | 10B | 20B | 30B | 50B |
|---|---|---|---|---|---|
| Baseline-XL | 2.804 | 2.650 | 2.540 | 2.480 | 2.426 |
| CASTLE-SWL-XL | **2.737** | **2.597** | 2.497 | 2.444 | **2.389** |
| CASTLE | 2.739 | 2.602 | **2.491** | **2.443** | 2.391 |

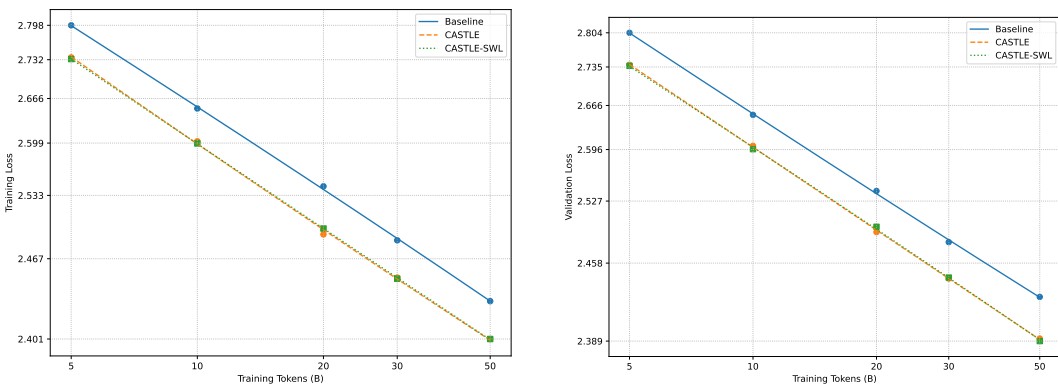

Figure 5: Scaling behavior of XL-size models as a function of training tokens. The $x$-axis is plotted in $\log$-scale over the token budget, while the $y$-axis uses the transformation $\log(\text{loss} - 2.1516)$. Both CASTLE and CASTLE-SWL exhibit consistently better scaling trends than the Baseline-XL model in training and validation loss.

## A.6 ADDITIONAL LOSS CURVES

This section presents additional training and validation loss curves for the small, medium, and large models, while the loss curves for the XL models are already shown in Figure 4. Each figure compares the baseline with CASTLE and CASTLE-SWL.

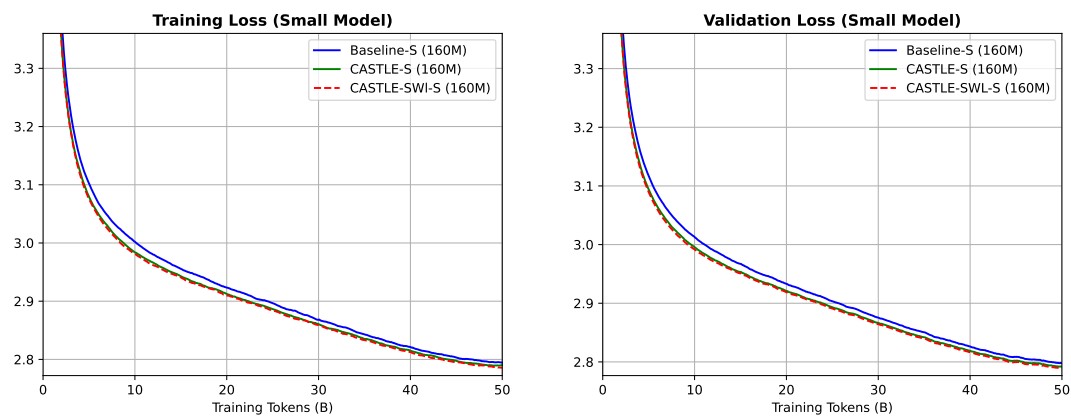

Figure 6: Training and validation loss curves of small models. Training loss curve is smoothened with a moving window of 2000 training steps. Validation loss is evaluated every 100 training steps on 40M tokens, and its curve is smoothened by a moving window of 20 evaluation intervals. As seen in Table 1 and in comparison with Figure 7, Figure 8 and Figure 4, CASTLE yields only marginal improvements over the baseline on small models. A likely explanation is that this is because the benefit of lookahead keys may lie in helping models capture global dependencies, but small models are capacity-limited and can primarily extract local features, making global relations less useful at this scale.

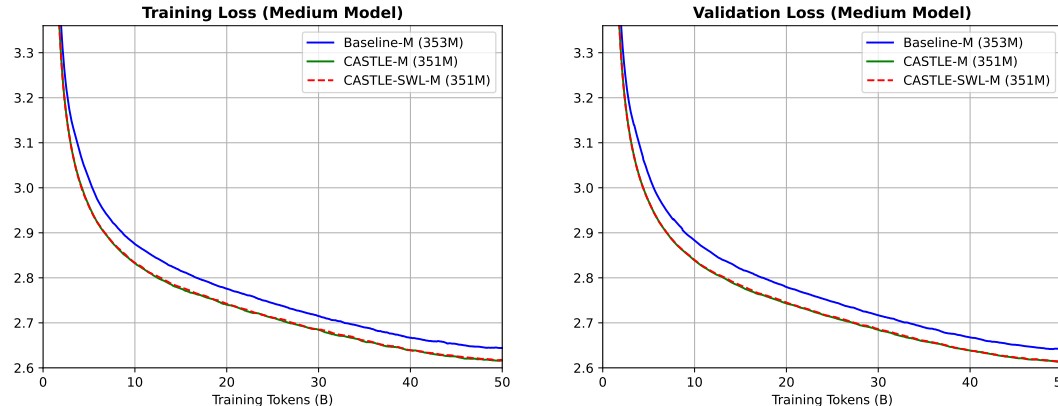

Figure 7: Training and validation loss curves of medium models. Training loss curve is smoothened with a moving window of 2000 training steps. Validation loss is evaluated every 100 training steps on 40M tokens, and its curve is smoothened by a moving window of 20 evaluation intervals. After 50B training tokens, CASTLE-M achieves a 0.0294 lower training loss and a 0.0245 lower validation loss compared to Baseline-M, while CASTLE-SWL-M achieves a 0.0232 lower training loss and a 0.0241 lower validation loss compared to Baseline-M

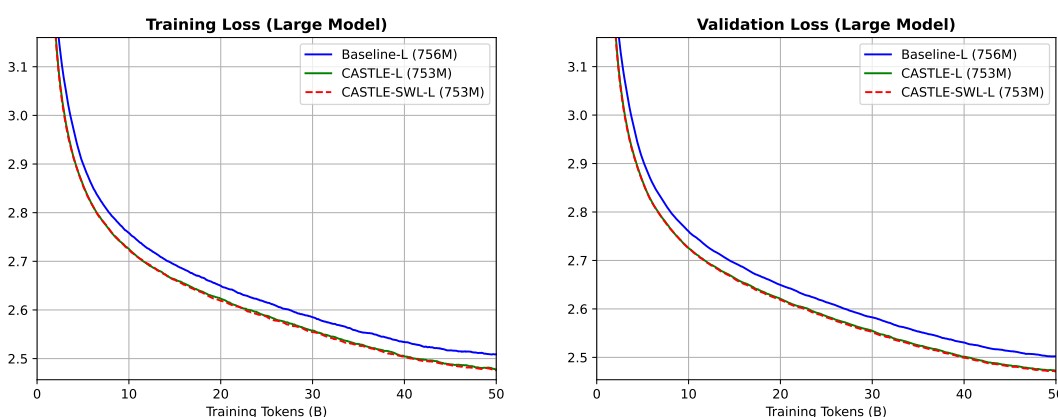

Figure 8: Training and validation loss curves of large models. Training loss curve is smoothened with a moving window of 2000 training steps. Validation loss is evaluated every 100 training steps on 40M tokens, and its curve is smoothened by a moving window of 20 evaluation intervals. After 50B training tokens, CASTLE-L achieves a 0.0371 lower training loss and a 0.0356 lower validation loss compared to Baseline-L, while CASTLE-SWL-L achieves a 0.0376 lower training loss and a 0.0366 lower validation loss compared to Baseline-L

## A.7 Efficiency

### A.7.1 Inference Efficiency

We implement our CASTLE and CASTLE-SWL inference kernels in Triton (Tillet et al., 2019) and measure the execution time of a single CASTLE(-SWL) block which as in Table 3 has 9 heads, head dimension 128, hidden dimension 2048. For comparison, we report the runtime of the FlashAttention-2 Triton implementation of standard causal attention block used in the Baseline-XL model, which uses 16 heads with the same head and hidden dimensions. We use batch size 256. The experiments are conducted on a single H100 GPU. The results are measured in milliseconds and summarized in Table 28. It is shown that CASTLE-SWL consistently runs faster than CASTLE-XL across all sequence lengths. CASTLE-SWL narrows the gap to the FlashAttention-2 Baseline-XL

implementation at longer sequence lengths. At a sequence length of 2K, CASTLE-SWL-XL is 22% slower than Baseline-XL, whereas at 16K it is only 12% slower.

We remark that our current inference kernels for CASTLE-SWL and CASTLE are not fully optimized. In particular, they exhibit lower memory–bandwidth utilization compared to the highly optimized FlashAttention-2 Triton kernels. This suggests that the inference performance of CASTLE(-SWL) still has substantial room for improvement.

Table 28: Inference efficiency (in milliseconds) comparison between FlashAttention-2 Triton implementation of standard causal attention and Triton implementation for inference kernel of CASTLE.

|      | Baseline-XL | CASTLE-SWL-XL | CASTLE-XL |
|------|-------------|---------------|-----------|
| 2K   | 1.4026      | 1.7986        | 2.4718    |
| 4K   | 2.7691      | 3.3795        | 4.9139    |
| 8K   | 5.5111      | 6.4824        | 9.8132    |
| 16K  | 11.0117     | 12.5904       | 19.6271   |

### A.7.2 TRAINING EFFICIENCY

We compare training throughput (tokens/s) of the CASTLE-SWL-XL, CASTLE-XL and Baseline-XL models. The experiments are conducted on a single H100 GPU. Results are provided in Table 29. CASTLE-SWL achieves notably higher throughput than CASTLE-XL. At longer sequences, the difference between CASTLE-SWL-XL and Baseline-XL narrows. At a sequence length of 16K, CASTLE-SWL-XL attains 21% lower throughput than Baseline-XL.

We remark that we made no attempts to optimize the training kernels. Developing more efficient kernel implementations is left for future work.

Table 29: Training throughput (tokens/s) comparison between FlashAttention-2 Triton implementation of standard causal attention and Triton implementation for inference kernel of CASTLE.

| SeqLen | BS | Baseline-XL | CASTLE-SWL-XL | CASTLE-XL |
|--------|----|-------------|---------------|-----------|
| 2K     | 8  | 39817       | 24651         | 17537     |
| 4K     | 4  | 32926       | 21339         | 10472     |
| 8K     | 2  | 24547       | 17338         | 5765      |
| 16K    | 1  | 16386       | 12933         | 3027      |

## B  PROOF OF THEOREM 1

First, recall the notations in Section 2.3. More specifically, consider inputs $\boldsymbol{X}^L = \begin{pmatrix} \boldsymbol{x}_1 \\ \boldsymbol{x}_2 \\ \vdots \\ \boldsymbol{x}_L \end{pmatrix} \in$

$\mathbb{R}^{L \times d_{\mathrm{model}}}$, where $\boldsymbol{x}_t$ is the representation of the $t$-th input token, $L$ is the sequence length and $d_{\mathrm{model}}$ is the hidden dimension.

For each $1 \le t \le L$, denote

$$\boldsymbol{q}_t^U = \boldsymbol{x}_t \boldsymbol{W}_Q^U, \quad \boldsymbol{k}_t^U = \boldsymbol{x}_t \boldsymbol{W}_K^U, \quad \boldsymbol{v}_t^U = \boldsymbol{x}_t \boldsymbol{W}_V^U,$$
$$\boldsymbol{q}_t^C = \boldsymbol{x}_t \boldsymbol{W}_Q^C, \quad \boldsymbol{k}_t^C = \boldsymbol{x}_t \boldsymbol{W}_K^C, \quad \boldsymbol{v}_t^C = \boldsymbol{x}_t \boldsymbol{W}_V^C.$$

$$
\boldsymbol{Q}_t^U = \begin{pmatrix} \boldsymbol{q}_1^U \\ \boldsymbol{q}_2^U \\ \vdots \\ \boldsymbol{q}_t^U \end{pmatrix} = \boldsymbol{X}^t \boldsymbol{W}_Q^U, \quad \boldsymbol{K}_t^U = \begin{pmatrix} \boldsymbol{k}_1^U \\ \boldsymbol{k}_2^U \\ \vdots \\ \boldsymbol{k}_t^U \end{pmatrix} = \boldsymbol{X}^t \boldsymbol{W}_K^U, \quad \boldsymbol{V}_t^U = \begin{pmatrix} \boldsymbol{v}_1^U \\ \boldsymbol{v}_2^U \\ \vdots \\ \boldsymbol{v}_t^U \end{pmatrix} = \boldsymbol{X}^t \boldsymbol{W}_V^U,
$$

$$
\boldsymbol{Q}_t^C = \begin{pmatrix} \boldsymbol{q}_1^C \\ \boldsymbol{q}_2^C \\ \vdots \\ \boldsymbol{q}_t^C \end{pmatrix} = \boldsymbol{X}^t \boldsymbol{W}_Q^C, \quad \boldsymbol{K}_t^C = \begin{pmatrix} \boldsymbol{k}_1^C \\ \boldsymbol{k}_2^C \\ \vdots \\ \boldsymbol{k}_t^C \end{pmatrix} = \boldsymbol{X}^t \boldsymbol{W}_K^C, \quad \boldsymbol{V}_t^C = \begin{pmatrix} \boldsymbol{v}_1^C \\ \boldsymbol{v}_2^C \\ \vdots \\ \boldsymbol{v}_t^C \end{pmatrix} = \boldsymbol{X}^t \boldsymbol{W}_V^C.
$$

And $\boldsymbol{M}_t^C$, $\widetilde{\boldsymbol{M}}_t^C$ and $\boldsymbol{M}_t^U$ are $t$-by-$t$ mask matrices. $\boldsymbol{M}_t^C$ is the $t$-by-$t$ causal mask which prevents tokens from attending to their future tokens, i.e., $[\boldsymbol{M}_t^C]_{ij} = 0$ if $i \geq j$ and $[\boldsymbol{M}_t^C]_{ij} = -\infty$ otherwise; $[\widetilde{\boldsymbol{M}}_t^C]_{ij} = 1$ if $i \geq j$ and $[\widetilde{\boldsymbol{M}}_t^C]_{ij} = 0$ otherwise. For CASTLE, $\boldsymbol{M}_t^U$ is defined in (3) and for CASTLE-SWL, $\boldsymbol{M}_t^U$ is defined in (8).

For the projection matrices of the entire sequence $\boldsymbol{X}^L$, we drop $L$ as in Theorem 1 for simplicity, i,e,

$$
\boldsymbol{Q}^U = \boldsymbol{Q}_L^U = \boldsymbol{X}^L \boldsymbol{W}_Q^U, \quad \boldsymbol{K}^U = \boldsymbol{K}_L^U = \boldsymbol{X}^L \boldsymbol{W}_K^U, \quad \boldsymbol{V}^U = \boldsymbol{V}_L^U = \boldsymbol{X}^L \boldsymbol{W}_V^U,
$$
$$
\boldsymbol{Q}^C = \boldsymbol{Q}_L^C = \boldsymbol{X}^L \boldsymbol{W}_Q^C, \quad \boldsymbol{K}^C = \boldsymbol{K}_L^C = \boldsymbol{X}^L \boldsymbol{W}_K^C, \quad \boldsymbol{V}^C = \boldsymbol{V}_L^C = \boldsymbol{X}^L \boldsymbol{W}_V^C.
$$

And $\boldsymbol{M}^U = \boldsymbol{M}_L^U$, $\boldsymbol{M}^C = \boldsymbol{M}_L^C$. Then, $\boldsymbol{M}_t^U$ is a $t$-by-$t$ submatrix of $\boldsymbol{M}^U = \boldsymbol{M}_L^U$. Similarly, $\boldsymbol{M}_t^C$ is also a submatrix of $\boldsymbol{M}^C$.

Consider when we are generating the $(t+1)$-th token. As in (2)

$$
\boldsymbol{U}^t = \begin{pmatrix} \boldsymbol{u}_1^t \\ \boldsymbol{u}_2^t \\ \vdots \\ \boldsymbol{u}_t^t \end{pmatrix} = \text{sigmoid}\Big(\frac{\boldsymbol{Q}_t^U \boldsymbol{K}_t^{U\top}}{\sqrt{d}} + \boldsymbol{M}_t^U\Big) \boldsymbol{V}_t^U \in \mathbb{R}^{t \times d}
$$

Then, the lookahead-key attention scores as in (5) are

$$
\boldsymbol{s}_t^U = \frac{\boldsymbol{q}_t^C \boldsymbol{U}^{t\top}}{\sqrt{d}} = \frac{\boldsymbol{q}_t^C \boldsymbol{V}_t^{U\top} \Big(\text{sigmoid}\big(\frac{\boldsymbol{Q}_t^U \boldsymbol{K}_t^{U\top}}{\sqrt{d}} + \boldsymbol{M}_t^U\big)\Big)^\top}{\sqrt{d}}. \tag{19}
$$

We will need the following lemma to proceed.

**Lemma 1.** *For any vector $\boldsymbol{a} \in \mathbb{R}^{1 \times t}$, let $\widetilde{\boldsymbol{a}} = (\boldsymbol{a}, \boldsymbol{0}^{1 \times (L-t)}) \in \mathbb{R}^{1 \times L}$, where $\boldsymbol{0}^{1 \times (L-t)}$ is the all-zeros vector of size $(1, L-t)$. Then,*

$$
\left(\boldsymbol{a}\left(\text{sigmoid}\Big(\frac{\boldsymbol{Q}_t^U \boldsymbol{K}_t^{U\top}}{\sqrt{d}} + \boldsymbol{M}_t^U\Big)\right)^\top, \boldsymbol{0}^{1 \times (L-t)}\right) = \widetilde{\boldsymbol{a}}\left(\text{sigmoid}\Big(\frac{\boldsymbol{Q}^U \boldsymbol{K}^{U\top}}{\sqrt{d}} + \boldsymbol{M}^U\Big)\right)^\top.
$$

*Proof of Lemma 1.* The proof is straightforward by the fact that

1. The upper triangular entries of the transposed matrix $\left(\text{sigmoid}\Big(\frac{\boldsymbol{Q}^U \boldsymbol{K}^{U\top}}{\sqrt{d}} + \boldsymbol{M}^U\Big)\right)^\top$ are all 0 by the definition of $\boldsymbol{M}^U$.

2. The matrix $\mathrm{sigmoid}\left(\frac{Q_t^U K_t^{U\top}}{\sqrt{d}} + M_t^U\right)$ equals an upper-left block of the matrix $\mathrm{sigmoid}\left(\frac{Q^U K^{U\top}}{\sqrt{d}} + M^U\right)$, i.e.,

$$\mathrm{sigmoid}\left(\frac{Q_t^U K_t^{U\top}}{\sqrt{d}} + M_t^U\right) = \left[\mathrm{sigmoid}\left(\frac{Q^U K^{U\top}}{\sqrt{d}} + M^U\right)\right]_{1:t,1:t},$$

where for any matrix $A$, $A_{1:t,1:t}$ refers to its top-left $t$-by-$t$ submatrix.

$\square$

Define the vector $a_t \in \mathbb{R}^{1\times L}$ as $[a_t]_i = \left(q_t^C V_t^U\right)_i$ if $1 \le i \le t$ and $[a_t]_i = 0$ otherwise. Denote $\widetilde{s}_t^U = \left(s_t^U, 0^{1\times(L-t)}\right)$. Then, by combining (19) with Lemma 1, we have

$$
\widetilde{s}_t^U = \left(s_t^U, 0^{1\times(L-t)}\right) = \left(\frac{q_t^C V_t^{U\top}\left(\mathrm{sigmoid}\left(\frac{Q_t^U K_t^{U\top}}{\sqrt{d}} + M_t^U\right)\right)^\top}{\sqrt{d}}, 0^{1\times(L-t)}\right)
$$

$$
= \frac{a_t\left(\mathrm{sigmoid}\left(\frac{Q^U K^{U\top}}{\sqrt{d}} + M^U\right)\right)^\top}{\sqrt{d}}.
$$

Since $V_t^U$ equals the submatrix which consists of first $t$ rows of $V^U$, $[a_t]_i = \left(q_t^C V^{U\top}\right)_i$ for $1 \le i \le t$. Then, by stacking $(a_j)$ together, we have

$$
\begin{pmatrix} a_1 \\ a_2 \\ \vdots \\ a_L \end{pmatrix} = \left(Q^C V^{U\top}\right) \odot \widetilde{M}^C,
$$

where $\widetilde{M}^C$ is defined in Theorem 1 with $\widetilde{M}_{ij}^C = 1$ if $i \ge j$ and $\widetilde{M}_{ij}^C = 0$ otherwise.

We concatenate $\left(\widetilde{s}_t^U\right)_{1\le t\le L}$ in an $L$-by-$L$ matrix. Then,

$$
\begin{pmatrix} \widetilde{s}_1^U \\ \widetilde{s}_2^U \\ \vdots \\ \widetilde{s}_L^U \end{pmatrix} = \frac{1}{\sqrt{d}}\begin{pmatrix} a_1 \\ a_2 \\ \vdots \\ a_L \end{pmatrix}\left(\mathrm{sigmoid}\left(\frac{Q^U K^{U\top}}{\sqrt{d}} + M^U\right)\right)^\top
$$

$$
= \left(\frac{Q^C V^{U\top}}{\sqrt{d}} \odot \widetilde{M}^C\right)\left(\mathrm{sigmoid}\left(\frac{Q^U K^{U\top}}{\sqrt{d}} + M^U\right)\right)^\top \tag{20}
$$

$$
= S^U,
$$

where $S^U$ is given in (10).

The causal-key attention scores $s_t^C$ in (4) is

$$
s_t^C = \frac{q_t^C K_t^{C\top}}{\sqrt{d}}.
$$

We also denote $\widetilde{\boldsymbol{s}}_t^C = \left(\boldsymbol{s}_t^C, (-\infty)^{1\times(L-t)}\right)$, where $(-\infty)^{1\times(L-t)}$ is a $(L-t)$-dimensional vector with all entries equaling $-\infty$. Then, by concatenating $\left(\widetilde{\boldsymbol{s}}_t^C\right)_{1\leq t\leq L}$ into $\boldsymbol{S}^C \in \mathbb{R}^{L\times L}$, we have

$$\boldsymbol{S}^C = \begin{pmatrix} \widetilde{\boldsymbol{s}}_1^C \\ \widetilde{\boldsymbol{s}}_2^C \\ \vdots \\ \widetilde{\boldsymbol{s}}_L^C \end{pmatrix} = \frac{\boldsymbol{Q}^C \boldsymbol{K}^{C\top}}{\sqrt{d}} + \boldsymbol{M}^C. \tag{21}$$

Then, the outputs $\mathrm{Attention}(\boldsymbol{X}^L)$ satisfies

$$\mathrm{Attention}\left(\boldsymbol{X}^L\right) = \begin{pmatrix} \mathrm{attention}\left(\boldsymbol{X}^1\right) \\ \mathrm{attention}\left(\boldsymbol{X}^2\right) \\ \vdots \\ \mathrm{attention}\left(\boldsymbol{X}^L\right) \end{pmatrix} = \begin{pmatrix} \mathrm{softmax}\left(\boldsymbol{s}_1^C - \mathrm{SiLU}(\boldsymbol{s}_1^U)\right)\boldsymbol{V}_1^C \\ \mathrm{softmax}\left(\boldsymbol{s}_2^C - \mathrm{SiLU}(\boldsymbol{s}_2^U)\right)\boldsymbol{V}_2^C \\ \vdots \\ \mathrm{softmax}\left(\boldsymbol{s}_L^C - \mathrm{SiLU}(\boldsymbol{s}_L^U)\right)\boldsymbol{V}_L^C \end{pmatrix}$$

$$= \begin{pmatrix} \mathrm{softmax}\left(\widetilde{\boldsymbol{s}}_1^C - \mathrm{SiLU}(\widetilde{\boldsymbol{s}}_1^U)\right) \\ \mathrm{softmax}\left(\widetilde{\boldsymbol{s}}_2^C - \mathrm{SiLU}(\widetilde{\boldsymbol{s}}_2^U)\right) \\ \vdots \\ \mathrm{softmax}\left(\widetilde{\boldsymbol{s}}_L^C - \mathrm{SiLU}(\widetilde{\boldsymbol{s}}_L^U)\right) \end{pmatrix} \boldsymbol{V}^C$$

$$= \mathrm{row\_softmax}\left(\boldsymbol{S}^C - \mathrm{SiLU}\left(\boldsymbol{S}^U\right)\right)\boldsymbol{V}^C$$

$$= \mathrm{row\_softmax}\left(\frac{\boldsymbol{Q}^C \boldsymbol{K}^{C\top}}{\sqrt{d}} + \boldsymbol{M}^C - \mathrm{SiLU}\left(\frac{\boldsymbol{S}^U}{\sqrt{d}}\right)\right)\boldsymbol{V}^C,$$

where the last inequality above is from (20) and (21).

This completes the proof of Theorem 1.

## C  FURTHER DETAILS ON MULTI-HEAD CASTLE

**Forward pass for multi-head CASTLE.** Denote $n = n_{\mathrm{head}}$. Given contextualized representations $\boldsymbol{X}^L$, for each head, we can get $\mathrm{Attention}_i(\boldsymbol{X}^L)$ as in (11). Then, the outputs of multi-head CAS-TLE can be obtained by

$$\mathrm{Multihead\text{-}Attention}(\boldsymbol{X}^L) = \mathrm{Concat}\left(\mathrm{Attention}_1(\boldsymbol{X}^L),\ldots,\mathrm{Attention}_n(\boldsymbol{X}^L)\right)\boldsymbol{W}^O \in \mathbb{R}^{L\times d}.$$

**Parameter count of multi-head CASTLE.** In each head, the learnable parameters are $\boldsymbol{W}_Q^U$, $\boldsymbol{W}_K^U$, $\boldsymbol{W}_V^U$, $\boldsymbol{W}_Q^C$, $\boldsymbol{W}_K^C$, $\boldsymbol{W}_V^C \in \mathbb{R}^{d_{\mathrm{model}}\times d}$. These parameters sum up to $6n_{\mathrm{head}}d_{\mathrm{model}}d$.

The matrix $\boldsymbol{W}^O$ has $n_{\mathrm{head}}dd_{\mathrm{model}}$ parameters. Thus, the multi-head CASTLE has $7n_{\mathrm{head}}dd_{\mathrm{model}}$ learnable parameters.

Multi-head CASTLE-SWL has identical formula and parameter counts with multi-head CASTLE and is omitted for clarity.

**Parameter count of standard causal attention.** The standard causal attention as in (1) has learnable parameters $\boldsymbol{W}_Q$, $\boldsymbol{W}_K$, $\boldsymbol{W}_V \in \mathbb{R}^{d_{\mathrm{model}}\times d}$ for each head and $\boldsymbol{W}^O \in \mathbb{R}^{nd\times d_{\mathrm{model}}}$. These parameters sum up to $4n_{\mathrm{head}}dd_{\mathrm{model}}$.

---

**Algorithm 1:** Efficient parallel forward pass

---

**Require:** $\boldsymbol{Q}^U$, $\boldsymbol{K}^U$, $\boldsymbol{V}^U$, $\boldsymbol{Q}^C$, $\boldsymbol{K}^C$, $\boldsymbol{V}^C \in \mathbb{R}^{L \times d}$

1 # Initialization
2 Initialize $\boldsymbol{D} = \boldsymbol{0}^{d \times L}$, $\boldsymbol{O} = \boldsymbol{0}^{L \times d}$, $\ell = \boldsymbol{0}^{L \times 1}$ and $\boldsymbol{m} = (-\infty)^{L \times 1}$ in HBM
3 # Diagonal blocks
4 **for** $j = 0, \ldots, N-1$ *(in parallel)* **do**
5      Load $\boldsymbol{Q}^C_{T_j,:}$, $\boldsymbol{K}^C_{T_j,:}$, $\boldsymbol{Q}^U_{T_j,:}$, $\boldsymbol{K}^U_{T_j,:}$, $\boldsymbol{V}^U_{T_j,:}$ from HBM to on-chip SRAM
6      On chip, compute $\boldsymbol{S}^U_{T_j,T_j}$ as in (26)
7      On chip, compute $\boldsymbol{A}_{T_j,T_j}$ as in (23)
8      Implement online softmax update for block $(T_j, T_j)$ by calling Algorithm 2
9 **end**
10 # 1-st, $\cdots$, $(N-1)$-th off-diagonal blocks
11 **for** $k = 1, \ldots, N-1$ *(sequential)* **do**
12      **for** $j = 0, \ldots, N-k-1$ *(in parallel)* **do**
13          Load $\boldsymbol{D}_{:,T_j}$, $\boldsymbol{Q}^U_{T_j,:}$, $\boldsymbol{Q}^C_{T_{j+k}}$, $\boldsymbol{K}^U_{T_{j+k-1},:}$, $\boldsymbol{K}^U_{T_{j+k},:}$, $\boldsymbol{V}^U_{T_{j+k-1},:}$, $\boldsymbol{V}^U_{T_{j+k},:}$ from HBM to
         on-chip SRAM
14          On chip, update $\boldsymbol{D}_{:,T_j}$ as in (27)
15          On chip, compute $\boldsymbol{S}^U_{T_{j+k},T_j}$ as in (28)
16          Write $\boldsymbol{D}_{:,T_j}$ to HBM
17          On chip, compute $\boldsymbol{A}_{T_{j+k},T_j}$ as in (23)
18          Implement online softmax update for block $(T_{j+k}, T_j)$ by calling Algorithm 2
19      **end**
20 **end**
21 Compute $\boldsymbol{O} \leftarrow \mathbf{diag}\,(\ell)^{-1}\,\boldsymbol{O} \in \mathbb{R}^{L \times d}$, $\boldsymbol{m} \leftarrow \boldsymbol{m} + \log(\ell) \in \mathbb{R}^{L \times 1}$
22 Save $\boldsymbol{m} \in \mathbb{R}^{L \times 1}$, $\boldsymbol{D} \in \mathbb{R}^{d \times L}$ for backward pass
23 Return the output $\boldsymbol{O} \in \mathbb{R}^{L \times d}$

---

**Algorithm 2:** Online softmax update for block $(T_i, T_j)$

---

**Require:** $\boldsymbol{A}_{T_i,T_j}$ on chip, $\ell$, $\boldsymbol{m}$, $\boldsymbol{O}$, $\boldsymbol{V}^C$ in HBM

1 Load $\ell_{T_i}$ and $\boldsymbol{m}_{T_i}$ from HBM to on-chip SRAM
2 On chip, compute $\boldsymbol{m}^{\text{new}}_{T_i} = \max(\boldsymbol{m}_{T_i}, \text{row\_max}(\boldsymbol{A}_{T_i,T_j}))$
3 On chip, compute $\widetilde{\boldsymbol{P}}_{T_i,T_j} = \exp(\boldsymbol{A}_{T_i,T_j} - \boldsymbol{m}^{\text{new}}_{T_i})$
4 On chip, compute $\ell^{\text{new}}_{T_i} = e^{\boldsymbol{m}_{T_i} - \boldsymbol{m}^{\text{new}}_{T_i}} \ell_{T_i} + \text{row\_sum}(\widetilde{\boldsymbol{P}}_{T_i,T_j})$
5 Write $\boldsymbol{O}_{T_i,:} \leftarrow \mathbf{diag}\left(e^{\boldsymbol{m}_{T_i} - \boldsymbol{m}^{\text{new}}_{T_i}}\right) \boldsymbol{O}_{T_i,:} + \widetilde{\boldsymbol{P}}_{T_i,T_j} \boldsymbol{V}^C_{T_j,:}$ to HBM
6 Write $\ell_{T_i} \leftarrow \ell^{\text{new}}_{T_i}$, $\boldsymbol{m}_{T_i} \leftarrow \boldsymbol{m}^{\text{new}}_{T_i}$ to HBM

---

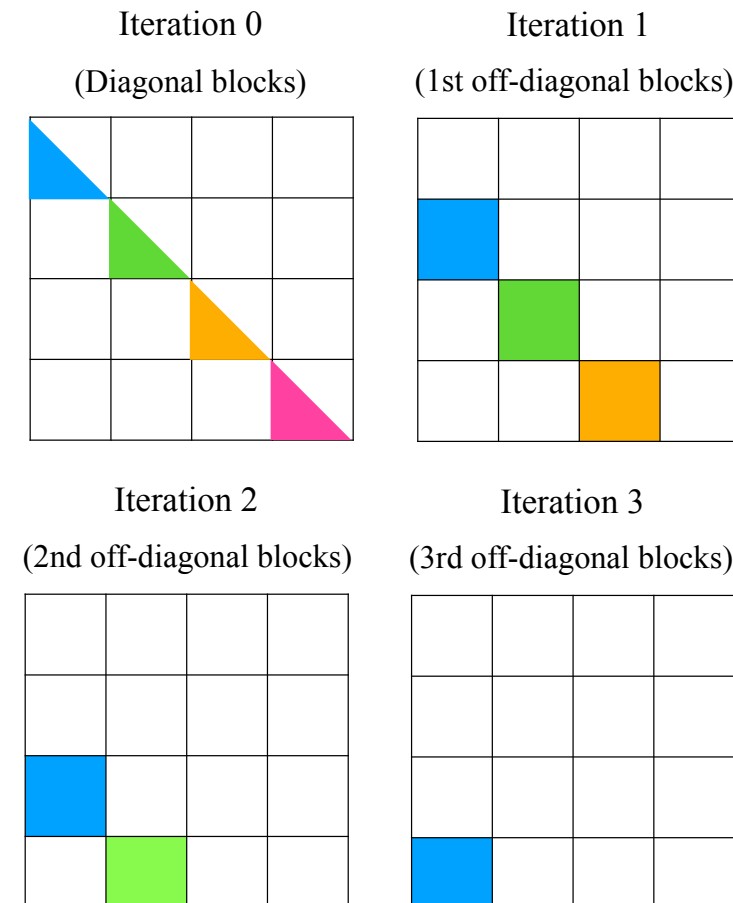

Figure 9: Parallel scheme of Algorithm 1 (forward pass). We begin by computing the diagonal blocks of the attention score matrix $\boldsymbol{A}$ (Iteration 0). In each subsequent iteration $k$, the $k$-th off-diagonal blocks are computed. Blocks with different colors represent different kernel instances. In each iteration, each kernel instance is responsible for computing a single block of $\boldsymbol{A}$ and applying online softmax (Algorithm 2) to it. Kernel instances within the same iteration are launched in parallel, while the iterations are executed sequentially. The parallel scheme of Algorithm 3 (backward pass) is the reverse of the forward pass's parallel scheme.

## D  EFFICIENT PARALLEL TRAINING ALGORITHM AND PROOF OF THEOREM 2

### D.1  FORWARD PASS

Theorem 1 has shown a matrix form of $\mathrm{Attention}(\boldsymbol{X}^L)$ as in (11). However, computing $\mathrm{Attention}(\boldsymbol{X}^L)$ directly from (11) still need $O(L^3)$ computational costs because to compute $\boldsymbol{S}^U$, we need matrix multiplication between $L$-by-$L$ matrices in (10). In this section, we give an efficient algorithm that enables efficient parallel training and reduces computational complexity to $O(L^2 d)$.

We first introduce the notations in this section. We divide the sequence $\{1, \ldots, L\}$ into blocks of size $B$. For simplicity, we assume that $L$ is divisible by $B$. Let $N = \frac{L}{B}$ and $T_i = \{i * B + 1, \ldots, (i+1) * B\}$ for $0 \leq i \leq N - 1$. Then, $\{1, 2, \ldots, L\} = \cup_{i=0}^{N-1} T_i$.

For any matrix $\boldsymbol{M}$, we use $\boldsymbol{M}_{T_i, T_j}$ to denote the submatrix whose row and column indexes are in $T_i$ and $T_j$, respectively. $\boldsymbol{M}_{T_i, :}$ refers to the submatrix whose row indexes are in $T_i$. Analogously, $\boldsymbol{M}_{:, T_j}$ refers to the submatrix whose column indexes are in $T_j$.

Denote the attention score matrix $\boldsymbol{A}$ by

$$\boldsymbol{A} = \frac{\boldsymbol{Q}^C \boldsymbol{K}^{C\top}}{\sqrt{d}} + \boldsymbol{M}^C - \mathrm{SiLU}\left(\boldsymbol{S}^U\right). \tag{22}$$

Then,

$$\mathrm{Attention}(\boldsymbol{X}^L) = \mathrm{row\_softmax}(\boldsymbol{A})\,\boldsymbol{V}^C$$

and for any $0 \le i, j \le N - 1$, the block $\boldsymbol{A}_{T_i, T_j}$ satisfies

$$\boldsymbol{A}_{T_i, T_j} = \frac{\boldsymbol{Q}^C_{T_i,:} \boldsymbol{K}^C_{T_j,:}{}^{\top}}{\sqrt{d}} + \boldsymbol{M}^C_{T_i, T_j} - \mathrm{SiLU}\left(\boldsymbol{S}^U_{T_i, T_j}\right). \tag{23}$$

As in FlashAttention-2 (Dao, 2024), we compute each block of $\boldsymbol{A}$ and apply online softmax (Algorithm 2) on it to obtain $\mathrm{Attention}(\boldsymbol{X}^L)$. The first term $\frac{\boldsymbol{Q}^C_{T_i,:} \boldsymbol{K}^C_{T_j,:}{}^{\top}}{\sqrt{d}}$ in $\boldsymbol{A}_{T_i, T_j}$ as given by (23) can be computed similarly to FlashAttention-2. We mainly focus on the second term $-\mathrm{SiLU}\left(\boldsymbol{S}^U_{T_i, T_j}\right)$ which can incur a total computational complexity of $O(L^3 + L^2 d)$ if we apply (10) directly. Notice that as in (10),

$$\boldsymbol{S}^U = \left(\frac{\boldsymbol{Q}^C \boldsymbol{V}^{U\top}}{\sqrt{d}} \odot \widetilde{\boldsymbol{M}}^C\right) \left(\mathrm{sigmoid}\left(\frac{\boldsymbol{Q}^U \boldsymbol{K}^{U\top}}{\sqrt{d}} + \boldsymbol{M}^U\right)\right)^{\top},$$

where the term $\left(\frac{\boldsymbol{Q}^C \boldsymbol{V}^{U\top}}{\sqrt{d}} \odot \widetilde{\boldsymbol{M}}^C\right)$ is a low-rank matrix multiplied by a mask matrix because $\boldsymbol{Q}^C \boldsymbol{V}^{U\top}$ has rank at most $d$ which is normally much smaller than sequence length $L$. This motivates our algorithm to compute $\boldsymbol{S}^U$ with computational complexity of $O(L^2 d)$ while still enabling parallel training.

First, as the upper triangular entries in $\boldsymbol{S}^U$ are all 0, we only need to focus on lower triangular entries in $\boldsymbol{S}^U$. We divide the computation of $\boldsymbol{S}^U$ into the following parts

- **diagonal blocks:** $\boldsymbol{S}^U_{T_j, T_j}$ with $0 \le j \le N - 1$.
- **k-th off-diagonal blocks:** $\boldsymbol{S}^U_{T_{j+k}, T_j}$ with $0 \le j \le N - 1 - k$

By the definition of $\boldsymbol{S}^U$, we can write $\boldsymbol{S}^U_{T_{j+k}, T_j}$ in a blockwise way as follows

$$\begin{aligned}
\boldsymbol{S}^U_{T_{j+k}, T_j} = \boldsymbol{Q}^C_{T_{j+k},:} \sum_{i=j}^{j+k-1} \frac{\boldsymbol{V}^U_{T_i,:}{}^{\top}}{\sqrt{d}} \left(\mathrm{sigmoid}\left(\frac{\boldsymbol{Q}^U_{T_j,:} \boldsymbol{K}^U_{T_i,:}{}^{\top}}{\sqrt{d}} + \boldsymbol{M}^U_{T_j, T_i}\right)\right)^{\top} \\
+ \left(\frac{\boldsymbol{Q}^C_{T_{j+k},:} \boldsymbol{V}^U_{T_{j+k},:}{}^{\top}}{\sqrt{d}} \odot \widetilde{\boldsymbol{M}}^C_{T_{j+k}, T_{j+k}}\right) \left(\mathrm{sigmoid}\left(\frac{\boldsymbol{Q}^U_{T_j,:} \boldsymbol{K}^U_{T_{j+k},:}{}^{\top}}{\sqrt{d}} + \boldsymbol{M}^U_{T_j, T_{j+k}}\right)\right)^{\top}.
\end{aligned} \tag{24}$$

However, computing each $\boldsymbol{S}^U_{T_{j+k}, T_j}$ this way will lead to $O(L^3)$ computational cost in total. We need more efficient way to reduce training costs.

The key observation in the development of this efficient parallel training algorithm is that the matrix $\left(\boldsymbol{Q}^C \boldsymbol{V}^{U\top}\right) \odot \widetilde{\boldsymbol{M}}^C$ is a low-rank matrix multiplied by a mask because $\boldsymbol{Q}^C \boldsymbol{V}^{U\top}$ has rank equal to or less than the head dimension $d$ rather than the sequence length $L$. This enables us to compute $\boldsymbol{S}^U$ defined in (10) with lower computational costs. More specifically, we will rely on the following auxiliary variable to reduce computational costs.

We first init an auxiliary variable $\boldsymbol{D}^{(0)} = \boldsymbol{0}^{d \times L}$. Then, when computing the $k$-th off-diagonal blocks, we will maintain the following relation that

$$\boldsymbol{D}^{(k)}_{:, T_j} = \sum_{i=j}^{j+k-1} \boldsymbol{V}^U_{T_i,:}{}^{\top} \left(\mathrm{sigmoid}\left(\frac{\boldsymbol{Q}^U_{T_j,:} \boldsymbol{K}^U_{T_i,:}{}^{\top}}{\sqrt{d}} + \boldsymbol{M}^U_{T_j, T_i}\right)\right)^{\top}. \tag{25}$$

We will compute the blocks of $\boldsymbol{S}^U$ in the following order: diagonal blocks, 1st off-diagonal blocks, 2nd off-diagonal blocks, $\cdots$, $(N-1)$-th off-diagonal blocks.

**Diagonal blocks of $\boldsymbol{S}^U$.** For any $0 \leq j \leq N - 1$,

$$
\boldsymbol{S}^U_{T_j,T_j} = \left( \frac{\boldsymbol{Q}^C_{T_j,:} {\boldsymbol{V}^U_{T_j,:}}^\top}{\sqrt{d}} \odot \widetilde{\boldsymbol{M}}^C_{T_j,T_j} \right) \left( \mathrm{sigmoid}\left( \frac{\boldsymbol{Q}^U_{T_j,:} {\boldsymbol{K}^U_{T_j,:}}^\top}{\sqrt{d}} + \boldsymbol{M}^U_{T_j,T_j} \right) \right)^\top . \tag{26}
$$

**1st off-diagonal blocks of $\boldsymbol{S}^U$.** For any $0 \leq j \leq N - 2$, update $\boldsymbol{D}^{(1)}_{:,T_j}$ as

$$
\boldsymbol{D}^{(1)}_{:,T_j} = {\boldsymbol{V}^U_{T_j,:}}^\top \left( \mathrm{sigmoid}\left( \frac{\boldsymbol{Q}^U_{T_j,:} {\boldsymbol{K}^U_{T_j,:}}^\top}{\sqrt{d}} + \boldsymbol{M}^U_{T_j,T_j} \right) \right)^\top
$$

This satisfies (25) with $k = 1$. Then, it follows from (24) that

$$
\boldsymbol{S}^U_{T_{j+1},T_j} = \frac{\boldsymbol{Q}^C_{T_{j+1},:} \boldsymbol{D}^{(1)}_{:,T_j}}{\sqrt{d}}
$$
$$
+ \left( \frac{\boldsymbol{Q}^C_{T_{j+1},:} {\boldsymbol{V}^U_{T_{j+1},:}}^\top}{\sqrt{d}} \odot \widetilde{\boldsymbol{M}}^C_{T_{j+1},T_{j+1}} \right) \left( \mathrm{sigmoid}\left( \frac{\boldsymbol{Q}^U_{T_j,:} {\boldsymbol{K}^U_{T_{j+1},:}}^\top}{\sqrt{d}} + \boldsymbol{M}^U_{T_j,T_{j+1}} \right) \right)^\top .
$$

**The $k$-th off-diagonal blocks of $\boldsymbol{S}^U$.** If we have already computed the $(k-1)$-th off-diagonal blocks and $\boldsymbol{D}^{(k-1)}$ satisfies (25) with $k - 1$, the $k$-th diagonal blocks can be computed in the following way. First, we update $\boldsymbol{D}^{(k)}$ as follows

$$
\boldsymbol{D}^{(k)}_{:,T_j} = \boldsymbol{D}^{(k-1)}_{:,T_j} + {\boldsymbol{V}^U_{T_{j+k-1},:}}^\top \left( \mathrm{sigmoid}\left( \frac{\boldsymbol{Q}^U_{T_j,:} {\boldsymbol{K}^U_{T_{j+k-1},:}}^\top}{\sqrt{d}} + \boldsymbol{M}^U_{T_j,T_{j+k-1}} \right) \right)^\top . \tag{27}
$$

By induction hypothesis (25), $\boldsymbol{D}^{(k)}_{:,T_j}$ also satisfies (25) with $k$. Then, it follows by (24) that

$$
\boldsymbol{S}^U_{T_{j+k},T_j} = \frac{\boldsymbol{Q}^C_{T_{j+k},:} \boldsymbol{D}^{(k)}_{:,T_j}}{\sqrt{d}}
$$
$$
+ \left( \frac{\boldsymbol{Q}^C_{T_{j+k},:} {\boldsymbol{V}^U_{T_{j+k},:}}^\top}{\sqrt{d}} \odot \widetilde{\boldsymbol{M}}^C_{T_{j+k},T_{j+k}} \right) \left( \mathrm{sigmoid}\left( \frac{\boldsymbol{Q}^U_{T_j,:} {\boldsymbol{K}^U_{T_{j+k},:}}^\top}{\sqrt{d}} + \boldsymbol{M}^U_{T_j,T_{j+k}} \right) \right)^\top . \tag{28}
$$

We can compute all $k$-th off-diagonal blocks in the above way. When computing each $\boldsymbol{S}^U_{T_{j+k},T_j}$, we only need to update $\boldsymbol{D}^{(k)}$ and then compute $\boldsymbol{S}^U_{T_{j+k},T_j}$ as (28). Both (27) and (28) take $O(L^2 d)$ FLOPs.

Next, we describe the design of a parallel algorithm. The parallelization scheme must satisfy the following requirements:

- $\boldsymbol{S}^U_{T_{j+k},T_j}$ must be computed after $\boldsymbol{S}^U_{T_{j+k-1},T_j}$ because computing $\boldsymbol{S}^U_{T_{j+k},T_j}$ in (28) requires $\boldsymbol{D}^{(k)}$ which is derived by updating $\boldsymbol{D}^{(k-1)}$ in (27). Therefore, $\boldsymbol{A}_{T_{j+k},T_j}$ should be computed after $\boldsymbol{A}_{T_{j+k-1},T_j}$.
- To ensure correctness, online softmax cannot be applied simultaneously to $\boldsymbol{A}_{T_i,T_j}$ and $\boldsymbol{A}_{T_i,T_k}$ for $j \neq k$.

To meet these constraints, we launch kernel instances to compute attention outputs with respect to blocks in the following order: starting with the diagonal blocks, followed by the first off-diagonal

blocks, then the second, and so on up to the $(N-1)$-th off-diagonal blocks. This parallel execution strategy for Algorithm 1 is illustrated in Figure 9.

As we can see, we launch one kernel instance for each block $\boldsymbol{S}_{T_i,T_j}$ with $i \geq j$. Thus, there are

$$\frac{L}{B} \cdot \left( \frac{L}{B} - 1 \right) = O\left( \frac{L^2}{B^2} \right) \quad \text{block.}$$

As in Algorithm 1, inside each block, we only have $O(1)$ matrix multiplications between $B$-by-$d$ matrices or $B$-by-$B$ matrices, where $B$ is the block size. Other operations are entrywise. In additional, we choose $B = O(d)$. (In practice, we found that $B = 64$ optimizes the performance of our training kernels.) Thus, each block has only $O(B^2 d + B^3) = O(B^2 d)$ FLOPs. Thus, the computational complexity is

$$O\left( \frac{L^2}{B^2} \right) \cdot O\left( B^2 d \right) = L^2 d.$$

It is straightforward that the space complexity is $O(Ld)$.

## D.2 BACKWARD PASS

In this section, we introduce the backward pass algorithm for efficient parallel training. It is mainly derived by reversing the forward pass.

Recall that in the forward pass, we compute the blocks in the following order: diagonal blocks, 1st off-diagonal blocks, 2nd off-diagonal blocks, $\cdots$, $(N-1)$-th off-diagonal blocks. Then, in the backward pass, we compute the derives in the inverse order as follows: $(N-1)$-th off-diagonal blocks, $(N-2)$-th off-diagonal blocks, $\cdots$, 1-st off-diagonal blocks, diagonal blocks.

As in (Dao et al., 2022), we first preprocess $\Delta \in \mathbb{R}^{L \times 1}$ with

$$\Delta_i = \mathrm{d}\boldsymbol{O}_{i,:}^{\top} \boldsymbol{O}_{i,:}.$$

Let $\boldsymbol{D}^{(N-1)}$ be the $\boldsymbol{D}$ saved for backward pass in Algorithm 1. Also, let $\boldsymbol{m}$ be the vector $\boldsymbol{m}$ saved for backward pass in Algorithm 1. Set $\mathrm{d}\boldsymbol{D}^{(N-1)} = \boldsymbol{0}^{d \times L}$.

Then, we iterate over $k = N-1, \ldots, 1$. In each iteration, we compute the corresponding gradients and update $\boldsymbol{D}^{(k)}$ to $\boldsymbol{D}^{(k-1)}$ inversely as in the forward pass. Thus, for each $k$, before we launch kernels for the $k$-th off-diagonal blocks, we already have $\boldsymbol{D}^{(k)}$.

**$k$-th off-diagonal blocks.** For any $0 \leq j \leq N-1-k$, we first compute $\boldsymbol{S}_{T_{j+k},T_j}^{U}$ as in (28) and $\boldsymbol{A}_{T_{j+k},T_j}$ as in (23). Recall that in the end of the forward pass (Algorithm 1), we have set $\boldsymbol{m} \leftarrow \boldsymbol{m} + \log(\ell)$. Then, the attention weights can be computed by

$$\boldsymbol{P}_{T_{j+k},T_j} = \exp\left( \boldsymbol{A}_{T_{j+k},T_k} - \boldsymbol{m}_{T_{j+k}} \right). \tag{29}$$

Compute the gradient of attention weights

$$\mathrm{d}\boldsymbol{P}_{T_{j+k},T_j} = \mathrm{d}\boldsymbol{O}_{T_{j+k},:} \boldsymbol{V}_{T_j,:}^{\top} \tag{30}$$

Then, update the gradient of $\boldsymbol{V}_{T_j,:}^{C}$ as follows

$$\mathrm{d}\boldsymbol{V}_{T_j,:}^{C} \leftarrow \mathrm{d}\boldsymbol{V}_{T_j,:}^{C} + \boldsymbol{P}_{T_{j+k},T_j}^{\top} \mathrm{d}\boldsymbol{O}_{T_{j+k},:}. \tag{31}$$

Then, compute the gradients of attention weights

$$\mathrm{d}\boldsymbol{S}_{T_{j+k},T_j}^{C} = \boldsymbol{P}_{T_{j+k},T_j} \odot \left( \mathrm{d}\boldsymbol{P}_{T_{j+k},T_j} - \Delta_{T_{j+k}} \right) \tag{32}$$

and

$$\mathrm{d}\boldsymbol{S}_{T_{j+k},T_j}^{U} = -\nabla \mathrm{SiLU}\left( \boldsymbol{S}_{T_{j+k},T_j}^{U} \right) \odot \boldsymbol{P}_{T_{j+k},T_j} \odot \left( \mathrm{d}\boldsymbol{P}_{T_{j+k},T_j} - \Delta_{T_{j+k}} \right). \tag{33}$$

---

**Algorithm 3:** Efficient parallel backward pass

---

**Require:** $\mathrm{d}\boldsymbol{O} \in \mathbb{R}^{L \times d_{\mathrm{model}}}$

1 # Initialization

2 Initialize $\mathrm{d}\boldsymbol{D} = \boldsymbol{0}^{d \times L}, \mathrm{d}\boldsymbol{Q}^U = \boldsymbol{0}^{L \times d}, \mathrm{d}\boldsymbol{K}^U = \boldsymbol{0}^{L \times d}, \mathrm{d}\boldsymbol{V}^U = \boldsymbol{0}^{L \times d}, \mathrm{d}\boldsymbol{Q}^C = \boldsymbol{0}^{L \times d},$
   $\mathrm{d}\boldsymbol{K}^C = \boldsymbol{0}^{L \times d}, \mathrm{d}\boldsymbol{V}^C = \boldsymbol{0}^{L \times d}$

3 Let $\boldsymbol{m} \in \mathbb{R}^{L \times 1}, \boldsymbol{D} \in \mathbb{R}^{d \times L}$ be the tensors saved in forward pass (Algorithm 1)

4 # Preprocess

5 Compute $\Delta \in \mathbb{R}^{L \times 1}$ with $\Delta_i = \mathrm{d}\boldsymbol{O}_{i,:}^\top \boldsymbol{O}_{i,:}$

6 # $(N-1)$-th, $\cdots$, 1st off-diagonal blocks

7 **for** $k = N-1, \ldots, 1$ *(sequential)* **do**

8     **for** $j = 0, \ldots, N-k-1$ *(in parallel)* **do**

9         Compute $\boldsymbol{S}_{T_{j+k},T_j}^U$ as in (28) and $\boldsymbol{A}_{T_{j+k},T_j}$ as in (23)

10         Compute $\boldsymbol{P}_{T_{j+k},T_j}$ and $\mathrm{d}\boldsymbol{P}_{T_{j+k},T_j}$ as in (29) and (30)

11         Compute $\mathrm{d}\boldsymbol{S}_{T_{j+k},T_j}^C$ and $\mathrm{d}\boldsymbol{S}_{T_{j+k},T_j}^U$ as in (32) and (33)

12         Update $\mathrm{d}\boldsymbol{V}_{T_j,:}^C, \mathrm{d}\boldsymbol{Q}_{T_{j+k},:}^C, \mathrm{d}\boldsymbol{K}_{T_j,:}^C$ as in (31) and (34)

13         Update $\mathrm{d}\boldsymbol{Q}_{T_{j+k},:}^C, \mathrm{d}\boldsymbol{D}_{:,T_j}^{(k)}, \mathrm{d}\boldsymbol{V}_{T_{j+k},:}^U, \mathrm{d}\boldsymbol{Q}_{T_j,:}^U, \boldsymbol{K}_{T_{j+k},:}^U$ as in (35)

14         Update $\mathrm{d}\boldsymbol{V}_{T_{j+k-1},:}^U, \mathrm{d}\boldsymbol{Q}_{T_j,:}^U, \mathrm{d}\boldsymbol{K}_{T_{j+k-1},:}^U$ as in (36)

15         Compute $\boldsymbol{D}_{:,T_j}^{(k-1)}$ as in (37)

16     **end**

17 **end**

18 # Diagonal blocks

19 **for** $j = 0, \ldots, N-1$ *(in parallel)* **do**

20     Compute $\boldsymbol{S}_{T_j,T_j}^U$ as in (26) and $\boldsymbol{A}_{T_j,T_j}$ as in (23)

21     Compute $\boldsymbol{P}_{T_j,T_j}$ and $\mathrm{d}\boldsymbol{P}_{T_j,T_j}$ as in (38) and (39)

22     Compute $\mathrm{d}\boldsymbol{S}_{T_j,T_j}^C$ and $\mathrm{d}\boldsymbol{S}_{T_j,T_j}^U$ as in (41) and (42)

23     Update $\mathrm{d}\boldsymbol{V}_{T_j,:}^C, \mathrm{d}\boldsymbol{Q}_{T_j,:}^C, \mathrm{d}\boldsymbol{V}_{T_j,:}^U, \mathrm{d}\boldsymbol{Q}_{T_j,:}^U, \mathrm{d}\boldsymbol{K}_{T_j,:}^U$ as in (40) and (43)

24 **end**

25 Return the gradients $\mathrm{d}\boldsymbol{Q}^U, \mathrm{d}\boldsymbol{K}^U, \mathrm{d}\boldsymbol{V}^U, \mathrm{d}\boldsymbol{Q}^C, \mathrm{d}\boldsymbol{K}^C, \mathrm{d}\boldsymbol{V}^C \in \mathbb{R}^{L \times d}$

---

Then, we update the gradients of $\boldsymbol{Q}^C_{T_{j+k},:}$, $\boldsymbol{K}^C_{T_j,:}$ through the backward pass of (23) as follows

$$
\begin{aligned}
\mathrm{d}\boldsymbol{Q}^C_{T_{j+k},:} &\leftarrow \mathrm{d}\boldsymbol{Q}^C_{T_{j+k},:} + \frac{\mathrm{d}\boldsymbol{S}^C_{T_{j+k},T_j}\boldsymbol{K}^C_{T_j,:}}{\sqrt{d}}, \\
\mathrm{d}\boldsymbol{K}^C_{T_j,:} &\leftarrow \mathrm{d}\boldsymbol{K}^C_{T_j,:} + \frac{\mathrm{d}\boldsymbol{S}^C_{T_{j+k},T_j}{}^\top \boldsymbol{Q}^C_{T_{j+k},:}}{\sqrt{d}}.
\end{aligned}
\tag{34}
$$

Then, we update the gradients of $\boldsymbol{Q}^C_{T_{j+k},:}$, $\boldsymbol{D}^{(k)}_{:,T_j}$, $\boldsymbol{V}^U_{T_{j+k},:}$, $\boldsymbol{Q}^U_{T_j,:}$, $\boldsymbol{K}^U_{T_{j+k},:}$ through the backward pass of (28) as follows

$$
\begin{aligned}
\mathrm{d}\boldsymbol{Q}^C_{T_{j+k},:} &\leftarrow \mathrm{d}\boldsymbol{Q}^C_{T_{j+k},:} + \frac{\mathrm{d}\boldsymbol{S}^U_{T_{j+k},T_j}\boldsymbol{D}^{(k)}_{:,T_j}{}^\top}{\sqrt{d}} + \frac{\boldsymbol{E}\,\boldsymbol{V}^U_{T_{j+k},:}}{\sqrt{d}} \\
\mathrm{d}\boldsymbol{D}^{(k)}_{:,T_j} &\leftarrow \mathrm{d}\boldsymbol{D}^{(k)}_{:,T_j} + \frac{\boldsymbol{Q}^C_{T_{j+k},:}{}^\top \mathrm{d}\boldsymbol{S}^U_{T_{j+k},T_j}}{\sqrt{d}} \\
\mathrm{d}\boldsymbol{V}^U_{T_{j+k},:} &\leftarrow \mathrm{d}\boldsymbol{V}^U_{T_{j+k},:} + \frac{\boldsymbol{E}^\top \boldsymbol{Q}^C_{T_{j+k},:}}{\sqrt{d}} \\
\mathrm{d}\boldsymbol{Q}^U_{T_j,:} &\leftarrow \mathrm{d}\boldsymbol{Q}^U_{T_j,:} + \frac{\boldsymbol{F}\,\boldsymbol{K}^U_{T_{j+k},:}}{\sqrt{d}} \\
\mathrm{d}\boldsymbol{K}^U_{T_{j+k},:} &\leftarrow \mathrm{d}\boldsymbol{K}^U_{T_{j+k},:} + \frac{\boldsymbol{F}^\top \boldsymbol{Q}^U_{T_j,:}}{\sqrt{d}},
\end{aligned}
\tag{35}
$$

where auxiliary matrices

$$
\boldsymbol{E} = \left( \mathrm{d}\boldsymbol{S}^U_{T_{j+k},T_j}\,\mathrm{sigmoid}\!\left( \frac{\boldsymbol{Q}^U_{T_j,:}\boldsymbol{K}^U_{T_{j+k},:}{}^\top}{\sqrt{d}} + \boldsymbol{M}^U_{T_j,T_{j+k}} \right) \right) \odot \widetilde{\boldsymbol{M}}^C_{T_{j+k},T_{j+k}},
$$

$$
\boldsymbol{F} = \left( \left( \mathrm{d}\boldsymbol{S}^U_{T_{j+k},T_j} \right)^\top \left( \frac{\boldsymbol{Q}^C_{T_{j+k},:}\boldsymbol{V}^U_{T_{j+k},:}{}^\top}{\sqrt{d}} \odot \widetilde{\boldsymbol{M}}^C_{T_{j+k},T_{j+k}} \right) \right)
$$

$$
\odot \nabla\mathrm{sigmoid}\!\left( \frac{\boldsymbol{Q}^U_{T_j,:}\boldsymbol{K}^U_{T_{j+k},:}{}^\top}{\sqrt{d}} + \boldsymbol{M}^U_{T_j,T_{j+k}} \right).
$$

Then, we update the gradients of $\boldsymbol{V}^U_{T_{j+k-1},:}$, $\boldsymbol{Q}^U_{T_j,:}$, $\boldsymbol{K}^U_{T_{j+k-1},:}$ from the backward pass of (27) as follows

$$
\begin{aligned}
\mathrm{d}\boldsymbol{V}^U_{T_{j+k-1},:} &\leftarrow \mathrm{d}\boldsymbol{V}^U_{T_{j+k-1},:} \\
&\quad + \left( \mathrm{sigmoid}\!\left( \frac{\boldsymbol{Q}^U_{T_j,:}\boldsymbol{K}^U_{T_{j+k-1},:}{}^\top}{\sqrt{d}} + \boldsymbol{M}^U_{T_j,T_{j+k-1}} \right) \right)^\top \left( \mathrm{d}\boldsymbol{D}^{(k)}_{:,T_j} \right)^\top \\
\mathrm{d}\boldsymbol{Q}^U_{T_j,:} &\leftarrow \mathrm{d}\boldsymbol{Q}^U_{T_j,:} + \frac{\boldsymbol{G}\,\boldsymbol{K}^U_{T_{j+k-1},:}}{\sqrt{d}} \\
\mathrm{d}\boldsymbol{K}^U_{T_{j+k-1},:} &\leftarrow \mathrm{d}\boldsymbol{K}^U_{T_{j+k-1},:} + \frac{\boldsymbol{G}^\top \boldsymbol{Q}^U_{T_j,:}}{\sqrt{d}}
\end{aligned}
\tag{36}
$$

where the auxiliary matrix

$$
\boldsymbol{G} = \left( \left( \mathrm{d}\boldsymbol{D}^{(k)}_{:,T_j} \right)^\top \boldsymbol{V}^U_{T_{j+k-1},:}{}^\top \right) \odot \nabla\mathrm{sigmoid}\!\left( \frac{\boldsymbol{Q}^U_{T_j,:}\boldsymbol{K}^U_{T_{j+k-1},:}{}^\top}{\sqrt{d}} + \boldsymbol{M}^U_{T_j,T_{j+k-1}} \right).
$$

After updating gradients, we set $\mathrm{d}\boldsymbol{D}^{(k-1)}_{:,T_j} \leftarrow \mathrm{d}\boldsymbol{D}^{(k)}_{:,T_j}$ and get $\boldsymbol{D}^{(k-1)}_{:,T_j}$ back from $\boldsymbol{D}^{(k)}_{:,T_j}$ by reversing (27) as follows

$$
\boldsymbol{D}^{(k-1)}_{:,T_j} = \boldsymbol{D}^{(k)}_{:,T_j} - \boldsymbol{V}^U_{T_{j+k-1},:}{}^\top \left( \mathrm{sigmoid}\!\left( \frac{\boldsymbol{Q}^U_{T_j,:}\boldsymbol{K}^U_{T_{j+k-1},:}{}^\top}{\sqrt{d}} + \boldsymbol{M}^U_{T_j,T_{j+k-1}} \right) \right)^\top.
\tag{37}
$$

**Diagonal blocks.** For any $0 \leq j \leq N-1$, we first compute $\boldsymbol{S}_{T_j,T_j}^U$ as in (26) and $\boldsymbol{A}_{T_j,T_j}$ as in (23). Then, compute the attention weights

$$\boldsymbol{P}_{T_j,T_j} = \exp\left(\boldsymbol{A}_{T_j,T_j} - \boldsymbol{m}_{T_j}\right). \tag{38}$$

Compute the gradient of attention weights

$$\mathrm{d}\boldsymbol{P}_{T_j,T_j} = \mathrm{d}\boldsymbol{O}_{T_j,:}\,\boldsymbol{V}_{T_j,:}^\top. \tag{39}$$

Then, update the gradient of $\boldsymbol{V}_{T_j,:}^C$ as follows

$$\mathrm{d}\boldsymbol{V}_{T_j,:}^C \leftarrow \mathrm{d}\boldsymbol{V}_{T_j,:}^C + \boldsymbol{P}_{T_j,T_j}^\top \mathrm{d}\boldsymbol{O}_{T_j,:}. \tag{40}$$

Then, compute the gradients of attention weights

$$\mathrm{d}\boldsymbol{S}_{T_j,T_j}^C = \boldsymbol{P}_{T_j,T_j} \odot \left(\mathrm{d}\boldsymbol{P}_{T_j,T_j} - \Delta_{T_j}\right) \tag{41}$$

and

$$\mathrm{d}\boldsymbol{S}_{T_j,T_j}^U = -\nabla\mathrm{SiLU}\left(\boldsymbol{S}_{T_j,T_j}^U\right) \odot \boldsymbol{P}_{T_j,T_j} \odot \left(\mathrm{d}\boldsymbol{P}_{T_j,T_j} - \Delta_{T_j}\right). \tag{42}$$

Then, we update the gradients of $\boldsymbol{Q}_{T_j,:}^C$, $\boldsymbol{V}_{T_j,:}^U$, $\boldsymbol{Q}_{T_j,:}^U$, $\boldsymbol{K}_{T_j,:}^U$ through the backward pass of (26) as follows

$$
\begin{aligned}
\mathrm{d}\boldsymbol{Q}_{T_j,:}^C &\leftarrow \mathrm{d}\boldsymbol{Q}_{T_j,:}^C + \frac{\boldsymbol{E}\,\boldsymbol{V}_{T_j,:}^U}{\sqrt{d}}, \\
\mathrm{d}\boldsymbol{V}_{T_j,:}^U &\leftarrow \mathrm{d}\boldsymbol{V}_{T_j,:}^U + \frac{\boldsymbol{E}^\top \boldsymbol{Q}_{T_j,:}^C}{\sqrt{d}}, \\
\mathrm{d}\boldsymbol{Q}_{T_j,:}^U &\leftarrow \mathrm{d}\boldsymbol{Q}_{T_j,:}^U + \frac{\boldsymbol{F}\,\boldsymbol{K}_{T_j,:}^U}{\sqrt{d}}, \\
\mathrm{d}\boldsymbol{K}_{T_j,:}^U &\leftarrow \mathrm{d}\boldsymbol{K}_{T_j,:}^U + \frac{\boldsymbol{F}^\top \boldsymbol{Q}_{T_j,:}^U}{\sqrt{d}},
\end{aligned}
\tag{43}
$$

where auxiliary matrices

$$\boldsymbol{E} = \left(\mathrm{d}\boldsymbol{S}_{T_j,T_j}^U \mathrm{sigmoid}\left(\frac{\boldsymbol{Q}_{T_j,:}^U {\boldsymbol{K}_{T_j,:}^U}^\top}{\sqrt{d}} + \boldsymbol{M}_{T_j,T_j}^U\right)\right) \odot \widetilde{\boldsymbol{M}}_{T_j,T_j}^C,$$

$$\boldsymbol{F} = \left(\left(\mathrm{d}\boldsymbol{S}_{T_j,T_j}^U\right)^\top \left(\frac{\boldsymbol{Q}_{T_j,:}^C {\boldsymbol{V}_{T_j,:}^U}^\top}{\sqrt{d}} \odot \widetilde{\boldsymbol{M}}_{T_j,T_j}^C\right)\right) \odot \nabla\mathrm{sigmoid}\left(\frac{\boldsymbol{Q}_{T_j,:}^U {\boldsymbol{K}_{T_j,:}^U}^\top}{\sqrt{d}} + \boldsymbol{M}_{T_j,T_j}^U\right).$$

Then, the pseudo-code of backward pass is illustrated in Algorithm 3. The backward pass's parallel scheme is naturally the reverse of the forward pass's parallel scheme. We remark that when computing the gradients with respect to the block $(T_{j+k}, T_j)$, we update both $\boldsymbol{V}_{T_{j+k},:}^U$ and $\boldsymbol{V}_{T_{j+k-1},:}^U$. Consequently, the block $\boldsymbol{V}_{T_{j+k-1},:}^U$ receives contributions from two sources: $(T_{j+k}, T_j)$ and $(T_{j+k-1}, T_{j-1})$. To prevent overlapping updates, we introduce two auxiliary variables for storing intermediate results in $\mathrm{d}\boldsymbol{V}^U$, namely $\mathrm{d}\widehat{\boldsymbol{V}}^U$ and $\mathrm{d}\widetilde{\boldsymbol{V}}^U$. Specifically, in block $(T_{j+k}, T_j)$, the update to $\mathrm{d}\boldsymbol{V}_{T_{j+k},:}^U$ is accumulated in $\mathrm{d}\widehat{\boldsymbol{V}}^U$, while the update to $\mathrm{d}\boldsymbol{V}_{T_{j+k-1},:}^U$ is accumulated in $\mathrm{d}\widetilde{\boldsymbol{V}}^U$. After all gradients with respect to off-diagonal blocks have been computed, we obtain the true gradient of $\mathrm{d}\boldsymbol{V}^U$ by $\mathrm{d}\boldsymbol{V}^U \leftarrow \mathrm{d}\widehat{\boldsymbol{V}}^U + \mathrm{d}\widetilde{\boldsymbol{V}}^U$. The same procedure is applide to $\mathrm{d}\boldsymbol{K}^U$.

Similar to the forward pass, we launch one kernel instance for each block $\boldsymbol{S}_{T_i,T_j}$ with $i \geq j$. Thus, there are

$$\frac{L}{B} \cdot \left(\frac{L}{B} - 1\right) = O\left(\frac{L^2}{B^2}\right) \quad \text{block.}$$

Inside each block, as in Algorithm 3, we only have $O(1)$ matrix multiplications between $B$-by-$d$ matrices or $B$-by-$B$ matrices, where $B$ is the block size. And other operations are entrywise. In practice, we always choose $B = O(d)$. (We found that $B = 64$ optimizes the performance of our training kernels.) Thus, each block has only $O(B^2d + B^3) = O(B^2d)$ FLOPs. Thus, the computational complexity is

$$O\left(\frac{L^2}{B^2}\right) \cdot O\left(B^2 d\right) = L^2 d.$$

It is also straightforward that the space complexity is $O(Ld)$.

## E    EFFICIENT INFERENCE WITH UQ-KV CACHE

In this section, we present a detailed description of CASTLE's inference algorithm. Because the prefilling stage requires computing the UQ–KV cache, we first motivate the need for this cache by outlining the decoding process in Appendix E.1. Appendix E.2 compares the size of the UQ–KV cache in CASTLE with the standard KV cache used in causal attention. Appendix E.3 then provides the prefilling algorithm.

### E.1    DECODING ALGORITHM

Throughout this section, we denote batch size and head dimension by $d$, the number of heads by $n_{\text{castle\_head}}$. We fix one $t$ and consider the decoding process when generating the $(t+1)$-th token.

Decoding Algorithm consists of 2 steps: (1) rank-1 updating step and (2) combining step.

**Rank-1 Updating Step.** Directly computing $U^t$ from (2) at each position will lead to $\Omega(L^3 d)$ computational costs in total, which is prohibitive. Fortunately, we can instead compute $U^t$ from $U^{t-1}$ through a rank-1 update as in (12) as follows

$$U^t = \begin{pmatrix} U^{t-1} + \text{sigmoid}\left(\frac{Q_{t-1}^U k_t^{U\top}}{\sqrt{d}} + [M_t^U]_{1:t-1,t}\right) v_t^U \\ \mathbf{0}^{1\times d} \end{pmatrix}.$$

We prove the equivalence between 2 and (12) as follows.

*Proof of* (12). By (2),

$$u_s^t = \sum_{j=1}^t \text{sigmoid}\left(\frac{q_s^U k_j^{U\top}}{\sqrt{d}} + [M_t^U]_{sj}\right) v_j^U.$$

Thus, for $1 \le s < t$,

$$u_s^t - u_s^{t-1} = \text{sigmoid}\left(\frac{q_s^U k_t^{U\top}}{\sqrt{d}} + [M_t^U]_{st}\right) v_t^U.$$

Since $[M_t^U]_{t,t} = 0$, the last row of $U^t$ is all-zeros. This yields (12). $\qquad\square$

This step is called updating step since we are getting $U^t$ by rank-1 update from $U^{t-1}$.

To compute (12) efficiently, obviously, we need to store $U^{t-1}$ from the last step, as we need to compute $U^t$ from $U^{t-1}$. The U-cache from the previous step has the following size:

- U-cache stored from previous step: $[B, t-1, n_{\text{castle\_head}}, d]$.

In rank-1 updating step, there is one matrix-vector multiplication: $Q_{t-1}^U k_t^{U\top}$. Note that $[M_t^U]_{1:t-1,t}$ is the subvector of the last column of $M_t^U \in \mathbb{R}^{t\times t}$ defined in (3) for CASTLE and (8) for CASTLE-SWL. We have slightly different ways to deal with this matrix-vector multiplication in CASTLE and CASTLE-SWL because $M_t^U$ is different in CASTLE and CASTLE-SWL. This leads to Q-cache in CASTLE and CASTLE-SWL having different sizes.

**Q-cache in CASTLE.** By the definition of $M_t^U$ in (3), $[M_t^U]_{1:t-1,t} \in \mathbb{R}^{(t-1)\times 1}$ is an all-zeros vector, thus, by storing the entire $Q_{t-1}^U$, we can compute $Q_{t-1}^U k_t^U$ directly without computing $Q_{t-1}^U = X^{t-1} W_Q^U$. Therefore, in CASTLE, Q-cache from the previous step has the following size

- Q-cache stored from previous step in CASTLE: $[B, t-1, n_{\text{castle\_head}}, d]$.

**Q-cache in CASTLE-SWL.** By the definition of $M_t^U$ in (8) (an illustration can be found in Figure 11), the last $W$ entries in $[M_t^U]_{1:t-1,t}$ are all 0, and the other entries are all $-\infty$. Thus, in the vector $\text{sigmoid}\left(\frac{Q_{t-1}^U k_t^{U\top}}{\sqrt{d}} + [M_t^U]_{1:t-1,t}\right) \in \mathbb{R}^{(t-1)\times 1}$, its $s$-th entry is 0 if $s < t - W$, where $W$ is the window size. Therefore, we only need to store the last $W$ rows in $Q_{t-1}^U$. In CASTLE-SWL, the Q-cache from the previous step is of the following size:

- Q-cache stored from previous step in CASTLE-SWL: $[B, (t-1) \wedge W, n_{\text{castle\_head}}, d]$.

Then, we can get $U^t$ by rank-1 update from $U^{t-1}$ using the column vector $\text{sigmoid}\left(\frac{Q_{t-1}^U k_t^{U\top}}{\sqrt{d}} + [M_t^U]_{:,t}\right) \in \mathbb{R}^{(t-1)\times 1}$ and the row vector $v_t^U \in \mathbb{R}^{1\times d}$. This completes the rank-1 update step.

**Combining step.** Next, since we already have $U^t$, we can compute the output by going through (4), (5), (6) and (7) sequentially. Then, by the same reason with using KV cache in standard causal attention, we need to use $K_{t-1}^C$ and $V_{t-1}^C$ stored from the previous step which are of sizes

- K-cache stored from previous step: $[B, t-1, n_{\text{castle\_head}}, d]$,
- V-cache stored from previous step: $[B, t-1, n_{\text{castle\_head}}, d]$.

After computing the output, we need to renew UQ-KV cache to reduce computational costs for the next step. This yields the size of UQ-KV cache:

- U-cache size: $[B, t, n_{\text{castle\_head}}, d]$,
- Q-cache size:
  - In CASTLE: $[B, t, n_{\text{castle\_head}}, d]$
  - In CASTLE-SWL: $[B, t \wedge W, n_{\text{castle\_head}}, d]$ ($W$ is the window size)
- K-cache size: $[B, t, n_{\text{castle\_head}}, d]$,
- V-cache size: $[B, t, n_{\text{castle\_head}}, d]$,

In practice, cache renewal is incorporated into the combining step. The pseudocode of the process described above is in Algorithm 4.

### E.2 MEMORY REQUIREMENT OF UQ-KV CACHE

In this section, we compare the memory usage of the UQ–KV cache in CASTLE with that of the standard KV cache used in causal attention. We use Baseline-XL and CASTLE(-SWL)-XL as an example.

Denote the number of heads in baseline model and CASTLE by $n_{\text{head}}$ and $n_{\text{castle\_head}}$, respectively. Denote batch size, sequence length and head dimension by $B$, $T$ and $d$.

In practice, we use Bfloat16 to store KV cache in baseline models and UQ-KV cache in CASTLE and CASTLE-SWL.

As in Table 3, we set $n_{\text{castle\_head}} = \frac{9}{16} n_{\text{head}}$ to align the number of parameters.

**KV cache in Standard Causal Attention.** Both K cache and V cache are of size $[B, T, n_{\text{head}}, d]$. Then, the total memory requirement is

$$2 * B * T * n_{\text{head}} * 2 \text{ Byte} = 4\, BT n_{\text{head}}\, d \text{ Bytes.}$$

---

**Algorithm 4:** Decoding Algorithm (generating the $(t+1)$-th token)

---

**Require:** representation $\boldsymbol{x}_t \in \mathbb{R}^{1 \times d_{\text{model}}}$, UQ-KV cache $\boldsymbol{U}^{t-1}$, $\boldsymbol{Q}^U_{t-1}$, $\boldsymbol{K}^C_{t-1}$,
$\quad\quad \boldsymbol{V}^C_{t-1} \in \mathbb{R}^{(t-1) \times d}$

1 Compute $\boldsymbol{q}^U_t = \boldsymbol{x}_t \boldsymbol{W}^U_Q$, $\boldsymbol{k}^U_t = \boldsymbol{x}_t \boldsymbol{W}^U_K$, $\boldsymbol{v}^U_t = \boldsymbol{x}_t \boldsymbol{W}^U_V$

2 Compute $\boldsymbol{q}^C_t = \boldsymbol{x}_t \boldsymbol{W}^C_Q$, $\boldsymbol{k}^C_t = \boldsymbol{x}_t \boldsymbol{W}^C_K$, $\boldsymbol{v}^C_t = \boldsymbol{x}_t \boldsymbol{W}^C_V$

3 # Rank-1 updating step

4 Update $\boldsymbol{U}^t$ as in (12)

5 # Combining step

6 Update $\boldsymbol{Q}^U_t = [\boldsymbol{Q}^U_{t-1}; \boldsymbol{q}^U_t]$

7 Update $\boldsymbol{K}^C_t = [\boldsymbol{K}^C_{t-1}; \boldsymbol{k}^C_t]$

8 Update $\boldsymbol{V}^C_t = [\boldsymbol{V}^C_{t-1}; \boldsymbol{v}^C_t]$

9 Compute $\boldsymbol{s}^C_t = \frac{\boldsymbol{q}^C_t \boldsymbol{K}^{C\top}_t}{\sqrt{d}} \in \mathbb{R}^{1 \times t}$ as in (4)

10 Compute $\boldsymbol{s}^U_t = \frac{\boldsymbol{q}^C_t \boldsymbol{U}^{t\top}}{\sqrt{d}} \in \mathbb{R}^{1 \times t}$ as in (5)

11 Compute $\boldsymbol{p}_t = \text{softmax}\left(\boldsymbol{s}^C_t - \text{SiLU}\left(\boldsymbol{s}^U_t\right)\right) \in \mathbb{R}^{1 \times t}$ as in (6)

12 Compute $\boldsymbol{o}_t = \boldsymbol{p}_t \boldsymbol{V}^C_t \in \mathbb{R}^{1 \times d}$ as in (7)

13 Return $\boldsymbol{o}_t \in \mathbb{R}^{1 \times d}$ and UQ-KV cache $\boldsymbol{U}^t$, $\boldsymbol{Q}^U_t$, $\boldsymbol{K}^C_t$, $\boldsymbol{V}^C_t \in \mathbb{R}^{t \times d}$

---

**UQ-KV cache in CASTLE.** By the analysis in Appendix E.1, UQ-KV cache requires memory

$$4 * B * n_{\text{castle\_head}} * d * 2 \text{ Bytes} = 8 B T n_{\text{castle\_head}} d \text{ Bytes} = \frac{9}{2} B T n_{\text{head}} d \text{ Bytes}.$$

**UQ-KV cache in CASTLE-SWL.** By the analysis in Appendix E.1, UQ-KV cahce in CASTLE requires memory

$$3 * B * n_{\text{castle\_head}} * d * 2 \text{ Bytes} + t \wedge W * B * n_{\text{castle\_head}} * d * 2 \text{ Bytes}.$$

- When $t \leq W$ (window size), UQ-KV cache in CASTLE-SWL uses $\frac{9}{2} B T n_{\text{head}} d$ Bytes.
- When $t \gg W$, UQ-KV cache uses $\approx \frac{27}{8} B T n_{\text{head}} d$.

Denote the memory requirement of KV cache in Baseline-XL by $M_{\text{KV}}$, and UQ-KV cache in CASTLE(-SWL) by $M_{\text{UQ-KV}}$. Then, we have

- $M_{\text{UQ-KV}} \leq \frac{9}{8} M_{\text{KV}}$ for any $t$.
- $M_{\text{UQ-KV}} \approx \frac{27}{32} M_{KV}$ when $t \gg W$ ($W$ is the window size of CASTLE-SWL).

Next, we illustrate the concrete memory requirements of the KV cache (Baseline-XL) and the UQ–KV cache (CASTLE(-SWL)-XL) at different sequence lengths. Table 30 reports the memory usage for a single layer. Notably, CASTLE-SWL-XL already achieves lower memory consumption than Baseline-XL at a sequence length of 1K.

Table 30: Memory usage (in MB) of the KV cache in Baseline-XL and the UQ–KV cache in CASTLE(-SWL)-XL for a single layer at different sequence lengths.

|  | 1K | 4K | 16K |
|---|---|---|---|
| Baseline-XL | 67.11 | 268.44 | 1073.74 |
| CASTLE-SWL-XL | **66.06** | **235.93** | **915.41** |
| CASTLE-XL | 75.50 | 301.99 | 1207.96 |

### E.3 PREFILLING ALGORITHM

In prefilling stage, we only need to compute $U^t$. Due to the smililarity between (2) and standard causal attention, we can implement prefilling by FlashAttention-like kernels. Its pseudocode is in Algorithm 5.

---

**Algorithm 5:** Prefilling Algorithm

---

**Require:** $Q^U, K^U, V^U, Q^C, K^C, V^C \in \mathbb{R}^{L \times d}$

1   # Get UQ-KV Cache

2   Initialize $U = 0^{L \times d}$

3   **for** $k = 0, \ldots, N - 1$ *(in parallel)* **do**

4      Load $Q^U_{T_k,:}$ from HBM to on-chip SRAM

5      **for** $j = k, \ldots, N - 1$ *(sequential)* **do**

6         Load $K^U_{T_j}, V^U_{T_j,:}, U_{T_j,:}$ from HBM to on-chip SRAM

7         On chip, compute $A^U_{T_k,T_j} = \text{sigmoid}\left( \dfrac{Q^U_{T_k,:} K^{U\top}_{T_j,:}}{\sqrt{d}} + M^U_{T_k,T_j} \right)$

8         On chip, $U_{T_k,:} \leftarrow U_{T_k,:} + A^U_{T_k,T_j} V^U_{T_j,:}$

9         Write $U_{T_k,:}$ to HBM

10      **end**

11 **end**

12 Save UQ-KV cache $U, Q^U, K^C, V^C \in \mathbb{R}^{L \times d}$

13 Call Algorithm 1 to get $O$

14 Return $O \in \mathbb{R}^{L \times d}$

---

## F    ADDITIONAL ILLUSTRATIONS IN THE DEFINITION OF CASTLE

In this section, we give additional illustrations of CASTLE and CASTLE-SWL.

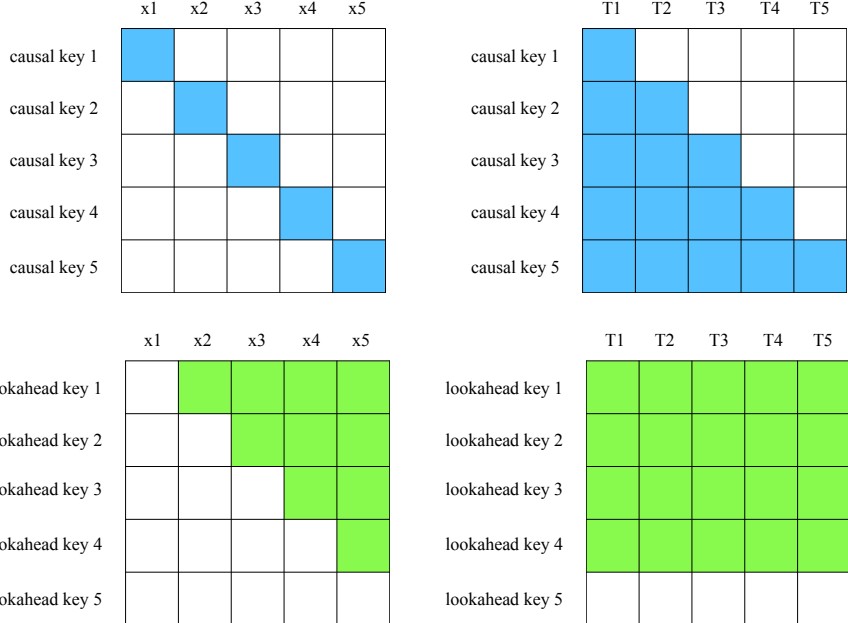

Figure 10: Receptive fields of causal keys and lookahead keys with respect to contextualized representations and tokens (excluding the first layer) when generating the 6th token. Tokens are denoted by $T_i$ and their contextualized representations by $x_i$. Causal key $i$ corresponds to the causal key of token $i$, while lookahead key $i$ corresponds to the lookahead key of token $i$. When generating the $(t + 1)$-th token, for token $s$ ($s < t + 1$), the causal key of token $s$ is a projection of $x_s$. Due to the softmax in attention, except in the first layer, causal keys of $s$ attend over tokens $\{T_1, \cdots, T_s\}$. For token $s < t$, lookahead keys of $s$ incorporate information from $\{x_{s+1}, \cdots, x_t\}$ and attend over all existing tokens $\{T_1, \cdots, T_t\}$. Since the last row of $M^U$ is defined as $[M_t^U]_{t,:} = (-\infty)^{1 \times t}$, lookahead keys of token $t$ are all-zeros vectors and thus have empty receptive fields when generating the $(t + 1)$-th token.

$$\boldsymbol{M}_t^U \in \mathbb{R}^{t \times t}$$

| $-\infty$ | 0 | 0 | 0 | $-\infty$ | $-\infty$ |
|-----------|---|---|---|-----------|-----------|
| $-\infty$ | $-\infty$ | 0 | 0 | 0 | $-\infty$ |
| $-\infty$ | $-\infty$ | $-\infty$ | 0 | 0 | 0 |
| $-\infty$ | $-\infty$ | $-\infty$ | $-\infty$ | 0 | 0 |
| $-\infty$ | $-\infty$ | $-\infty$ | $-\infty$ | $-\infty$ | 0 |
| $-\infty$ | $-\infty$ | $-\infty$ | $-\infty$ | $-\infty$ | $-\infty$ |

Figure 11: CASTLE-SWL differs with CASTLE only in the definition of $\boldsymbol{M}_t^U$. This figure shows an illustration of $\boldsymbol{M}_t^U$ in the lookahead keys of CASTLE-SWL, as defined in (8), when generating the $(t+1)$-th token. Each lookahead key of token $s$ ($s < t$) has access to representations $\{\boldsymbol{x}_k : s+1 \leq k \leq \min\{t,, s+W\}\}$, i.e., up to $W$ subsequent tokens. Also, no information from tokens which are not yet generated is leaked and thereby preserving the autoregressive property. In this figure, we set $t = 6$ and window size $W = 3$. For example, lookahead keys of token 2 can "see" $\{\boldsymbol{x}_3, \boldsymbol{x}_4, \boldsymbol{x}_5\}$, while lookahead keys of token 4 can "see" $\{\boldsymbol{x}_5, \boldsymbol{x}_6\}$. As analyzed in Section 2.3 and 2.4, CASTLE-SWL has the same complexities with CASTLE in both training and inference, however, it can further reduce FLOPs and improves efficiency in practice through the use of sliding windows.

