# OpenReview forum: "Causal Attention with Lookahead Keys"
_ICLR.cc/2026/Conference — Submitted to ICLR 2026_

### Official Review · Reviewer_iUuR · 2025-10-24

**Soundness:** 2
**Presentation:** 2
**Contribution:** 2
**Rating:** 4
**Confidence:** 4

**Summary:**

The paper proposes CASTLE (Causal Attention with Lookahead Keys), a novel attention mechanism for autoregressive language modeling. Unlike standard causal attention—where each token’s key is static and only encodes its preceding context—CASTLE dynamically updates the keys of all previously generated tokens to incorporate information from subsequent tokens (up to the current generation step). These “lookahead keys” are designed to enrich contextual representations while strictly preserving the autoregressive property (i.e., no future token information is leaked).

Although the mechanism appears sequential, the authors derive a mathematically equivalent parallelizable formulation that avoids explicitly materializing lookahead keys at every step, enabling efficient training with O(L²d) complexity. They also propose CASTLE-SWL, a sliding-window variant that further improves practical efficiency. Experiments across model scales (0.16B to 1.3B) show consistent improvements in language modeling (lower perplexity) and downstream task performance compared to standard causal attention.

**Strengths:**

The paper proposes CASTLE (Causal Attention with Lookahead Keys), a novel attention mechanism for autoregressive language modeling. Unlike standard causal attention—where each token’s key is static and only encodes its preceding context—CASTLE dynamically updates the keys of all previously generated tokens to incorporate information from subsequent tokens (up to the current generation step). These “lookahead keys” are designed to enrich contextual representations while strictly preserving the autoregressive property (i.e., no future token information is leaked).

Although the mechanism appears sequential, the authors derive a mathematically equivalent parallelizable formulation that avoids explicitly materializing lookahead keys at every step, enabling efficient training with O(L²d) complexity. They also propose CASTLE-SWL, a sliding-window variant that further improves practical efficiency. Experiments across model scales (0.16B to 1.3B) show consistent improvements in language modeling (lower perplexity) and downstream task performance compared to standard causal attention.

**Weaknesses:**

1. Clear motivation: The paper identifies a well-known limitation of causal attention—its inability to revise early token representations using later context (e.g., in garden-path sentences)—and proposes an elegant solution.
2. Theoretically sound design: The derivation of an equivalent parallel form is non-trivial and crucial for scalability. The use of a hybrid key structure (causal + lookahead) is well-justified.
3. Comprehensive ablation studies: The authors validate key design choices (e.g., necessity of causal keys, role of SiLU, effect of sliding window size) and show that gains are not simply due to increased parameter or key count.
4. Consistent empirical gains: Improvements are observed across model sizes and multiple downstream benchmarks, suggesting the method generalizes well.

**Questions:**

1. Insufficient baseline comparisons: The paper only compares against vanilla causal attention, but numerous recent works have proposed alternative attention mechanisms to address similar limitations (e.g., PaTH Attention, ENTP, Selective Attention, Re-Reading, BeLLM). The absence of comparisons with these methods makes it difficult to assess whether CASTLE’s gains are truly novel or simply reflect a generic benefit of modifying attention. Meanwhile, in Table2, the CASTLE only outperforms baselines by a small margin in S and L model size (only 0.3 points on average). In contrast, the SWL variant shows further improvement. Therefore, I doubt that the superiority of CASTLE-SWL stems mainly from SWL instead of CASTLE. The authors should only compare baselines with SWL (or similar philosophy implementation) as an ablation study to further reflect the effectivenss of CASTLE.
2. Lack of efficiency analysis: Despite claims of “efficient” training and inference, the paper does not report actual training throughput (tokens/sec), FLOPs, memory usage, or inference latency. Given that CASTLE introduces additional projections and a larger cache (UQ-KV cache stores more data), the computational overhead could be substantial. Without this data, it is unclear whether the performance gains come at an unacceptable compute cost.
3. Ambiguous “token-efficiency” claim: The introduction emphasizes improving token efficiency, yet the experiments fix the training budget at 50B tokens and only compare final performance—not convergence speed or sample efficiency. Thus, the token-efficiency claim is not empirically supported.
4. Additional experiments: The paper shows SiLU is crucial for stability, but doesn’t test whether it’s the gating property or just nonlinearity that helps.

---

> ### Author Response · Authors · 2025-11-24
> **Response to Reviewer iUuR (Part 1)**
>
> Thank you very much for your constructive and helpful comments. We address them as follows.
>
> # **W1**
> > Insufficient baseline comparisons: The paper only compares against vanilla causal attention, but numerous recent works have proposed alternative attention mechanisms to address similar limitations (e.g., PaTH Attention, ENTP, Selective Attention, Re-Reading, BeLLM). The absence of comparisons with these methods makes it difficult to assess whether CASTLE’s gains are truly novel or simply reflect a generic benefit of modifying attention.
>
> Thank you for your comments. Below, we provide our comparison results with these methods, or explain why certain methods cannot be compared in our setting.
>
> ## Path
> ### **CASTLE is orthogonal to PaTH and can be combined with PaTH directly**
> PaTH Attention is essentially a positional encoding algorithm. CASTLE is a new attention mechanism. CASTLE and PaTH Attention are not competitors. Actually, CASTLE can be directly combined with PaTH Attention by simply replacing RoPE in causal keys of CASTLE by PaTH.
>
> We mention PaTH in our introduction because in their inference algorithm, they also need to update KV cache dynamically.
>
> ### **Although CASTLE is orthogonal to PaTH and can be combined with PaTH directly, we still compare CASTLE and PaTH Attention here for completeness**
>
> The results are reported in tables below. It is shown that CASTLE and PaTH attention has comparable results while CASTLE outperform PaTH slightly in training and validation loss and average accuracy of downstream evaluation
>
> |Model|n_params|TrainLoss|TrainPPL|EvalLoss|EvalPPL|
> |-----|--------|---------|--------|--------|-------|
> |Baseline-XL|1.310B|2.430|11.360|2.426|11.309|
> |CASTLE-SWL-XL|1.304B|**2.401**|**11.036**|**2.389**|**10.900**|
> |PaTH|1.315B|2.405|11.073|2.396|10.980|
>
> |Model|ARC-C|ARC-E|BoolQ|Hella.|MMLU|OBQA|PIQA|Wino.|Avg.|
> |-----|-----|-----|-----|------|----|----|----|-----|----|
> |Baseline-XL|35.58|65.78|61.07|50.84|**26.71**|36.20|71.27|52.72|50.02|
> |CASTLE-SWL-XL|**38.99**|70.08|**61.74**|52.35|25.85|37.20|72.52|**56.75**|**51.93**|
> |PaTH|36.86|**71.34**|57.95|**55.19**|25.25|**37.40**|**73.07**|54.38|51.43|
>
> ## ENTP
> As mentioned in our introduction, ENTP has $O(L^3 d)$ training costs, which is computationally infeasible for us to evaluate.
>
> ## Selective Attention
> Selective Attention is not open-sourced, and we are unable to reproduce it and conduct a direct comparison with it currently.
>
> ## BeLLM and Re-Reading
> BeLLM is an algorithm to improve sentence embedding tasks. Re-Reading works by duplicating questions to improve the models' reasoning accuracy. Both BeLLM and Re-Reading are irrelevant to pretraining. We mention BeLLM and Re-Reading in our introduction because it is also highlighted in their papers that causal mask, by preventing tokens from accessing future information, can hinder a model’s ability to capture global context and degrade its natural language understanding. This serves as our motivation, while the algorithms BeLLM and Re-Reading cannot be used in pretraining.
>
> # **W2**
> > Meanwhile, in Table2, the CASTLE only outperforms baselines by a small margin in S and L model size (only 0.3 points on average). In contrast, the SWL variant shows further improvement. Therefore, I doubt that the superiority of CASTLE-SWL stems mainly from SWL instead of CASTLE. The authors should only compare baselines with SWL (or similar philosophy implementation) as an ablation study to further reflect the effectiveness of CASTLE.
>
> Thank you for your suggestion. We add ablations in comparison with baseline equipped with sliding windows in Appendix A.5.5. For fair comparison, we use the same size of sliding windows for CASTLE-SWL and Baseline-SW (baseline model with sliding windows).
>
> The following tables report the results. It is observed that both **Baseline-SW-S and Baseline-SW-L experience substantial degradation in training and validation loss. Their downstream task performance is either comparable to or worse than the original baselines.**
>
> These results indicate that applying a sliding window to lookahead keys is fundamentally different from applying it to the causal keys in standard causal attention. And **the advantages of CASTLE-SWL cannot be attributed to the use of a sliding-window mechanism.**
>
> | Model Name      | n_params | Train Loss | Train PPL | Eval Loss | Eval PPL |
> |-----------------|----------|------------|-----------|-----------|----------|
> | Baseline-S | 160M | 2.795| 16.364| 2.798 | 16.411 |
> | Baseline-SW-S | 160M | 2.861 | 17.487 | 2.868 | 17.608 |
> | CASTLE-SWL-S    | 160M | **2.786**| **16.213** | **2.790**| **16.273**|
>
> | Model Name      | n_params | Train Loss | Train PPL | Eval Loss | Eval PPL |
> |-----------------|----------|------------|-----------|-----------|----------|
> | Baseline-L| 756M | 2.513 | 12.346 | 2.507 | 12.269|
> | Baseline-SW-L| 756M| 2.526 | 12.503| 2.524 | 12.480|
> | CASTLE-SWL-L| 753M|**2.476**|**11.890** |**2.471**|**11.832** |

---

> ### Author Response · Authors · 2025-11-24
> **Response to Reviewer iUuR (Part 2)**
>
> (continuation of Part 1)
>
> | |ARC-C|ARC-E|BoolQ|Hella.|MMLU|OBQA|PIQA|Wino. |Avg.|
> |------------------|-------|-------|-------|--------|-------|------|------|-------|-------|
> | Baseline-S| 25.68 | 54.97 |**56.09**| 33.81  | 25.54 | 28.20 | 63.98 | 52.57 | 42.60 |
> | Baseline-SW-S| 26.28 | 53.91 | 54.62 | **35.51** | **25.83** | 29.00 | 62.73 | **53.20** | 42.63 |
> | CASTLE-SWL-S| **27.39** | **56.19** |**56.09**| 35.46 | 25.34 | **30.20** | **64.85** | 51.22 | **43.34** |
>
> | |ARC-C|ARC-E|BoolQ|Hella.|MMLU|OBQA|PIQA|Wino. |Avg.|
> |------------------|-------|-------|-------|--------|-------|------|------|-------|-------|
> | Baseline-L  | 33.36 | 63.64 | 59.24 | 46.16 | **26.82** | 33.40 | **69.53** | 54.06 | 48.28 |
> | Baseline-SW-L| 32.94 |65.36| 59.24 | 46.05 | 24.86 | 32.40 | 69.31 |**54.14**| 48.04 |
> | CASTLE-SWL-L | **36.26** |**65.53**| **60.58** | **48.55** | 24.70 | **34.80** | 69.10 |53.67| **49.15** |
>
> # **W3**
> > Lack of efficiency analysis: Despite claims of “efficient” training and inference, the paper does not report actual training throughput (tokens/sec), FLOPs, memory usage, or inference latency. Given that CASTLE introduces additional projections and a larger cache (UQ-KV cache stores more data), the computational overhead could be substantial. Without this data, it is unclear whether the performance gains come at an unacceptable compute cost.
>
> Thank you very much for your comments. Please see the Efficiency section of the General Response.
>
> Regarding ``Given that CASTLE introduces additional projections and a larger cache (UQ-KV cache stores more data)'', we would like to clarify that the **UQ-KV cache of CASTLE-SWL can actually require less memory than the KV cache in standard causal attention.**
>
> ### **UQ-KV cache memory requirement**
> We show in Appendix E.2 of the revision, CASTLE-SWL's UQ-KV cache actually requires less memory than the KV cache in standard causal attention. And CASTLE requires at most 9/8 memory of KV cache in standard causal attention.
> We give detailed analysis in Appendix E.2 of the revision. Here, we provide an intuitive explanation.
>
> Although for each head, standard causal attention only stores 2 tensors K and V, CASTLE(-SWL) requires 4 tensors U, $Q^U$, $K^C$ and $V^C$. However, to align the number of parameters, we cut the number of heads in CASTLE almost by half as compared to the baseline MHA model.
>
> In addition, per our discussions in Appendix E.1, in CASTLE-SWL, we do not need to store the full $Q^U$, actually, we only need to store the length of window size in $Q^U$. Thus, CASTLE-SWL can actually save cache memory.
>
> We report the exact KV cache / UQ-KV cache memory usage (in MB) below. The results show that the UQ-KV cache in CASTLE-SWL requires less memory than the baseline model even at a sequence length of 1K, and saves more when the sequence length increases.
>
> | |1K|4K|16K|
> |-------------------|-------|--------|---------|
> |Baseline-XL|67.11| 268.44|1073.74|
> |CASTLE-SWL-XL|**66.06**|**235.93**|**915.41**|
> |CASTLE-XL |75.50|301.99|1207.96|
>
> # **W4**
> > Ambiguous “token-efficiency” claim: The introduction emphasizes improving token efficiency, yet the experiments fix the training budget at 50B tokens and only compare final performance—not convergence speed or sample efficiency. Thus, the token-efficiency claim is not empirically supported
>
> Thank you for your comments.
>
> We train XL-size models on 5B, 10B, 20B, 30B and 50B and report the results in Appendix A.5.7. We attach part of results here for convenience.
>
> The training loss is reported in the following table. Validation loss can also be found in Appendix A.5.7 following the same trend with training loss.
>
> | | 5B| 10B| 20B| 30B| 50B|
> |-----------------|-------|-------|-------|-------|-------|
> | Baseline-XL  |2.798|2.650| 2.544 | 2.485 | 2.430 |
> | CASTLE-SWL-XL|**2.733**|**2.599**|2.497|**2.449**|**2.401**|
> | CASTLE | 2.736 | 2.602 |**2.491**| 2.450|**2.401**|
>
> Scaling-plot can be found in Figure 5 of the revision.
>
> It is shown that both CASTLE and CASTLE-SWL consistently outperform the Baseline-XL model across all training-token budgets from 5B to 50B tokens. The scaling trends appear approximately linear when plotted in log-log space (Figure 5 in Appendix A.5.7 of the revision). The token-scaling experiments validate that
> CASTLE and CASTLE-SWL achieves substantially better scaling properties than the baseline, indicating significantly superior sample efficiency.
>
> # **W5**
> > Additional experiments: The paper shows SiLU is crucial for stability, but doesn’t test whether it’s the gating property or just nonlinearity that helps
>
> Thank you very much for your comments. Please see the section “How We Come Up With SiLU in Eq. (6)” in the General Response, where we report additional experiments comparing linear functions, different nonlinear functions. Ablation studies show the superiority of SiLU, while other choices also outperform the baseline

---

> > ### Comment · Reviewer_iUuR · 2025-11-24
> >
> > I appreciate the detailed illustration and revised manuscript provided by the authors. However, some of my concerns still remain:
> > 1. If my understanding is correct, in the global response, your method is worse than the baseline in terms of training and inference efficiency. Meanwhile, this inferiority is more severe for the SWL variant. This makes me worry that the improvement of CASTLE-SWL is a tradeoff between efficiency and efficacy. This seems to contradict the authors' claim of "efficient parallel training.". If possible, try to deactivate the flash-attention implementation of the baseline and present the comparison under fair conditions.
> > 2. Line1006 and 1007, the two $p_t$ are equivalent. Why do the authors claim that they are not equivalent?
> > 3. The explanation of using SiLU seems too casual. It looks like the authors try many possible solutions and pick the only one that improves a lot.
> >
> > Other concerns are eliminated, the authors' responses demonstrate that their method
> > 1. is memory-wise efficient
> > 2. outperforms Path baseline on XL-size model
> >
> > If the above three concerns could be eliminated, I would raise my ratings.

---

> > > ### Author Response · Authors · 2025-11-25
> > > **Response to Reviewer iUuR**
> > >
> > > Thank you for your response. We address them as follows.
> > >
> > > 1. > If my understanding is correct, in the global response, your method is worse than the baseline in terms of training and inference efficiency. Meanwhile, this inferiority is more severe for the SWL variant. This makes me worry that the improvement of CASTLE-SWL is a tradeoff between efficiency and efficacy. If possible, try to deactivate the flash-attention implementation of the baseline and present the comparison under fair conditions
> > >
> > > We believe there is a misunderstanding here. While the first table reports the inference time (the lower the better), the second table reports the training throughput (the higher the better). So both tables indicate that in all cases, CASTLE-SWL is more efficient than CASTLE.
> > >
> > > 2. > Line1006 and 1007, the two  are equivalent. Why do the authors claim that they are not equivalent?
> > >
> > > We apologize for this typo. We have fixed our typo in the updated revision as ``$p_t = \mathrm{softmax}(s^C_t - \mathrm{SiLU}(s^U_t))$ and $p_t = \mathrm{softmax}(s^C_t + \mathrm{SiLU}(s^U_t))$ are not equivalent''.
> > >
> > > 3. > The explanation of using SiLU seems too casual. It looks like the authors try many possible solutions and pick the only one that improves a lot.
> > >
> > > While we agree that SiLU may not be the optimal design choice, we would like to emphasize that it is not an arbitrary one. As explained in our general response, this choice is justified as follows:
> > >
> > > (1)  **Attempt I, II and III are the most standard and natural choices to solve unstable problems in LLM training.** More specifically, when we find instability when using $p_t = \mathrm{softmax}(s^C_t - s^U_t)$ directly, natural choices are
> > >
> > > - scaling $s^U_t$ down by some factor (which corresponds to Attempt I)
> > >
> > > - clipping $s^U_t$. Here, clipping on both sides corresponds to Attempt II. Clipping on one side corresponds to Attempt III.
> > >
> > > These are common go-to approaches to stabilize specific terms in LLM training. **Therefore, these three variants are not casual explorations but rather rational and canonical choices for stabilized training.**
> > >
> > > (2)  **Different choices for Equation(6) all outperform baseline. This demonstrates that the lookahead mechanism itself is effective.**
> > >
> > > - Attempt I reduces the validation loss by ~0.01, and improves downstream accuracy from 50.02 → 50.59
> > >
> > > - Attempt II improves downstream accuracy from 45.40 → 46.69
> > >
> > > Despite their different mathematical forms, all three approaches lead to nontrivial gains. This strongly indicates that the lookahead mechanism itself is responsible for the improvements.
> > >
> > > (3) **Why do we prefer SiLU over Attempt I (scale $s^U_t$ down by some factor)?**
> > >
> > > While Attempt I is successful and promising, it has one extra hyperparameter (i.e., scaling factor C). Thus, we choose SiLU over Attempt I.
> > >
> > > (4) **Why do we prefer SiLU over Attempt II (tanh)?**
> > >
> > > Once we confirmed that SiLU stabilizes training, applying symmetric clipping (tanh) is unnecessary. This is because clipping on both sides (tanh) may overly constrain model's expressivity.
> > >
> > > Finally, we would like to emphasize that although SiLU may not be the optimal choice, determining an "optimal" design for attention mechanisms is inherently difficult. Unlike in machine learning theory or theoretical computer science, architectural design is often guided by empirical intuition (even the Transformer itself is largely heuristic). For the purpose of incorporating lookahead keys, SiLU is a reasonable and sufficiently effective choice.
> > >
> > > If the reviewer has alternative design suggestions, we would be happy to explore it during the reminder of the rebuttal period or as part of future work.

---

> > > > ### Comment · Reviewer_iUuR · 2025-11-27
> > > >
> > > > Thanks for the authors' detailed responses. I apologize for my misunderstandings. I would like to emphasize that my concern mainly lies in the efficiency inferiority.
> > > >
> > > > Could you please show that your method is faster than the vanilla method if removing flash-attention?
> > > > Or, could you please show how to design a CUDA operator like flashattention, which could speed up your module at a device level?
> > > > These are just small points and I believe that if the authors could provide these comparisons, the manuscript could be more solid.
> > > >
> > > > Anyway, I decided to raise my score to 6. The authors' responses on the SiLU is solid enough and should be highlighted in the main text.

---

> > > > > ### Author Response · Authors · 2025-12-03
> > > > > **Response to Reviewer iUuR**
> > > > >
> > > > > Thank you for raising the score.
> > > > >
> > > > > > Could you please show that your method is faster than the vanilla method if removing flash-attention? Or, could you please show how to design a CUDA operator like flashattention, which could speed up your module at a device level? These are just small points and I believe that if the authors could provide these comparisons, the manuscript could be more solid.
> > > > >
> > > > > We would like to clarify that CASTLE and CASTLE-SWL adopt FlashAttention-style block processing and we are not aware of efficient pure PyTorch implementations of CASTLE and CASTLE-SWL which reduces computational complexities to $O(L^2 d)$. Consequently, a fair comparison using only PyTorch may not be feasible.
> > > > > For CUDA kernels, Algorithm 1 and Algorithm 3 describe exactly the work done by each CTA. Since CASTLE and CASTLE-SWL are FlashAttention-style, their CUDA kernels will largely mirror FlashAttention, with matrix multiplication operations replaced by their CASTLE (or CASTLE-SWL) counterparts. We also demonstrate that CUDA kernel speedup depends on low-level implementation details such as memory layout, warp specialization, and asynchronous pipelines. Achieving optimal performance requires step-by-step tuning, which we leave for future work.
> > > > >
> > > > > > Anyway, I decided to raise my score to 6. The authors' responses on the SiLU is solid enough and should be highlighted in the main text.
> > > > >
> > > > > Thank you for your comment. We added ``We give a comprehensive study of different variants of (6) and why we use SiLU in Appendix A.5.3'' in the main text.

---

### Official Review · Reviewer_T64n · 2025-10-31

**Soundness:** 3
**Presentation:** 3
**Contribution:** 2
**Rating:** 4
**Confidence:** 3

**Summary:**

The paper proposes a novel attention mechanism for causal language models, where key vectors are allowed to attend not only to tokens in preceding positions but also to those in subsequent positions. Since the lookahead operation is restricted within the generated sequence, the authors argue that their model preserves the autoregressive property of traditional LMs. The paper offers extensive theoretical analysis and a detailed discussion of parallel training strategies. Empirical evaluations on training loss and several downstream tasks demonstrate the method’s potential effectiveness.

Overall, this is an interesting work with solid theoretical grounding and promising empirical results. However, I have two major concerns:

(1) KV Cache. Although the paper introduces the so-called UQ-KV cache, I am not convinced that such a cache remains feasible when lookahead keys are involved. If my understanding is correct, the keys must be updated token by token, making KV caching essentially inapplicable during inference. A possible workaround might be the use of a sliding-window mechanism, but even within a fixed window, caching would still be difficult to maintain efficiently. This raises concerns about the inference efficiency of the proposed method.

(2) Efficiency Discussion. Building on the above concern, I expected to see a detailed empirical comparison between the proposed attention and standard causal attention in terms of inference efficiency—e.g., FLOPs, throughput, or per-token decoding latency. Unfortunately, such results are absent. Without this analysis, the experimental section cannot fully convince me that the proposed method is practically useful.

**Strengths:**

an new attention mechanism for causal LMs
theoretical analysis for the proposed mechanism

**Weaknesses:**

see my comments in Summary

**Questions:**

see my comments in Summary

---

> ### Author Response · Authors · 2025-11-24
> **Response to Reviewer T64n**
>
> Thank you very much for your insightful comments. We address them as follows.
>
> # **Q1**
> > KV Cache. Although the paper introduces the so-called UQ-KV cache, I am not convinced that such a cache remains feasible when lookahead keys are involved. If my understanding is correct, the keys must be updated token by token, making KV caching essentially inapplicable during inference. A possible workaround might be the use of a sliding-window mechanism, but even within a fixed window, caching would still be difficult to maintain efficiently. This raises concerns about the inference efficiency of the proposed method.
>
> Thank you very much for your comments. Our UQ-KV cache is not static like traditional KV cache, but can still reduce I/O and computational costs during inference to $O(td)$ when generating the $(t+1)$-th token by avoiding recomputing previous token representations. Detailed discussions are in Appendix E of the revision. We briefly explain these here for convenience.
>
> During inference, we have 2 steps as described in Section 2.4.
>
> (1) Rank-1 update to U cache. **We proved in Appendix E that we do not need to compute $U^t$ from scratch during inference, actually, $U^t$ can be obtained from $U^{t-1}$ by a rank-1 update as in Equation (12).**
>
> (2) Compute attention scores and outputs.
>
> Step (2) is similar to the inference of standard causal attention. Step (1) corresponds to the dynamic update of lookahead keys (rank-1 update to the U-cache during inference).
>
> **Although we need to update U cache during inference, the rank-1 update only has $O(t d)$ I/O and $O(t d)$ computational costs when generating the $(t+1)$-th token.**
> Even without sliding window, rank-1 update still only needs $O(t d)$ I/O and $O(t d)$ computation. Inference of standard causal attention also requires $O(t d)$ I/O and $O(t d)$ computation when generating the $(t+1)$-th token.
>
> In addition, we wrote fused kernel for Step (1) and Step (2) so that all data is only moved from DRAM to SRAM once, and the updated U cache is only moved from SRAM back to DRAM once. This leads to comparable data movement with standard causal attention.
>
> # **Q2**
> > 2. Efficiency Discussion. Building on the above concern, I expected to see a detailed empirical comparison between the proposed attention and standard causal attention in terms of inference efficiency—e.g., FLOPs, throughput, or per-token decoding latency. Unfortunately, such results are absent. Without this analysis, the experimental section cannot fully convince me that the proposed method is practically useful.
>
> Thank you very much for your comments. Please see the Efficiency section in the General Response

---

### Official Review · Reviewer_TsVF · 2025-11-01

**Soundness:** 3
**Presentation:** 3
**Contribution:** 3
**Rating:** 4
**Confidence:** 3

**Summary:**

This paper proposes an optimization for traditional causal attention: lookahead keys, which allow the keys to utilize information from tokens that appear after their own positions.

**Strengths:**

1. The use of lookahead keys without breaking the autoregressive property appears to be well-motivated.
2. Performance improvements are achieved in both perplexity and downstream experiments.
3. The paper provides detailed and clear descriptions of efficient training and inference algorithms.

**Weaknesses:**

1. Although the experimental results in Table 1 and Table 2 demonstrate the effectiveness of CASTLE, the paper does not sufficiently explain (1) what representational problem the lookahead keys actually solve—perhaps a visual explanation is needed; and (2) the comparison of FLOPs between training and inference. A moderate increase in FLOPs is acceptable, but excessive computational cost would be prohibitive.
2. The analysis experiments are lacking in depth and detail.

**Questions:**

1. The definition and motivation for using SiLU(x) are not clearly stated, which makes the computation in Equation (6) seem rather odd.

2. Regarding training complexity: in traditional causal attention, since each token’s representation remains consistent across different time steps, one can compute representations for all tokens in parallel for an input of length L, thereby significantly reducing complexity. In the proposed method, however, each token’s representation varies across time steps, making parallel computation of different time steps impossible. Is Theorem 1 intended to address this issue? The authors should clearly explain Theorem 1. Points I do not fully understand include: (a) if SiLU is removed from Equation (11), then (11) essentially becomes attention with a causal mask—how can that capture changes in representations across different time steps? (b) Please directly compare the forward and backward propagation complexity for a sequence of length L in traditional causal attention versus CASTLE.

If the authors can satisfactorily answer the above questions, I would be willing to raise the score.

---

> ### Author Response · Authors · 2025-11-24
> **Response to Reviewer TsVF (Part 1)**
>
> Thank you very much for your valuable comments and suggestions. We address them as follows.
>
> # **Q1**
> > The definition and motivation for using SiLU(x) are not clearly stated, which makes the computation in Equation (6) seem rather odd.
>
> Thank you very much for raising up this question. Please see the ``How We Come Up With SiLU in Eq(6)'' section of the General Response
>
> # **Q2**
> > Regarding training complexity: in traditional causal attention, since each token’s representation remains consistent across different time steps, one can compute representations for all tokens in parallel for an input of length L, thereby significantly reducing complexity. In the proposed method, however, each token’s representation varies across time steps, making parallel computation of different time steps impossible. Is Theorem 1 intended to address this issue? The authors should clearly explain Theorem 1. Points I do not fully understand include: (a) if SiLU is removed from Equation (11), then (11) essentially becomes attention with a causal mask—how can that capture changes in representations across different time steps? (b) Please directly compare the forward and backward propagation complexity for a sequence of length L in traditional causal attention versus CASTLE
>
> Thank you for your comments. We clarify the structure of Section 2.3 as follows.
> ## **What is the role of Theorem 1?**
> Before Section 2.3, we first introduce the formal definition of CASTLE in its recurrent form. We write the recurrent definition first in Section 2.2 because it materializes each lookahead key specifically and is thus most easy to understand.
>
> Since it is impossible to have parallel training if we materialize each lookahead key at each position, the recurrent form cannot be used in training directly.
> So first, in Section 2.3, we derive a matrix form of CASTLE which is equivalent to its recurrent definition. **This matrix form generates outputs exactly the same as the concatenation of the outputs from the recurrent form at different positions, and based on this matrix formulation, it is easier for us to design parallel training algorithms.**
>
> More specifically, in the matrix formulation of Theorem 1, we denote the output when generating the $(t+1)$-th token by $o^t$, and we concatenate $o^t$'s into matrix $O^L = (o_1, \dots, o_L)$, where $L$ is the sequence length in training. And we also concatenate the inputs together into matrix $X^L = (x_1, \dots, x_L)$.
> Then, in Theorem 1, we derive a matrix form, i.e., we find a function F, which mainly consists of some matrix multiplications such that $O^L = F(X^L)$.
> Theorem 1 provides an equivalent definition of CASTLE(-SWL), which has two benefits:
> (1) we eliminate the need to materialize individual lookahead keys
> (2) Instead of generating outputs one by one as in the recurrent definition of Section 2.2, we can compute all outputs $O^L$ in a single expression.
>
> **These two benefits make it easier for us to design efficient parallel algorithms in Theorem 2. Theorem 1 can be regarded as a bridge between the recurrent definition in Section 2.2 and the efficient parallel algorithm in Theorem 2.**
>
> The formal proof, which shows that the LHS of Equation (11) is exactly the concatenation of the outputs we defined in recurrent form, is provided in Appendix B.
>
> ## **What is the role of Theorem 2?**
> Although in Theorem 1, we can write the outputs from different positions in one formula which removes the need to materialize each lookahead key specifically, Equation (11) itself still has $O(L^3)$ complexity because there is a matrix multiplication between $n \times n$ matrices.
> Fortunately, the matrix $(Q^C (V^U)^T) \odot \tilde{M}^C$ is **a low-rank matrix multiplied by a mask**. Theorem 2 presents an algorithm which **utilizes this structure to compute the LHS of Equation (11) efficiently.** The resulting algorithm can generate the same outputs with concatenation of outputs in the recurrent definition, and has only $O(L^2 d)$ complexity.
>
> ## **For any autoregressive attention mechanism, attention score matrix must be a lower triangular matrix**
> For any attention mechanism which has the form $O = softmax(A)V$, $A$ must be a lower triangular matrix. This is because, if $A_{ij} > - \infty$ with $i < j$, then, $o_i$ which is the $i$-th row of $O$ will have information from $v_j$. As $i < j$, this causes information leakage. Thus, any attention score matrix $A$ should be a lower triangular matrix if the corresponding attention mechanism is autoregressive.

---

> ### Author Response · Authors · 2025-11-24
> **Response to Reviewer TsVF (Part 2)**
>
> (continuation of Part 1)
>
> ## **How can the lower triangular matrix $S^U$ in Equation(10) capture changes in representations across different time steps?**
>
> We give a vivid and intuitive explanation here. A rigorous proof is in Appendix B. Recall that the formal definition of lookahead keys in Equation (2),
> $$U^t=\mathrm{sigmoid}(\frac{Q^U_t (K^U_t)^T}{\sqrt{d}} + M^U_t) V^U_t.$$
> Here, the mask $M^U_t$ makes the matrix $\mathrm{sigmoid}(\frac{Q^U_t (K^U_t)^T}{\sqrt{d}}+M^U_t)$ an upper triangular matrix. Then, the $i$-th row of $U^t$ denoted by $u^t_i$ naturally has information of $x_{i+1}, ..., x_t$. We also remark that as we have mentioned in Section 2.2, $u^t_i$ is the lookahead key of token $i$ when generating the $(t+1)$-th token.
>
> Then, the lookahead-key attention scores $s^U_t$ are generated by $q_t (U^t)^T$ as in Equation(5), i.e.,
> $$s^U_t=q_t (V^U_t)^T\mathrm{sigmoid}(\frac{Q^U_t (K^U_t)^T}{\sqrt{d}}+M^U_t)^T\in\mathbb{R}^{1\times t}.$$
>
> The $i$-th entry of $s^U_t$, denoted by $s^U_t(i)$, equals $q_t(u^t_i)^T$. As we have mentioned that $u^t_i$ has information of $x_{i+1}, ..., x_t$, then $s^U_t(i)$ has information of $x_{i+1}, ..., x_t$ which comes from $u^t_i$.
>
> Each $s^U_t$ is a t-dimensional vectors. We append 0's to its end to make it a L-dimensional vector. Then, we concatenate all these vectors for each $t\in [L]$ together into one matrix. In Equation(20) of Appendix B, we rigorously prove that this concatenated matrix exactly equals the $S^U$ defined in Equation(10). Thus, the $(t, i)$-entry ($t \geq i$) $S^U_{t,i}$ equals $s^U_t(i)$ which is the $i$-th entry of $s^U_t$.
>
> As we have mentioned, $s^U_t(i)$ has access to information of $x_{i+1}, ..., x_t$ which comes from $u^t_i$, i.e., for any $t\geq i$, the lower triangular entry $S^U_{t,i}$ has access to information of $x_{i+1}, ..., x_t$  which comes from $u^t_i$ ($u^t_i$ is the lookahead key of token $i$ when generating the $(t+1)$-th token). This is exactly our lookahead mechanism.
>
> A rigorous proof can be found in Appendix B
>
> # **Both forward pass and backward pass have $O(L^2 d)$ computational complexity and $O(L d)$ space complexity**
> Detailed proof is in Appendix D. We briefly explain the reason here for convenience.
>
> We divide the sequence of length L into blocks {$T_0$, ..., $T_{N-1}$}. Each block $T_i$ is of size B. We assume $L$ is divisible by $B$ for simplicity.
>
> For both forward pass (Algorithm 1) and backward pass (Algorithm 3), we launch one kernel instance for each block $S_{T_i, T_j}$ with $i \geq j$. Thus, there are
> $$
>    \frac{L}{B} \cdot \left(\frac{L}{B} - 1\right) = O\left(\frac{L^2}{B^2}\right) \text{\quad block.}
> $$
> Inside each block, as in Algorithm 1 (forward pass) and Algorithm 3 (backward pass), we only have $O(1)$ matrix multiplications between $B$-by-$d$ matrices or $B$-by-$B$ matrices, where $B$ is the block size. There are also O(1) entrywise operations on each entry of the $B$-by-$B$ block. Algorithm 2 is called by Algorithm 1, it also only contains matrix multiplications between $B$-by-$d$ matrices or $B$-by-$B$ matrices and O(1) entrywise operations on each entry  of the corresponding $B$-by-$B$ block.
>
> In addition, we choose $B = O(d)$. In practice, we find that $B = 64$ optimizes our training kernels' performance. Thus, each block has only $O(B^2 d + B^3 + B^2) = O(B^2 d)$ FLOPs.
>
> Thus, the computational complexity is
> $$
>     O\left(\frac{L^2}{B^2}\right) \cdot O(B^2 d) = L^2  d.
> $$
> The space complexity $O(L d)$ is straightforward.
> The detailed analysis is in Appendix D
>
> # **W1**
> > The paper does not sufficiently explain the comparison of FLOPs between training and inference. A moderate increase in FLOPs is acceptable, but excessive computational cost would be prohibitive
>
> Thank you very much for your comments. Please see the Efficiency section of the General Response

---

### Official Review · Reviewer_aBKr · 2025-11-01

**Soundness:** 3
**Presentation:** 3
**Contribution:** 3
**Rating:** 6
**Confidence:** 4

**Summary:**

This paper introduces CASTLE, a novel attention mechanism that dynamically updates the keys of past tokens as the sequence grows. Recognizing that a naive implementation would be computationally intractable ($O(L^3d)$), the authors derive a mathematically equivalent parallel formulation that reduces training complexity to $O(L^2d)$, matching standard attention. Experiments on models ranging from 160M to 1.3B parameters show consistent improvements in perplexity and downstream tasks. A sliding-window variant, CASTLE-SWL, is also proposed for additional efficiency.

**Strengths:**

- Well-motivated solution addressing the limitation of static keys in causal attention with an intuitive and elegant mechanism.
- Non-trivial technical achievement: the parallel training algorithm successfully reduces complexity from $O(L^3)$ to $O(L^2)$ by proving a mathematical equivalence (Theorem 1) and leveraging properties of masked low-rank matrices (Theorem 2); and with practical inference via "UQ-KV cache" maintaining linear complexity ($O(td)$).
- The authors provide a strong set of ablation studies in the appendix that address several key design choices and trade-offs, strengthening the paper's conclusions.

**Weaknesses:**

1. Lacks **direct evidence** of the mechanism's benefit through targeted tasks (e.g., garden-path sentences or long-range dependencies), beyond general improvements in PPL and some downstream tasks.
2. Limited to 2k context length. Given the importance of long-context capabilities in modern LLMs, scalability to longer sequences and compatibility with existing RoPE-based **length extrapolation** techniques (e.g., YaRN) is a crucial unanswered question, and it remains unclear.
3. **CASTLE-SWL results are counterintuitive**: the smallest window (128) performs best (Table 11), yet limiting lookahead context seemingly contradicts the motivation for handling long-range forward dependencies ("question appears at the end of the input", line 42). The argument about noise (line 268) is unconvincing.
4. Furthermore, the ablation study on window size feels incomplete. No exploration of windows smaller than 128, leaving the performance peak unidentified (Given that window size 0 falls back to the standard attention baseline).
5. No **empirical efficiency metrics**. Including metrics such as end-to-end training wall time, throughput (FLOPs/s), and inference speed (tokens/s) against a standard attention baseline would provide evidence of the algorithm's practical efficiency, despite strong theoretical complexity ($O(L^2d)$).
6. Also, CASTLE-SWL efficiency claims ("can reduce FLOPs", line 269) lack empirical support.
7. Ablations use only perplexity; **downstream task metrics** would provide better insight into design choices, considering that downstream evaluation is relatively inexpensive compared to pre-training.
8. Figure 1 is uninformative; architectural diagrams (Figures 7, 9) should be moved to the main body.

**Questions:**

Have **simpler** alternatives for generating "lookahead keys" been explored? For example: (1) **Reusing causal keys**. Despite the intuition that "the way a model learns to encode information into keys of token s from past tokens may differ from how it encodes information from subsequent tokens" (line 184), an empirical ablation would be valuable; (2) **A direct linear map** $U=QW_U$, which mirrors the causal key path instead of employing a full attention mechanism, would be simpler to implement and avoid the $O(L^3)$ issue entirely, I believe. Ablations on the architectural design itself, rather than just hyperparameters, would robustly justify the added complexity.

---

> ### Author Response · Authors · 2025-11-24
> **Response to Reviewer aBKr (Part 1)**
>
> Thank you very much for your valuable feedback and suggestions. We address them as follows.
>
> # W1
> > Limited to 2k context length. Given the importance of long-context capabilities in modern LLMs, scalability to longer sequences and compatibility with existing RoPE-based length extrapolation techniques (e.g., YaRN) is a crucial unanswered question, and it remains unclear
>
> We thank the reviewer for highlighting this point. To address this, we evaluated both the baseline models and CASTLE-SWL on retrieval tasks (needle-in-a-haystack, or NIAH for short), and we report these results in Appendix A.3. Furthermore, we applied YaRN to both models to examine their behavior under RoPE-based length extrapolation; the corresponding results are provided in Appendix A.4. For convenience, we also include the key results here:
>
> ## **Needle-in-a-haystack (NIAH) without finetuning or YaRN**
> The following table reports the accuracy on the RULER S-NIAH tasks across different sequence lengths without finetuning or YaRN.  It can be seen that CASTLE-SWL-XL demonstrates consistently stronger retrieval capabilities than Baseline-X. In particular, on the S-NIAH-1 task, CASTLE-SWL-XL achieves nonzero scores at the 4K sequence length
>
> || S-NIAH-1 1K | S-NIAH-1 2K | S-NIAH-1 4K | S-NIAH-2 1K | S-NIAH-2 2K | S-NIAH-2 4K | S-NIAH-3 1K | S-NIAH-3 2K | S-NIAH-3 4K |
> |-----------------|-------------|-------------|-------------|-------------|-------------|-------------|-------------|-------------|-------------|
> | Baseline-XL |97.0| 26.6| 0.0| 97.2| 54.2| 0.0| 85.8 | 73.2| 0.0|
> | CASTLE-SWL-XL| **99.4** | **85.4** | **24.6**| **99.2** | **86.0**|2.2| **89.4**|**84.2** | 0.2|
>
> ## **Needle-in-a-haystack with YaRN**
> The following table reports the accuracy on the RULER S-NIAH tasks under YaRN-based length extrapolation. Across all tasks, CASTLE-SWL-XL achieves substantially higher accuracy than Baseline-XL, indicating that CASTLE-SWL has better compatibility with YaRN than the baseline MHA model
>
> | | S-NIAH-1 4K | S-NIAH-1 8K | S-NIAH-2 4K | S-NIAH-2 8K | S-NIAH-3 4K | S-NIAH-3 8K |
> |-----------------|-------------|-------------|-------------|-------------|-------------|-------------|
> | Baseline-XL |11.0| 0.0 |47.8|1.8| 8.6|0.0|
> | CASTLE-SWL-XL | **63.4**| **39.0**| **68.4**|**35.0**|**30.2**|1.4 |
>
> More details and discussions can be found in Appendix A.3 and A.4
>
> # **W2**
> > CASTLE-SWL results are counterintuitive: the smallest window (128) performs best (Table 11), yet limiting lookahead context seemingly contradicts the motivation for handling long-range forward dependencies ("question appears at the end of the input", line 42). The argument about noise (line 268) is unconvincing
>
> Thank you for raising this question. We will clarify it as follows. In our experiments in Appendix A.5.4, **only small model is optimized with window size 128, while medium, large, XL models all prefer window size 512**
>
> ### **Why small models perform best at window size 128?**
> (1) This reason is that the benefit of lookahead keys lies in helping models capture global dependencies, but small models are capacity-limited and can primarily extract local features, making global relations less useful at this scale. This coincides with the fact that larger models (medium, large, XL) having better capabilities all prefer larger window size.
>
> (2) In addition, the models are trained on FineWeb-Edu based on the nanoGPT codebase, where each training sample is randomly drawn from the dataset. As a result, a sample may begin in the middle of one sentence, span multiple unrelated texts, and end in the middle of another. Under such conditions, using excessively large lookahead-key window sizes can introduce additional noise rather than useful context. Therefore, selecting the window size involves balancing the information gained against the noise introduced
>
> # **W3**
> > Furthermore, the ablation study on window size feels incomplete. No exploration of windows smaller than 128, leaving the performance peak unidentified (Given that window size 0 falls back to the standard attention baseline)
>
> Thank you for your comment. We have added supplementary experiments for window size of 64 in Appendix A.5.4 (Tables 17-20 in the revision).  We attach the table for XL models including newly added results for window size 64 here for convenience. Tables for small, medium, large as well as XL models can be found in Appendix A.5.4. We also include downstream evaluation comparison for XL models in Appendix A.5.4
>
> It can be seen that, for the medium, large, and XL models, using a window size of 64 results in **substantial performance degradation**. In both training and validation metrics, CASTLE-SWL64 performs at least 0.01 worse than its counterpart CASTLE-SWL512. Moreover, in downstream evaluation, CASTLE-SWL64 shows a clear disadvantage not only compared to CASTLE-SWL512, but even relative to the baseline model

---

> ### Author Response · Authors · 2025-11-24
> **Response to Reviewer aBKr (Part 2)**
>
> (continuation of Part 1)
>
> |Model|n_params|TrainLoss|TrainPPL|EvalLoss|EvalPPL|
> |-----|--------|---------|--------|--------|-------|
> |Baseline-XL|1.310B|2.548|12.779|2.543|12.723|
> |CASTLE-SWL64-XL|1.304B|2.530|12.548|2.525|12.491|
> |CASTLE-SWL512-XL|1.304B|**2.506**|**12.255**|**2.503**|**12.217**|
>
> |Model|ARC-C|ARC-E|BoolQ|Hella.|MMLU|OBQA|PIQA|Wino.|Avg.|
> |-----|-----|-----|-----|------|----|----|----|-----|----|
> |Baseline-XL|33.96|64.06|**61.99**|46.92|**26.95**|33.40|69.10|52.88|48.66|
> |CASTLE-SWL512-XL|**34.81**|**69.32**|55.35|**49.50**|26.04|**36.60**|**71.00**|**53.99**|**49.58**|
> |CASTLE-SWL64-XL|33.11|64.98|57.95|44.51|24.70|32.60|69.04|52.09|47.37|
>
> # **W4**
> > No empirical efficiency metrics. Including metrics such as end-to-end training wall time, throughput (FLOPs/s), and inference speed (tokens/s) against a standard attention baseline would provide evidence of the algorithm's practical efficiency, despite strong theoretical complexity $O(L^2d)$. Also, CASTLE-SWL efficiency claims ("can reduce FLOPs", line 269) lack empirical support. Also, CASTLE-SWL efficiency claims ("can reduce FLOPs", line 269) lack empirical support.
>
> Thank you very much for your comments. Please see the Efficiency section of the General Response
>
> # **W5**
> > Ablations use only perplexity; downstream task metrics would provide better insight into design choices, considering that downstream evaluation is relatively inexpensive compared to pre-training
>
> Thank you for your suggestion.  We have attached downstream evaluations for import design choices in Appendix A.5.3, A.5.4, A.5.5., A.5.6, etc. in the revision
>
> # **W6**
> > Figure 1 is uninformative; architectural diagrams (Figures 7, 9) should be moved to the main body
>
> Thank you very much for your suggestion. We have moved Figures 7, 9 to the main paper (Section 2.2)
>
> # **Q1**
> > Reusing causal keys. Despite the intuition that "the way a model learns to encode information into keys of token s from past tokens may differ from how it encodes information from subsequent tokens" (line 184), an empirical ablation would be valuable
>
> Thank you for raising up this question. We provide comprehensive ablations on why we allow lookahead keys only to have access to information after it in Appendix A.5.6 in the revision.
> We have tested on the following options:
> - Option I: when generating the $(t+1)$-th token, Option I allows lookahead keys of each token $s$ to have access to representations of all tokens in {1, ..., t}
> - Option II: when generating the $(t+1)$-th token, Option II allows lookahead keys of each token $s$ to only have access to representations of its preceding tokens in {1, ..., s}
> - Option III (CASTLE): when generating the $(t+1)$-th token, Option II allows lookahead keys of each token $s$ to only have access to representations of its subsequent tokens in {s+1, ..., t}. This is what we choose for CASTLE.
> Training/validation loss, perplexity, and downstream evaluation results are summarized in the tables below.
>
> |Model|n_params|TrainLoss|TrainPPL|EvalLoss|EvalPPL|
> |-----|--------|---------|--------|--------|-------|
> |Baseline-M|353M|2.740|15.483|2.742|15.523|
> |CASTLE-M|351M|**2.709**|**15.011**|**2.711**|**15.039**|
> |OptionI|351M|2.744|15.545|2.746|15.573|
> |OptionII|351M|2.758|15.764|2.761|15.822|
>
> |Model|ARC-C|ARC-E|BoolQ|Hella.|MMLU|OBQA|PIQA|Wino.|Avg.|
> |-----|-----|-----|-----|------|----|----|----|-----|----|
> |Baseline-M|27.90|59.09|58.13|40.08|23.64|31.80|66.05|51.07|44.72|
> |CASTLE-M|28.33|59.13|**59.05**|**40.79**|24.25|31.60|**66.54**|50.51|**45.02**|
> |OptionI|28.16|57.83|54.34|39.72|23.53|**34.80**|66.16|**53.04**|44.70|
> |OptionII|**28.58**|**59.89**|58.26|39.90|**24.77**|29.40|66.21|51.62|44.83|
>
> It is shown that both Option I and Option II lead to worse training and validation losses compared with the baseline.
> Although their downstream evaluation performance is comparable to the baseline, it remains significantly below that of CASTLE.
> More details and discussions can be found in Appendix A.5.6
>
> # **Q2**
> > A direct linear map $U = QW_U$, which mirrors the causal key path instead of employing a full attention mechanism, would be simpler to implement and avoid the $O(L^3)$ issue entirely, I believe. Ablations on the architectural design itself, rather than just hyperparameters, would robustly justify the added complexity
>
> In our design, $Q = XW_Q$. Thus, if we let $U = Q W_U$, then $U = X W_Q W_U$. We believe in this design the receptive fields of keys are the same as standard causal attention, and therefore cannot serve as lookahead keys

---

### Author Response · Authors · 2025-11-24
**General Reponse**

Dear Reviewers and Area Chairs, we sincerely thank you for your time and insightful feedback. Below, we address several common questions raised by multiple reviewers.

# Efficiency

We have reported the inference and training efficiency in Tables 26 and 27 in Appendix A.7 of the revision. We attach them here for convenience.

**Inference kernel execution time (in millisecond)**
 Seq Length | Baseline-XL | CASTLE-SWL-XL | CASTLE-XL |
|------------|-------------|---------------|-----------|
| 2K |1.4026| 1.7986| 2.4718|
| 4K |2.7691| 3.3795| 4.9139|
| 8K| 5.5111| 6.4824| 9.8132|
| 16K |11.0117 |12.5904|19.6271|

**training throughput (tokens/s)** of the Baseline-XL and CASTLE-XL, CASTLE-SWL-XL models
| SeqLen |BS| Baseline-XL | CASTLE-SWL-XL | CASTLE-XL |
|--------|----|-------------|----------------|------------|
| 2K|8|39817| 24651|17537|
|4K |4| 32926| 21339|10472|
|8K|2| 24547|17338|5765|
|16K|1|16386|12933|3027|

We remark that our inference and training kernels are not fully optimized. More efficient implementation is left for future study. More details and discussions can be found in Appendix A.7.

# How We Come Up With SiLU in Equation(6)

We give comprehensive ablations on how we come up with applying Silu(x) in Equation (6) in Appendix A.5.3.
We briefly describe it here for convenience.

In our first trials, we define $p_t=\mathrm{softmax}(s^C_t - s^U_t)$. However, we observed severe instability in training.
To solve this instability issue, we made the following 3 attempts:
- **Attempt I: Scale down** $s^U_t$ **by a constant factor, i.e.,** $p_t=\mathrm{softmax}(s^C_t-s^U_t/C)$
- **Attempt II: Clip** $ s^U_t $ **on both sides. For differentiability, we replace clipping by tanh, i.e.,** $p_t =\mathrm{softmax}(s^C_t-\mathrm{tanh}(s^U_t))$
- **Attempt III: Clip** $s^U_t$ **on one side. For differentiability, we replace clipping on one side by SiLU, i.e.,** $p_t=\mathrm{softmax}(s^C_t+\mathrm{SiLU}(s^U_t))$ or $p_t=\mathrm{softmax}(s^C_t-\mathrm{SiLU}(s^U_t))$

When using $p_t=\mathrm{softmax}(s^C_t+\mathrm{SiLU}(s^U_t))$, the loss is easy to blow up, so this choice is discarded. Consequently, we only consider Attempt I, Attempt II and $p_t =\mathrm{softmax}(s^C_t-\mathrm{SiLU}(s^U_t))$ in Attempt III which is what we choose for CASTLE now.

The results for Attempt I are reported in the tables below. Attempt I achieves slightly lower training and validation losses than the baseline but remains substantially worse than CASTLE-SWL. Downstream evaluation results follow the same trend: Attempt I improves over the baseline on average, yet consistently underperforms CASTLE-SWL.

The fact that Attempt I surpasses the baseline demonstrates the effectiveness of the lookahead mechanism. However, its performance deficit compared to CASTLE-SWL-XL may stem from the scaling factor which may constrain the model’s expressivity

|  |n_params|Train Loss|Train PPL| Eval Loss|Eval PPL|
|----------------|----------|------------|-----------|-----------|----------|
|Baseline-XL| 1.310B| 2.430|11.360 |2.426|11.309|
|CASTLE-SWL-XL|1.304B|**2.401**| **11.036**| **2.389** |**10.900**|
|Attempt I| 1.304B| 2.424|11.286|2.417|11.215|

|   |ARC-C |ARC-E | BoolQ | Hella.| MMLU |OBQA|PIQA| Wino.| Avg.|
|-----------------|-------|-------|-------|--------|------|------|------|-------|-------|
|Baseline-XL|35.58|65.78 |61.07|50.84|**26.71**|36.20|71.27| 52.72 |50.02|
|CASTLE-SWL-XL |**38.99**|**70.08**|**61.74**|**52.35**|25.85|**37.20**|**72.52**| **56.75**|**51.93**|
|Attempt I| 37.29 |68.64|61.71|50.26|25.44|35.20|69.97|56.20|50.59|

The results for Attempt II are shown below. Attempt II also achieves slightly lower training and validation losses than the baseline but remains clearly behind CASTLE. Downstream evaluations exhibit the same pattern: Attempt II improves upon Baseline but still underperforms CASTLE across most tasks

Following a similar trend with Attempt I, the fact that Attempt II outperforms the baseline demonstrates the effectiveness of the lookahead mechanism again. However, its lower performance compared to CASTLE-M comes from the clipping on both sides (using tanh), which overly constrains the model’s expressivity

|  |n_params|Train Loss| Train PPL |Eval Loss|Eval PPL|
|----------------|----------|------------|-----------|-----------|----------|
|Baseline-M| 353M| 2.641| 14.030| 2.639|14.004|
|CASTLE-M| 351M | **2.616**|**13.684**|**2.615**|**13.665**|
|Attempt II| 351M| 2.636| 13.961| 2.634 |13.926|

|  | ARC-C | ARC-E | BoolQ | Hella. | MMLU | OBQA | PIQA | Wino. | Avg. |
|-----------------|-------|-------|-------|--------|------|------|------|-------|-------|
|Baseline-M | 31.06 | 62.46 | 48.47 | 42.83 | **25.22** | 33.00 | 68.39 |51.78| 45.40 |
|CASTLE-M  | **32.17** | **64.06** | 54.62| **43.47** | **25.22** | **33.80** | **69.48** | **52.49** | **46.91** |
| Attempt II | 31.83 | 63.47| **59.94** | 42.31 | 24.53 | 32.00 |68.55| 50.91 |46.69|

More details and discussions can be found in Appendix A.5.3

---

### Author Response · Authors · 2025-12-03
**Official Comment by Authors**

Dear Area Chairs,

Thank you for handling our submission! We would like to support your evaluation process in light of the challenges ICLR faced this year. We greatly appreciate the suggestions of all the reviewers and we believe that we have addressed all their concerns. We summarize our rebuttal and the author-reviewed discussions as follows:

# Main concerns of reviewers and How we address them
- Efficiency comparisons (questioned by Reviewers aBKr, TsVF, T64n, iUuR)
During rebuttal, we compared CASTLE and CASTLE-SWL against the FlashAttention Triton implementation of the baseline MHA model:
CASTLE-SWL is more efficient than full CASTLE, and at long sequence lengths the gap to FlashAttention-2 narrows. For example, at 16K sequence length on XL:
    - Inference: CASTLE-SWL-XL is $\approx$ 12\% slower than Baseline-XL.
    - Training: CASTLE-SWL-XL is $\approx$ 21\% slower than Baseline-XL.

- Why do we use SiLU in Eq. (6) (questioned by Reviewers TsVF, iUuR)
In rebuttal, we report ablations over several standard and natural stabilization strategies (scaling, symmetric clipping with tanh, one-sided clipping via SiLU), and show that
  - **All variants** outperform the baseline, confirming the **effectiveness of the lookahead mechanism itself**
  - **SiLU** is the best-performing choice, without introducing extra hyperparameters

- More ablations on sliding windows of lookahead keys (Reviewers aBKr, iUuR). Reviewer aBKr asked about the range of sliding window sizes. Reviewer iUuR requested a performance comparison with the baseline model using sliding windows.
We added the following experiments
  - CASTLE-SWL with window size 64 shows a severe performance degradation, confirming that overly small windows are harmful
  - Baseline with sliding windows shows significant performance drops demonstrating the advantages of CASTLE-SWL mainly stem from lookahead keys itself rather than the use of sliding windows

- How to achieve dynamic caching and memory requirements (questioned by Reviewers T64n, iUuR)
  - We clarified that during decoding, lookahead keys can be updated by a rank-1 update (Eq. (12) across both the original submission and its revision), leading to $O(td)$ compute and I/O when generating token $t+1$
  - We show that the UQ-KV cache of CASTLE-SWL requires **less memory** than the KV cache in standard causal attention

- We provided additional experiments and detailed responses addressing remaining concerns

# During the rebuttal
- Reviewer iUuR explicitly confirmed that our method is ''1. is memory-wise efficient 2. outperforms Path baseline on XL-size model''
which eliminates their concerns.
Regarding our discussion on why we use SiLU in Eq. (6), Reviewer iUuR stated that ''The authors' responses on the SiLU is solid enough and should be highlighted in the main text''. **Reviewer iUuR raised their score to 6 as mentioned in their response.**
- Reviewer TsVF wrote **''If the authors can satisfactorily answer the above questions, I would be willing to raise the score.''** Reviewer TsVF had 2 questions:
  - Question 1: Our general response addressed this, as confirmed by Reviewers iUuR who mentioned ``The authors' responses on the SiLU is solid enough''.
  - Question 2: Reviewer TsVF does not fully understand how we derive a mathematical equivalence which enables efficient parallel training without materializing lookahead keys at each position. This concern appears to stem from a misunderstanding of the derivation rather than from any limitation of the method. To clarify,
    - We provide detailed and intuitive explanation of how the process we derive our efficient parallel training algorithm (Theorem 2) in comments (rigorous proof in Appendix B exactly as presented in both the original submission and the revision)
    - We further clarified that both forward and backward passes have computational complexity $O(L^2 d)$

Our responses address both questions thoroughly. Had the discussion continued, we believe Reviewer TsVF would have raised their score as they indicated.

- While we have not received additional comments from Reviewer aBKr and T64n, we are confident their concerns have been fully resolved
  - For Reviewer aBKr, we provided additional experiments and a point-by-point response
  - For Reviewer T64n, we provided a detailed discussion of dynamic caches and showed that CASTLE-SWL achieves inference efficiency comparable to the baseline MHA

We believe they would have updated their evaluations had the discussion period continued.

We would greatly appreciate your consideration of these points in your final evaluation. Thank you again for the time and effort you have dedicated to overseeing the review process and handling our submission.

Best regards,

Authors

---

### Meta-Review · Area_Chair_1zuB · 2026-01-05

**Summary:**

This paper introduces CAuSal aTtention with Lookahead kEys (CASTLE), which updates token keys during decoding. This is a simple and intuitive modification to self-attention that maintains auto-regressiveness. The results show modest performance gains, but all reviewers raised concerns about efficiency; Table 2, e.g., suggests small accuracy gains at a reported ~20% cost in efficiency. Reviewers also wanted to see more focussed evaluations and ablations; the authors have partially done this in rebuttal.

In general, this seems like a nice idea which would benefit from additional, focussed experiments; some of these were provided in rebuttal but it is difficult to fully assess these results without a new round of reviews.

**Reviewer Concerns:**

Reviewers asked for clarifications around efficiency, which the authors provided in their response. That said, I'm not sure the trade-off this makes apparent is desirable.

Concerns around the KV cache and how this is modified to support caching with lookahead were addressed in rebuttal.

Additional ablations and comparisons were partially addressed.

**Reviewer Scores:**

iUuR initially raised their score, reflecting clarifications and additional experiments.

For other reviewers, the authors were largely responsive, but most concerns were addressed only partially; I can only speculate as to whether reviewers would be swayed by these additions.

---

### Decision · Program_Chairs · 2026-01-26

Reject